## Perspective

volcanology

copper mining, volcanoes, ore deposits, geothermal energy

**Author for correspondence:**
Jon Blundy
e-mail: jonathan.blundy@earth.ox.ac.uk

# The economic potential of metalliferous sub-volcanic brines

Jon Blundy[1], Andrey Afanasyev[2], Brian Tattitch[3], Steve Sparks[3], Oleg Melnik[2], Ivan Utkin[2] and Alison Rust[3]

[1]Department of Earth Sciences, University of Oxford, South Parks Road, Oxford OX1 3AN, UK
[2]Institute of Mechanics, Moscow State University, 1 Michurinsky Prospekt, Moscow 119192, Russia
[3]School of Earth Sciences, University of Bristol, Wills Memorial Building, Bristol BS8 1RJ, UK

JB, 0000-0001-7263-8925; AA, 0000-0002-2284-7144;
BT, 0000-0002-2725-8031; SS, 0000-0001-7173-2899;
OM, 0000-0002-4655-5269; AR, 0000-0001-5095-7749

The transition to a low-carbon economy will increase demand for a wide range of metals, notably copper, which is used extensively in power generation and in electric vehicles. Increased demand will require new, sustainable approaches to copper exploration and extraction. Conventional copper mining entails energy-intensive extraction of relatively low-grade ore from large open pits or underground mines and subsequent ore refining. Most copper derives ultimately from hot, hydrous magmatic fluids. Ore formation involves phase separation of these fluids to form copper-rich hypersaline liquids (or 'brines') and subsequent precipitation of copper sulfides. Geophysical surveys of many volcanoes reveal electrically conductive bodies at around 2 km depth, consistent with lenses of brine hosted in porous rock. Building upon emerging concepts in crustal magmatism, we explore the potential of sub-volcanic brines as an *in situ* source of copper and other metals. Using hydrodynamic simulations, we show that 10 000 years of magma degassing can generate a Cu-rich brine lens containing up to 1.4 Mt Cu in a rock volume of a few km$^3$ at approximately 2 km depth. Direct extraction of metal-rich brines represents a novel development in metal resource extraction that obviates the need for conventional mines, and generates geothermal power as a by-product.

## 1. Introduction

Our modern world is reliant on natural resources of metals, from steel for construction to rare earth elements for high-tech devices. As we transition from fossil fuel-dominated to more sustainable

economies, demand for certain metals will rise dramatically. In a world powered by wind, sun and tides, for example, copper demand will increase more than fivefold, surpassing known global reserves well before 2100 [1,2]. The rise in demand for 'critical metals' (e.g. lithium, scandium, cobalt, rare earths) will be even greater [3]. As existing reserves become depleted and new deposits ever harder to find, it is unclear how the extra demand can be met; recycling alone will be insufficient [1].

Discovering natural mineral resources is, at heart, a geological problem. The majority of non-ferrous metals derive ultimately from igneous processes associated with magmatism. Non-ferrous metal ores can be viewed as extreme end-products of igneous geochemical cycles that begin in Earth's mantle and conclude with discharge of hot magmas and gases at the surface. Volcanic volatile phases are particularly efficient transporters of metals as evidenced by epithermal ore deposits and the chemistry of fumarole gases. Some active, subduction-related basaltic volcanoes discharge metal-rich gases with a time-averaged flux to the atmosphere of more than $10^4$ kg day$^{-1}$ of copper and zinc, along with a slew of other metals (e.g. silver, tungsten, indium, tin, lead, molybdenum) at fluxes of up to 1000 kg day$^{-1}$ [4]. This observation re-emphasizes the centrality of igneous processes to formation of mineral resources [5], and raises the possibility of recovering metals directly from modern volcanic fluids as an alternative to mining ancient solid ores. Here, we discuss the potential for *in situ* mining of hot, metalliferous volcanic fluids,[1] with a particular emphasis on copper, in light of recent developments in our understanding of the dynamics and architecture of crustal magmatic systems.

# 2. Porphyry copper deposits

Porphyry copper deposits (PCDs) provide approximately 75% of the world's copper, plus significant molybdenum and gold. PCDs are usefully subdivided into hypogene deposits that form primarily by the action of hydrothermal fluids, and supergene deposits that have been modified and enriched through reactions of hypogene ore at or close to the water table. Our focus here is exclusively on hypogene PCDs, on which there is a substantial body of literature; our intention here is not to review all of this prior work, but instead to summarize some key features of PCDs that place them in the context of crustal magmatic processes. For comprehensive summaries of hypogene PCDs the reader is referred to Hedenquist & Lowenstern [5], Sillitoe [6] and Richards [7].

PCDs are typically associated with subduction-related, oxidized, $H_2O$-rich silicic magmas, themselves derived by differentiation of mantle-derived, hydrous basalts [7]. Magmatic differentiation within the crust elevates contents of dissolved $H_2O$ and other volatiles, including important metal-complexing ligands, such as $Cl^-$ and $HS^-$, while imparting geochemical signatures characteristic of differentiation at lower- or mid-crustal pressures [8–11]. Unless the parental basalts are unusually Cu-rich, for which there is little evidence [12], a key process for Cu enrichment and ore formation is efficient extraction of Cu by hydrous fluids exsolved from evolved magmas as they ascend and crystallize. Such fluids are associated with active dacitic, andesitic and basaltic andesite volcanoes (e.g. [5,13–15]) both during and between eruptions of magma [16,17]. The fluids are predominantly $H_2O$ with lesser quantities of $CO_2$, $SO_2$, $H_2S$ and halogens and a wide range of trace metals [18,19].

The relationship between magmatic fluids involved in sub-surface PCD formation and those discharged directly to the atmosphere is complex. Phase changes associated with fluid decompression and cooling modify the chemistry of the original fluid at the point of exsolution from magma. For example, phase separation of $H_2O$-NaCl fluids into coexisting low-density (vapour) and high-density (liquid) fluids can have a profound effect on metal contents. To form an ore deposit, some component of the original magmatic fluid must become trapped and cool within the crust, depositing its metal load in the form of ore minerals. Copper mineralization typically takes the form of copper-bearing sulfides, such as chalcopyrite and bornite. Reduced sulfur ($S^{2-}$) is therefore a key ingredient in the formation of PCDs. Similarly, the primacy of the chloride ($Cl^-$) ligand in transporting Cu [20–25] requires that ore-forming magmatic fluids are saline, that is, they contain chloride, often expressed in terms of an equivalent amount of sodium chloride ($NaCl_{eq}$), noting that cations in addition to $Na^+$ can be present in significant concentrations (e.g. $K^+$, $Ca^{2+}$ etc.).

[1]We use the term *fluid* to describe any flowing geological phase, including volcanic gas, low-salinity vapour and hypersaline liquid, with a density less than that of silicate melt at the same pressure and temperature. *Supercritical fluids* are those beyond the critical end point in volatile-dominated systems, i.e. at pressures above phase separation into liquid and vapour. Magmatic *volatiles* include the species $H_2O$, $CO_2$, $SO_2$, $H_2S$, NaCl, HCl and other halides. The process of volatile release from silicate melts is described broadly as *degassing* regardless of the exact state of the fluid released.

Characteristic and extensive rock alteration haloes around PCDs, containing a variety of secondary minerals, such as sericite mica and clays (see review in [6]), testify to the highly reactive nature of hot magmatic fluids when they come into contact with cooler igneous rocks. The reactant fluids responsible for alteration include both primary, hot ore-forming fluids and cooler, acidic exhaust gases produced once mineralization has occurred. Heated external water (groundwater, seawater) may also mix with magmatic fluids and participate in hydrothermal reactions (e.g. [26]). Formation of extensive, hydrothermally altered 'lithocaps' above PCDs results from reactions involving fluids of different provenance [6]. The eventual style of alteration is a complex interplay between the temperature and composition of these fluids and their ascent paths.

PCDs are spatially associated with hypabyssal intrusions emplaced at depths of 1–4 km [6] that provide focused pathways for magmatic fluids exsolved deeper in the system [27]. These depths correspond to the upper reaches of magma reservoirs (or 'chambers') that underlie active volcanoes (e.g. [28]). The hypabyssal intrusions are sourced from these magma reservoirs; some intrusions are 'blind' (i.e. do not reach the surface), whereas others may represent magma-filled conduits for volcanic or phreatomagmatic eruptions, lava dome extrusions or brecciation events. Solidified shallow magma reservoirs take the form of plutons with a carapace of thermally metamorphosed, and variously hydrothermally altered, country rock.

# 3. Transcrustal magmatic systems

PCDs form in the upper crust and hitherto the focus of attention with respect to the magmatic sources of ore-forming fluids has tended to be a large, degassing shallow magma chamber (e.g. [27,29]). However, as the paradigm of crustal magmatism evolves it has become clear that processes pertinent to the origin of PCD-forming magmas and fluids extend over much greater depths. Shallow, melt-rich magma chambers are increasingly viewed as an ephemeral expression of much larger, vertically extensive, and longer-lived magma systems in which partially molten rocks, or 'mushes', dominate [30–32]. Such systems may extend throughout the crust as a continuous connected system or as discrete separated reservoirs that become transiently interconnected during magma ascent. The architecture of transcrustal magmatic systems has some important implications for ore formation.

In the transcrustal concept (figure 1) magmas and fluids stored in and released from shallow sub-volcanic reservoirs represent the time-integrated products of magmatic differentiation and fluid exsolution processes operating over a significant vertical extent. This is particularly important in the case of magmatic volatile species because of their strongly pressure-sensitive solubility in silicate melts [33,34]. Consequently, the mass and composition of the magmatic volatile phases exsolved from magmas are sensitive functions of the pressure (depth) at which degassing occurs.

The transcrustal, hot-zone concepts have arisen from studies in volcanic geology and stratigraphy, geochemistry, petrology, geophysics and geochronology, with valuable additional insights provided by numerical models [9,35–40]. In combination, these different approaches show how hot zones can form over hundreds of thousands to several million years through the sequential emplacement of mantle-derived magmas into the mid- or lower crust. Over time, temperatures and melt fractions in the hot zone increase, eventually forming a substantial, vertically extensive body of partially molten rock. The deep-seated, partially molten nature of hot zones beneath a thick insulating roof confers a longevity that is not tenable for shallow, liquid-rich magma chambers that are limited by thermal constraints [41]. Shallow large magma chambers are now considered ephemeral components of the magmatic system, associated with episodes of relatively high magma transfer from the hot zone into the upper crust. The generation of silicic melts in transcrustal systems seems to be focused in the middle crust where intermediate and mafic magmas that differentiated in the lower crust and mantle stall.

Hot zones are not static; they are dynamic mixtures of relatively dense solids and buoyant melts and fluids. Reaction of ascending melts and fluids with their surrounding rocks, including partially solidified ancestral magmas, is considered to be an important process in chemical evolution of magmas (e.g. [37,40]) and fluids (e.g. [42]). Reactive flow enables melts and fluids generated deeper in the system (i.e. at higher pressure) to encounter and react with shallow-stored (lower pressure) components of the same magmatic system. The intrinsic gravitational instability of mush systems means that melt and fluid ascent can be either percolative (steady state) or episodic [31]. Release of melts from hot zones may lead to eruption at the surface or formation of shallow-level magma chambers and plutons, themselves constructed incrementally over extended periods of time [43]. Release of highly reactive fluids from deeper parts of a magmatic system has a number of implications for ore deposits.

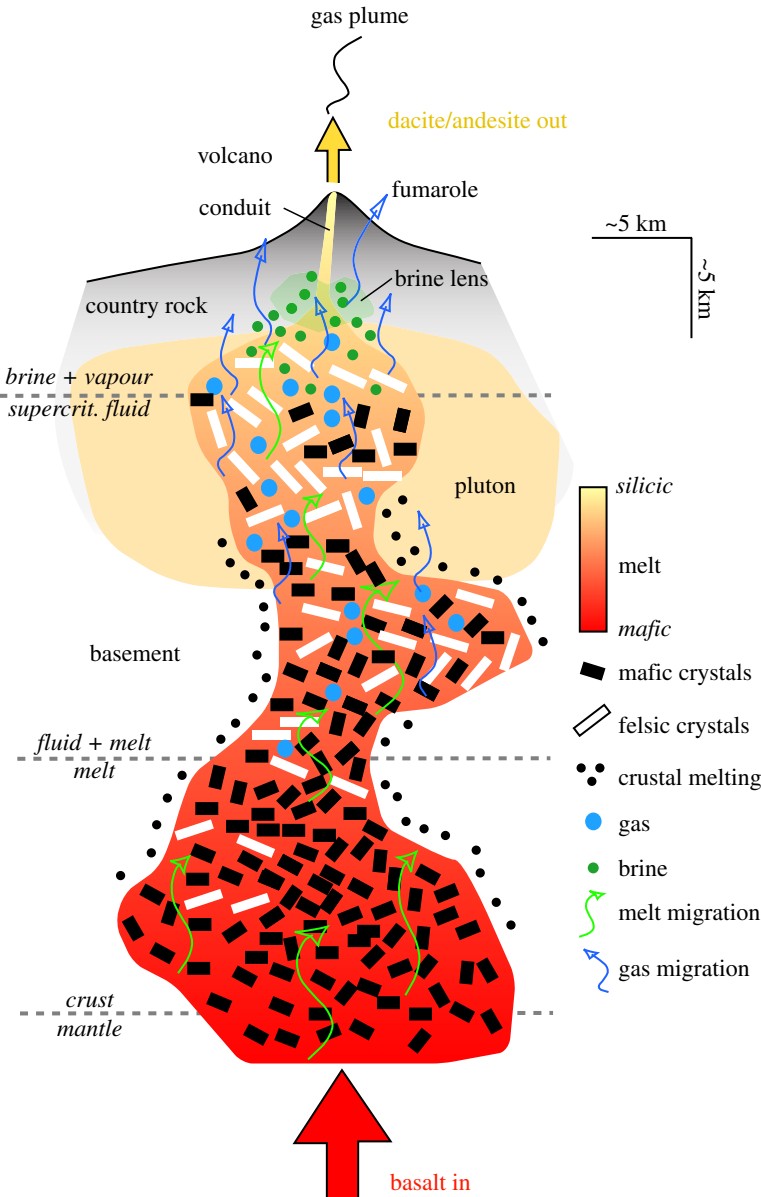

**Figure 1.** Schematic of a typical transcrustal magmatic system showing features described in the text. Note that proportion of melt and dimensions of crystals and bubbles are exaggerated for clarity. Ascent of melt (green arrows) and fluids (blue arrows) is decoupled, and may be either steady-state (percolative) or episodic. Crustal melting may occur in a thermal aureole around the magmatic system, as shown by black dots. Depths of the crust-mantle transition (Moho), volatile saturation and supercritical fluid phase-separation are denoted by horizontal dashed lines. Sub-volcanic brine lenses, the focus of this paper, occur at the shallow extremity of the system (pale green shading), created and sustained by fluid flow from the underlying system.

Fluids released from hot zones transport heat and fluid-mobile elements, including copper, upwards. The ascent of melt and fluid may be decoupled in time and space [17]. The endowment of the world's most economic PCDs (greater than 10 Mt contained Cu) requires that copper be extracted efficiently from a much greater volume of magma than the small intrusions associated with the deposits themselves. The abundance of brecciation and veining features in PCDs [6], and high-resolution radiometric data suggest that pulses of mineralization tend to occur at the end of relatively long-lived magmatic episodes (e.g. [44–47]), possibly via catastrophic, large-scale fluid release events. Thermal modelling of incrementally assembled igneous bodies shows that fluid release can occur in relatively brief pulses against a background of longer-lived magma accumulation and crystallization [11,48]. Ultimately the upward flux of melt and fluids, and the extent to which fluxes are steady-state or episodic, are controlled by the evolving permeability of the reservoirs in which they form [49].

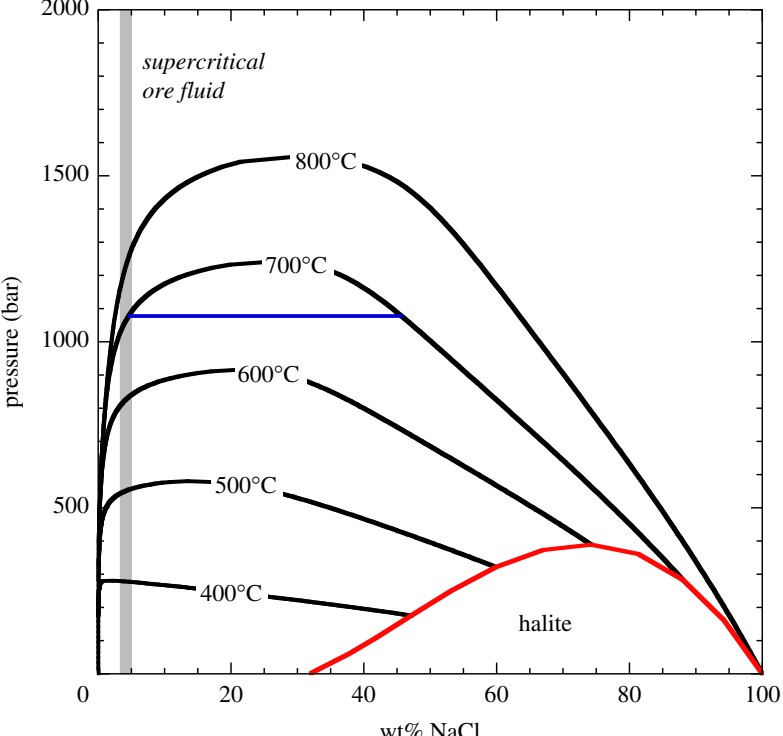

**Figure 2.** Phase relations in the system NaCl–$H_2O$ calculated using SOWAT [55]. Solid lines show isothermal projections of the solvus that describe coexisting hypersaline liquid (brine) and low-salinity vapour. Upon decompression a single-phase fluid of intermediate density will undergo phase separation when it encounters the solvus. The compositions of coexisting phases depend on pressure and temperature; the compositions of coexisting vapour and hypersaline liquid at 1100 bars and 700°C are shown for reference by the horizontal blue line. An illustrative supercritical magmatic input fluid with 4 wt% $NaCl_{eq}$ (as used in the models) is shown by the vertical grey bar. The halite saturation field is delineated in red. Addition of $CO_2$ to the NaCl–$H_2O$ system expands the solvus and extends the region of phase separation.

Currently, this is a poorly understood aspect of the multiphase, mushy systems that are thought to underlie volcanoes, but is likely to be very important in the context of giant ore deposits.

## 4. Magmatic volatiles

The composition of magmatic fluids depends on the relative solubilities of the key volatile species in silicate melts that in turn depend on pressure, redox state and composition (and to a much lesser extent, temperature). Generally speaking, fluids released deeper in the system are richer in $CO_2$ than those released at shallower levels, due to the greater solubility of $H_2O$ than $CO_2$ (e.g. [34]). Thus, fluids released from magmas at depth in the crust will differ chemically from fluids in equilibrium with magma stored at shallow depth, with abundant opportunities for chemical reaction, including flushing of shallow magmatic systems with deep-derived $CO_2$, for which there is compelling petrological evidence (e.g. [50]). The behaviour of minor fluid species (e.g. NaCl, HCl, $SO_2$, $H_2S$) is more complex and, in the case of sulfur species, dependent on redox state. For example, in their experimental study of degassing of oxidized basalt magma Lesne *et al.* [33] show that magmatic chlorine is retained in melts to lower pressures than is $SO_2$.

The volcanic fluids emitted from active basaltic volcanoes are dominated by $H_2O$ ($\pm CO_2$) and have relatively low salinity (less than 1 wt% $NaCl_{eq}$; [4,51]); their composition at the surface closely resembles that of the fluid at the point of shallow exsolution. The situation is different for magmatic fluids that exsolve from evolved, more chlorine-rich silicic magmas at greater depths where higher salinity (greater than 1 wt% $NaCl_{eq}$) fluids are generated [52,53]. Thermodynamics of the NaCl–$H_2O$ system [54,55] define a two-phase region of hypersaline liquid (often referred to as 'brine') and low-salinity vapour. The two-phase region is delimited by a solvus whose location depends on temperature and pressure (figure 2). At the pressure and temperature of initial volatile exsolution

from $H_2O$-rich silicic magmas, most magmatic fluids probably exist as a single, super-critical fluid of intermediate density (salinity approx. 2 to 12 wt% $NaCl_{eq}$; [24]) due to volatile saturation at pressures above the $NaCl$–$H_2O$ solvus [29]. However, direct exsolution of high salinity fluids from some silicic magmas is known to occur [56–59]. The $NaCl$–$H_2O$ solvus is influenced by the presence of other volatile species, notably $CO_2$ [60,61], which greatly expands the two-phase field. In these cases, not considered explicitly here, phase separation of hypersaline liquids will occur over a much wider range of pressure–temperature conditions than for the simple $NaCl$–$H_2O$ system, further expanding their stability field. Other cations ($K^+$, $Ca^{2+}$ etc.) also modify phase relations and ore mineral solubilities in hydrous fluids, although in most mineralizing magmatic systems $Na^+$ is by far the most abundant.

Fluid inclusions from PCDs support involvement of parental, single-phase intermediate-density (ID) fluids of modest salinity (4–10 wt% $NaCl_{eq}$; [62–65]). Although primary ID fluids typically contain less than or equal to 2000 ppm Cu ([24]), the proclivity of copper (and many other metals) for the chloride ligand means that the separated hypersaline liquid phase becomes strongly metal-enriched relative to its parent fluid. Upon phase separation, Cu partitions into the liquid, in almost direct proportion to the salinity ratio between liquid and vapour [24,25]. This ratio depends on the pressure and temperature of phase separation (figure 2), thus higher Cu contents are found in higher salinity liquids formed at lower pressures and higher temperatures. Globally, Cu contents of hypersaline fluid inclusions (salinities of 25 to 73 wt% $NaCl_{eq}$) from PCDs range from 3 ppm to 3 wt% [24]. The upper end of this range is considerably more Cu-rich than the original fluid exsolved from the parent silicic magma, a testament to the Cu-sequestering ability of brine. The low salinity vapour counterpart that discharges to the atmosphere is correspondingly depleted in metals.

The importance of brines to ore formation has long been recognized (e.g. [66,67]). Numerical models of magmatic fluid flow through the shallow crust [68–70] show how the dense, hypersaline liquid formed upon phase separation can be retained at depth in rock pore space, such as intergranular pores, cavities and fractures. Thus a significant fraction of the original metal budget of silicic volcanoes can become trapped within fluids in porous rocks at depth, rather than be discharged to the atmosphere or retained in the cooled crystallized mass. These considerations raise the possibility of the existence of metal-rich hypersaline liquid reservoirs beneath active volcanoes with attendant economic potential. Although few, if any, major PCD-like deposits are likely to underlie modern volcanoes, the ubiquity of active volcanoes (and associated shallow intrusions) may partly counterbalance their small size, justifying a careful evaluation of the economic potential of their associated brines.

# 5. Metals in geothermal systems

The heat and fluids associated with shallow magmatic systems are often harnessed through several forms of geothermal energy production. Geothermal energy is extracted from a range of volcano types and in a variety of tectonic settings. Conventional geothermal wells are drilled into the upper reaches of magmatic hydrothermal systems, and provide insights into the composition and distribution of fluid phases. Typically, these fluids are dominated by a variety of non-magmatic fluids, including sea water, meteoric water and regional brines, variably modified by heating, water–rock interactions, phase separation and mixing, with relatively limited chemical input from magma-derived fluids, as evidenced, for example, by stable isotope chemistry [71].[2] This is because, in general, the region of active hydrothermal convection is separated physically from the conductive magmatic heat source by a seal of relatively low permeability rock. This physical barrier is often ascribed to the brittle–ductile transition in rocks at depth at temperatures of around 400°C, although permeability reduction can also be caused by mineral precipitation (e.g. [67,72]). Below the rock-seal, fluids are predominantly magmatic in origin; above it they are predominantly non-magmatic. Evidently, fluids of magmatic origin and those of meteoric origin can be efficiently isolated from each other in geothermal systems; most geothermal wells access only the shallower, convecting hydrothermal realm. The upper boundary of this realm is defined typically by a second impermeable barrier, often referred to as the clay cap (e.g. [71]).

A lesser number of high-temperature or 'supercritical' geothermal systems [73,74] use wells that penetrate higher temperature (greater than 400°C) accumulations of magma-derived fluids close to or below the interface between hydrothermal and magmatic realms. In a few of the deepest wells, drilled into young, hot magmatic systems, samples of true magmatic hypersaline liquids have been

---

[2]Although it is common in many of these geothermal systems to refer to any fluid with total dissolved solids (TDS) greater than 0.1 to 1 wt% as 'brine', these are distinct from the truly hypersaline liquids that are our focus here.

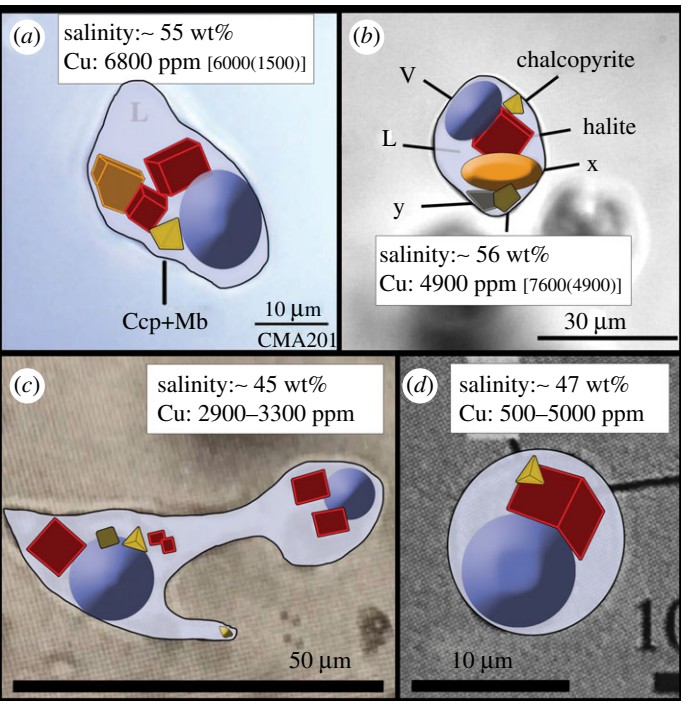

**Figure 3.** Three-dimensional phase maps of experimental and natural metal-bearing, hypersaline fluid inclusions used to estimate original copper contents. Maps were generated from photomicrographs of hypersaline liquid (L) inclusions from: (a) experiments [25]; (b) a porphyry copper deposit (Bajo de la Alumbrera; [85]); and the Kakkonda deep geothermal system (c, [84]; d, [83]). All inclusions show large halite daughter minerals due to high salinity, along with numerous opaque daughter sulfide minerals (Ccp, chalcopyrite; Mb, molybdenite; x and y are unidentified phases). Inclusions (a) and (b) come from systems with positive confirmation of chalcopyrite daughter minerals and known Cu contents. Pixel maps of the inclusions yield volume estimates for halite (red), sylvite (orange), and opaque daughter crystals along with total volume estimates based on assuming elliptical inclusion geometries and/or thicknesses given by vapour bubble (V) size and shape. Salinity estimates (expressed as wt% $NaCl_{eq}$) for all inclusions based on volume estimates roughly match those reported from fluid inclusion microthermometry. Estimates of the possible volume of chalcopyrite (yellow) (34% Cu) or Cu-bearing Fe-sulfide (brown) (approx. 3.5%) have been compared with total fluid mass in the inclusion to determine the range of Cu concentrations (given in ppm; table 1). For comparison, Cu concentrations from LA-ICPMS analyses of inclusions in (a) and (b) are given in square brackets (with 1 s.d. uncertainty in curved brackets), and are a reasonable match to the estimated Cu contents. No explicit measure of uncertainty is considered but large variations in inclusion thickness from those modelled would result in unrealistic deviations in estimates of inclusion salinity when compared with the measured values.

recovered. At Larderello (Italy) Cathelinau *et al.* [75] report quartz-hosted hypersaline liquid inclusions (30–70 wt% $NaCl_{eq}$) recovered from granite host rocks at depths of 2.7–3.6 km and at temperatures of 425–650°C, overlapping the highest measured bottom-hole temperatures (330–450°C). The studied samples come from several hundred metres above the seal between convective and conductive regimes (the 'K-horizon'; [76]).

At Kakkonda (Japan) active accumulations of hypersaline liquids (39 to 55 wt% $NaCl_{eq}$) have been identified at depths in excess of 3708 m and temperatures over 500°C in well WD-1a [77,78]. These depths lie below the conductive–convective interface (3.2 km in WD-1a; [79]) and consequently represent truly magmatic fluids. Fluids recovered directly from the well-bore contain 3400 ppm Zn and 1200 ppm Pb, with estimates of greater than 5000 ppm Zn in the original (approx. 55 wt% $NaCl_{eq}$) hypersaline liquid [77,78,80]. Copper, however, is at concentrations below detection in the recovered hypersaline fluids; this is thought to be due to rapid Cu-sulfide precipitation during cooling and decompression of fluids in the well-bore prior to sampling [77]. Measurements of fluid inclusions from Kakkonda drill-core indicate temperatures above 500°C and salinities above 50 wt% $NaCl_{eq}$ [81] consistent with sampling conditions of the well-bore fluids.

The Larderello and Kakkonda examples indicate that fluid inclusions from drill-core can provide reliable insights into the temperature and chemistry of deep geothermal fluids. Unfortunately, there are currently no direct trace metal analyses of fluid inclusions from Kakkonda or Larderello. However, insights into their chemistry can be gained from published petrographic studies. Detailed examination

of quartz-hosted fluid inclusion images from Kakkonda [82–84] reveals that they contain abundant opaque daughter minerals indicative of substantial chalcophile metal concentrations (figure 3a–d), similar to those observed in metalliferous hypersaline liquid inclusions from experiments [22,25] and from ore deposits (e.g. [85]). From the images, we can estimate the bulk composition of the fluid inclusions using pixel maps and volume calculations for cubic halite crystals, tetrahedral or cubic sulfides (chalcopyrite or pyrite/pyrrhotite) and spherical bubbles, compared with the remaining liquid volume of the inclusion. For fluids with known salinities and Cu contents (figure 3a,b), a combination of volume calculations with densities of observed solid phases recovers salinities (45–60 wt% $NaCl_{eq}$) and Cu contents (5000–7000 ppm) that lie within 1 sigma of the average for the range measured by laser-ablation inductively coupled mass spectrometry (LA-ICPMS) on the same inclusions. This gives us confidence in our methodology when applied to images of fluid inclusions with unknown Cu concentrations. Assuming, on the basis of crystal shapes, that the opaque phases from Kakkonda are a mixture of chalcopyrite (34% Cu) and pyrite/pyrrhotite that is saturated in chalcopyrite (approx. 3.5% Cu), we estimate Cu concentrations in the range of 500–5000 ppm (figure 3c,d and table 1). No explicit measure of the uncertainty on these Cu estimates is possible. However, any large variations in inclusion thickness would create a mismatch between the calculated inclusion salinity and that reported. Our estimated values are not meant to define accurately the Cu concentration, but simply to show that fluid inclusion petrography is consistent with Cu concentrations at levels similar to those observed in magmatic brines from PCDs.

Direct analyses of fluid inclusions from high-temperature, magma-driven geothermal systems, such as those described here, by LA-ICPMS can provide much more accurate constraints of the contents of Cu and other metals. To date, very few such analyses have been reported in the literature. One set of fluid inclusion LA-ICPMS data is available for moderate temperature (approx. 400°C) Kakkonda hypersaline liquids [82]. Measured Zn/Cu $\approx$ 7 for some of the assemblages. Combined with other direct measurements and estimates of Zn in the recovered fluids [77], this corresponds to Cu concentrations from approximately 500 ppm to approximately 1500 ppm in the undisturbed high-temperature hypersaline liquid reservoir at the well-bottom (table 1). These values are probably minima given the evidence for substantial precipitation of Cu, and later Zn, as the fluid migrates through its host rocks and cools (e.g. [82,84]). Copper loss from fluids through precipitation of sulfides was exacerbated during well-sampling at Kakkonda, consistent with the lack of measurable Cu in fluid samples extracted to the surface [77,82,84], and the occurrence of abundant Cu-rich well-bore scales [86]. Scales collected from variable depths in drill holes #13, #19 and #21 contained chalcocite, Cu-Fe arsenides, sphalerite, galena and pyrite; each scale also contains significant, but variable, amounts of silica-alumina phases, sulfates and carbonates [84]. All of these mineral phases are likely to have precipitated directly from hot, ascending fluids, testifying to their metal-rich character. Cu contents of the analysed Kakkonda scales are in the range 0.27 to 14 wt% Cu, with highly variable Zn/Cu ratios of 0.002 to 74 [86].

The Kakkonda example of deep geothermal drilling suggests that high-temperature, magmatic brines can have significant base metal concentrations, including Cu, consistent with the composition of hypersaline liquids observed in porphyry ore-forming environments. Fluids recovered from Kakkonda well WD-1b, at depths of greater than or equal to 2625 m, have lower salinity than those from WD-1a, and show stable isotope evidence for dilution of magmatic fluids with meteoric water [87]. Mixing of meteoric water or seawater with magmatic fluids is evident in other geothermal reservoirs drilled to shallower depths and lower temperatures, such as Geysers (USA; [88]) and Mori (Japan; [89]). The mixing process probably dilutes base metal contents as well as salinity, but no fluid trace metal analyses are available for these hybrid fluids.

# 6. Analyses of geothermal brine fluid inclusions

In order to explore further the metal content of shallow magmatic brines, we have performed analyses of quartz-hosted fluid inclusions from two geothermal wells associated with young magmatic systems: Larderello, Italy; and Montserrat, Eastern Caribbean. The Larderello-Travale geothermal field is associated with late Miocene, post-orogenic crustal extension in the Northern Apennines of Tuscany. Geothermal heat, provided by a suite of 3.8 to 1.3 Ma anatectic granite intrusions [90], has been exploited at Larderello since the early nineteenth century. It is unclear whether Larderello itself was ever connected to a volcanic edifice at the surface, although contemporaneous rhyolitic volcanic rocks outcrop nearby at San Vincenzo and at Roccastrada [90], and the 0.3 to 0.2 Ma Mount Amiata volcano

**Table 1.** Composition and trapping conditions of quartz-hosted brine inclusions from Montserrat, Larderello and Kakkonda.

| location | Soufrière Hills, Montserrat | | | | Larderello, Italy | | | Kakkonda, Japan | |
|---|---|---|---|---|---|---|---|---|---|
| sample no. | MSV01-H[a] | MSV01-HB[a] | MSV01-M[a] | MSV02-M[a] | Trav1s-B[a] | Rad26-B[a] | Rad26-M[a] | [85] | [84] |
| core depth | 1481 m | 1481 m | 1481 m | 1484 m | 4100 m | 4600 m | 4600 m | 2937 m | 3729 m |
| well temp. | 210°C | 210°C | 210°C | 210°C | >325°C | 300–360°C | 300–360°C | 370°C | >500°C |
| $n$[b] | (9):(5) | (10):(4) | (12):(5) | (9):(7) | (12):(8) | (31):(11) | (11):(7) | (33)[c] | (11)[c] |
| $T_m^{Halite}$ (°C) | 280–330[e] | 410–450[e] | 415–470[e] | 415–450[e] | 190–280[e] | 230–275[e] | 300–375[e] | 151–462 | 194–322 |
| $T_h^{(L-V)}$ (°C) | 200–250 | 290–350 | 590–670 | 550–650 | 325–385 | 400–450 | 440–520 | 339–562 | 483–611 |
| salinity (NaCl$_{eq}$)[f] | 36–40 | 41–54 | 49–56 | 45–55 | 30–36 | 33–46 | 38–54 | 35–55 | 35–45 |
| $T_h$ | 280–330[g] | 410–450[g] | 590–670 | 550–650 | 325–385 | 350–450 | 440–520+[h] | 339–562 | 483–611 |
| Na[d] | 53 200 (11 200) | 79 300 (2600) | 57 800 (7600) | 73 600 (33 000) | 103 500 (17 500) | 116 300 (23 200) | 82 100 (39 800) | 68 000–81 000[i] | |
| K | 57 200 (7600) | 73 200 (7500) | 103 200 (23 000) | 105 100 (48 000) | 74 500 (12 000) | 61 000 (16 300) | 121 200 (39 200) | 42 000–50 000[i] | |
| Ca | BDL | BDL | BDL | BDL | 54 700 (13 700) | 53 700 (11 700) | 36 800 (35 700) | 19 000–22 000[i] | |
| Fe | 62 400 (2900) | 82 100 (4300) | 82 200 (18 200) | 52 800 (21 200) | 8400 (3200) | 6600 (1100) | 55 400 (37 900) | 49 000–59 000[i] | |
| Cu | 6.2 (5.6) | 150 (50) | 2700 (700) | 1200 (300) | 14.1 (9.4) | 400 (200) | 2500 (1100) | 2900–3300[j] | 500–5000[j] |
| Zn | 1300 (300) | 2200 (1100) | 1800 (800) | 2400 (2100) | 1200 (500) | 1500 (400) | 8700 (7300) | 3100–4200[j] | |
| Pb | 400 (100) | 600 (300) | 400 (200) | 500 (400) | 500 (200) | 500 (100) | 2100 (1100) | 1000–1400[j] | |

[a]Fluid inclusions are designated as halite-saturated (-H), hypersaline (-B) and magmatic (-M) brine.

[b]Number of analyses, expressed as (microthermometry):(LA-ICPMS).

[c]Representative microthermometry data for inclusions from these WD-1a core depths are reported in [82].

[d]All compositions reported as ppm with 1 s.d. in parentheses; BDL denotes below detection.

[e]Halite was often the only phase that could be assigned a precise dissolution point. Occasionally another phase (presumed sylvite) could be observed dissolving between 80–150°C.

[f]Fluid salinity (wt%) was determined by using either halite dissolution or halite and sylvite dissolution. Corrections to actual wt% salt accounting for K and Fe have been made iteratively.

[g]Some Montserrat inclusions homogenized by halite disappearance which may indicate trapping under halite-saturated conditions or pressure fluctuations in the brine reservoir.

[h]Decrepitation of inclusions prevented homogenization of the highest temperature brine trails. Based on rate of bubble shrinkage $T_h$ is probably greater than 550°C for highest temperature brines.

[i]Estimated range for major cation chemistry calculated using the recovered fluids from [77] and their estimated dilution factors to back-calculate to brines of 45–55 wt% salt.

[j]Concentrations for base metals calculated based on recovered fluids (Zn and Pb) and by image analysis of brine inclusions (Cu, figure 3) due to expectation of Cu precipitation on recovery.

(and associated geothermal field) lies approximately 50 km to the southeast [76]. By contrast, Montserrat, in the Lesser Antilles volcanic arc, is associated with active westwards subduction of the Atlantic seafloor beneath the Caribbean Plate. Montserrat volcanic centres have previously erupted a range of magma types from basalt to dacite. The island's Soufrière Hills volcano was almost continuously active from 1995 to 2013 [91] erupting andesite lava domes with associated pyroclastic flows. Montserrat is currently the subject of active geothermal exploration [92].

## 6.1. Samples

We studied two core samples taken at 4100 and 4600 m depth from, respectively, geothermal wells Travale 1 sud and Radicondoli 26 in the eastern sector of Larderello geothermal field [58,93,94]. The sample context for Travale 1 sud is described in detail by Fulignati [58]. Briefly, the well lies on 455 m of dolomitic limestone and anhydrite of the Tuscan Nappe, underlain to 1630 m by Palaeozoic phyllites and phyllitic sandstone with intercalations of anhydrites and limestone. From 1630 m to 2680 m, the well passes through intercalated hornfels and skarn in contact with a granite intrusion at 2680 m that extends to the bottom of the hole at 4150 m, where temperatures reached approximately 350°C [93]. The intrusion is a leucogranite comprising quartz, plagioclase, K-feldspar, biotite, cordierite and tourmaline. The Radicondoli 26 well, roughly 5 km to the northwest of Travale 1 sud and in a similar setting, is described by Carella *et al*. [94]. The well passes through limestones until 1350 m depth followed by phyllite and calc-silicate intercalations until intersecting a small number of metabasite layers at 2350 m. Phyllite gives way to hornfels by 3700 m before reaching granite at approximately 4000 m at temperatures greater than or equal to 300°C. Quartz crystals from granites of Travale 1 sud and Radicondoli 26 were prepared for fluid inclusion study.

The Montserrat samples are from core recovered from depths of 1480 m from the MON-1 well, one of two exploratory geothermal holes drilled into the western flank of Soufrière Hills volcano [95,96]. MON-1 well was drilled through a 530 m thick sequence of andesite lava flows and breccias related to the young Soufrière Hills volcanic complex overlying a sequence of sandstones, mudstones and clays (530–1210 m depth). These in turn overlie silicified tuff, breccias and sedimentary rocks, including limestones, interpreted as older Centre Hills volcanic units and sediments deposited on the volcano's flanks, that extend to the bottom of the hole at 2298 m depth. The studied intervals are tuffs from this lower volcanic segment. Fluid inclusions were identified in small quartz crystals (less than 3 mm) filling pore spaces in two core samples, on loan from the British Geological Survey, designated MSV01 (MVOGP002-SK77147-1480.8 m) and MSV02 (MVOGP006-SK77151-1484.05 m). The present-day well temperature at the sampled depth is 210°C [92].

## 6.2. Methods

Doubly-polished sections of the quartz host minerals (less than 500 µm thick) were thinned until the fluid inclusion trails of interest were within 50 µm of the surface. Some trails could be matched to growth bands in the host minerals, but many could not be distinguished as primary or secondary due to the limited number of inclusions available at this preliminary stage. Microthermometric measurements were made on brine fluid inclusions using a Linkam TS1400XY stage at University of Bristol. Halite dissolution measurements were made on all inclusions, whereas sylvite dissolution was only observed and measured in some inclusions. Homogenization measurements were made for inclusions that homogenized by halite disappearance as well as by vapour bubble disappearance in different fluid inclusion assemblages. Halite disappearance temperatures were used to estimate total salinity (wt% $NaCl_{eq}$) using the SALTY salinity model [97] and were adopted as an internal standard in subsequent LA-ICPMS analyses. Homogenization temperatures were used to help separate out groups of inclusion types within the samples where other determinations of fluid generations were not possible.

The major and trace element compositions of fluid inclusions were determined by LA-ICPMS at the Open University using an Agilent 8900 QQQ ICPMS and a GeoLas 193 nm excimer laser. The 'triple-quadrupole' mass spectrometer allows for near-simultaneous analysis of the relevant major elements in the brines (Na, K, Ca, Fe) along with several trace metals of interest (Cu, Zn, Pb) [98–100]. Analyses were run using an ARIS (Aerosol Rapid Introduction System) from Teledyne CETAC designed for the Agilent 8900 that allows for rapid flushing of the sample cell and better signal-to-background ratio. This helped to ensure good sensitivity for the trace metals despite the small size of the inclusions. NIST610 was used as a primary element standard and to monitor for instrument drift.

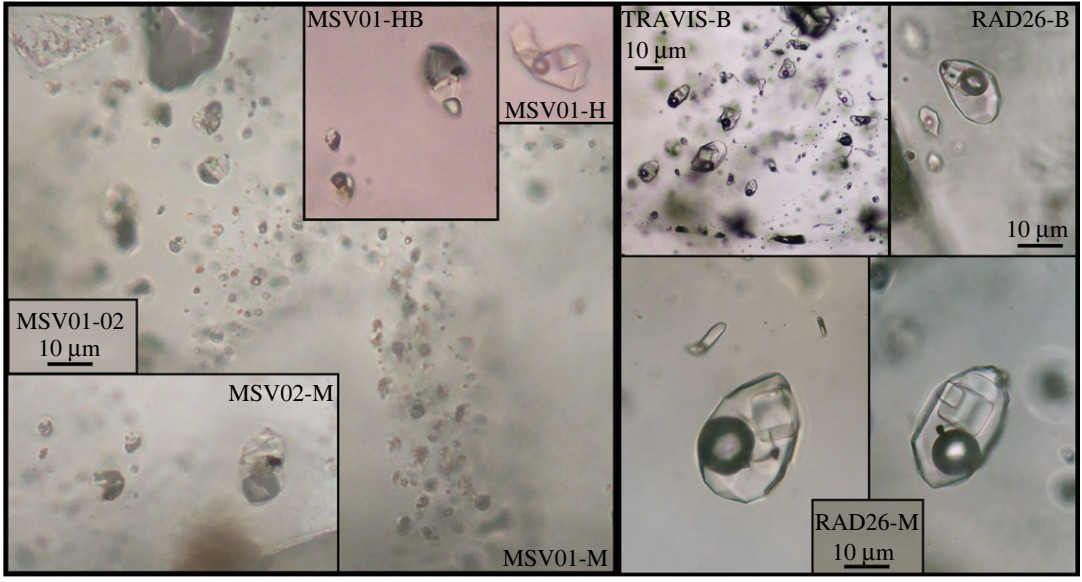

**Figure 4.** Photomicrographs of representative, quartz-hosted hypersaline (brine) fluid inclusions recovered from drill-core from Soufrière Hills volcano, Montserrat (left) and Larderello geothermal field, Italy (right). For sample details, including depths and well-bore temperatures, see table 1 and text. Brine inclusions in quartz were observed both along apparent crystal growth boundaries and as secondary fracture fill. The brine inclusions contain halite (and occasionally sylvite) daughter minerals and approximately 20–40 vol% vapour bubble. Opaque daughter minerals are readily identified in all but the lowest temperature inclusions trapped under halite-saturated conditions (−H). Some opaque minerals have a triangular shape, and are inferred to be chalcopyrite. Microthermometry data and LA-ICPMS analyses of these inclusions are given in table 1; histograms of microthermometric results are given in electronic supplementary material, figure S1.

Only inclusions without significant evidence of necking, and lacking other inclusions within the analytical volume, were selected for analysis [101]. An energy density of approximately $10 \, \text{J cm}^{-2}$ and a pulse rate of 10 to 50 Hz were used to ablate through the overlying quartz host and into the individual inclusions while stepwise increasing spot size from 10 µm to just larger than the inclusion diameter. Element ratios (Na/K/Ca/Fe) for individual inclusions were determined by integrating the complete analytical signal. The salinity of the inclusions, as determined by microthermometry, was then used as internal standard to balance (as $NaCl_{eq}$) the Na, K, Ca and Fe present in the inclusions and so calculate absolute abundances of Cu, Zn and Pb from element ratios determined by LA-ICPMS. All individual fluid inclusion analyses from each group were averaged to generate the reported composition of the brines.

## 6.3. Results

Representative data are presented in table 1; histograms of all fluid inclusion analyses are provided in electronic supplementary material, figure S1. The samples contained a range of fluid inclusion types including low-density vapour inclusions (vapour bubble greater than 60 vol%), low-temperature aqueous inclusions (vapour bubble less than 20 vol%), melt inclusions (glass ± bubble), as well as the hypersaline brine inclusions of interest. All hypersaline inclusions had vapour bubble fractions of approximately 20–40 vol% and contained halite daughter minerals along with other daughter minerals inferred to be sylvite and other chlorides (figure 4). Haematite and opaque daughter minerals (inferred to include pyrite, magnetite and chalcopyrite) were also observed in many inclusions. Halite disappearance temperatures (+/− sylvite disappearance temperatures) for the brine inclusions yield salinities ranging from 30 to 56 wt% $NaCl_{eq}$. Samples MSV01-H, MSV01-B (table 1) showed homogenization via halite disappearance and may have been trapped under halite-saturated conditions. The other samples showed indications of near-magmatic trapping conditions with homogenization to a liquid (brine) at temperatures from 500 to 670°C, indicating brines were initially present at much higher temperatures than the current reservoir, 210°C (table 1). At Larderello, fluid inclusion homogenization temperatures range from 325°C to more than 520°C, which overlaps the upper end of the measured reservoir temperatures (300 to 360°C; [58]).

Major element abundances (Na/K/Ca/Fe) and trace metal analyses (Cu-Zn-Pb) for the analysed brine inclusions are reported in table 1. All of the hypersaline brine inclusions have a broadly similar Na/K ratio within each sample, ranging from 0.6 to 1.1 for Montserrat and 0.7 to 1.9 for Larderello. Brines from Montserrat are slightly richer in Fe than those from Larderello, except at the highest temperature (Rad26-M) where Fe contents are similar. Conversely, Larderello brines contain appreciable (4 to 5 wt%) Ca, while those from Montserrat have Ca contents below detection; this is probably due to differences in the composition of the melt from which the fluids exsolved and the composition of the host rocks in which the fluids were trapped. Overall, ore metal contents are similar in both sets of brines, and are comparable to those estimated above for Kakkonda brine inclusions (table 1). At Montserrat, ore metal contents increase significantly with increasing temperature up to 0.24 wt% Cu, 0.27 wt% Zn and 0.05 wt% Pb. At Larderello, ore metal contents show a less obvious progression with temperature. The most enriched brine inclusions, from sample Rad26-M (with 0.25 wt% Cu, 0.87 wt% Zn and 0.21 wt% Pb), probably record the highest temperatures (greater than 520°C) from Larderello, although decrepitation precluded accurate $T_h$ determination (table 1). The increase in Cu content of the brines from approximately 10 ppm at 310°C to approximately 2750 ppm at 630°C is broadly in line with the temperature dependence of chalcopyrite solubility at elevated salinities ([24]; and below). The Cu, Zn and Pb contents of the analysed brines from these young active geothermal systems lie within the range of those of brine fluid inclusions associated with PCDs as compiled by Kouzmanov & Pokrovski [24]: e.g. 0.02 to 1 wt% Cu, 0.1 to 1 wt% Zn and 0.03 to 0.9 wt% Pb at 40 wt% $NaCl_{eq}$.

Our representative fluid inclusion analyses indicate that hot, metalliferous brines existed in both systems at the time fluid inclusions were trapped, and may still be present today as intergranular fluids of the type observed in the Kakkonda well-bore. However, in detail, the metal endowment of brines varies from one field to another, probably in relation to tectonic setting or other magmatic factors controlling the chemistry (salinity, metal content) of the input fluids. At both Montserrat and Larderello, the brine inclusion homogenization temperatures are higher than recorded bottom-hole temperatures suggestive of cooling of the reservoir since the time that the inclusions were trapped. Cooling is associated with a marked decrease in Cu contents. Montserrat and Larderello wells do not penetrate the base of their active hydrothermal reservoirs, thus neither of these examples directly accesses any current reservoir of hot, predominantly magmatic brines. Consequently, in both locations the brine metal contents of the inclusion fluids probably have been diluted through interactions with convecting meteoric water. Only at Kakkonda well WD-1a, where the boundary between the hydrothermal convective zone and thermal conductive zone (approx. 3.2 km depth; [79]) lies above the sampling depth, do fluid compositions reflect the deeper magmatic reservoir.

# 7. The importance of sulfur

In addition to metals, reduced sulfur ($S^{2-}$) is essential to formation of sulfide ore minerals, yet gauging the concentration and redox state of sulfur in magmatic fluids faces considerable analytical challenges. Consequently, the sulfur budget of ore-forming fluids remains a critical unknown in our understanding of PCD formation. A key question is whether ore-forming fluids carry sufficient metals *and* sulfur, or whether an external source of sulfur is required (e.g. [102,103]). For relatively oxidized magmas, such as those commonly associated with PCDs, the dominant sulfur species in the fluid is $SO_2$. Thus a further question concerns the mechanism and timing of reduction of $S^{4+}$ to $S^{2-}$. These questions have some bearing on the ability of magmatic brines to maintain high dissolved metal contents as they cool in the sub-volcanic environment.

Limited available data for intermediate density (ID) fluid inclusions from PCDs indicate sulfur contents in the range 200 ppm to 3 wt% [24] albeit with analytical uncertainties of at least ±20–50% relative (e.g. [65]). In almost all ID fluid inclusion analyses assembled by Kouzmanov and Pokrovski [24] sulfur is stoichiometrically in excess of that required to form chalcopyrite ($CuFeS_2$). Similarly, Seo *et al.* [65] used analyses of S in fluid inclusions from Bingham Canyon PCD, Utah, to argue that ample sulfur, of unknown redox state, is available in stoichiometric proportion to dissolved Cu + Fe in the same inclusions to form chalcopyrite, such that there is no need for any later addition of sulfur. If primary magmatic, saline fluids contain sufficient reduced sulfur ($S^{2-}$) to precipitate sulfides, then phase separation and ore mineralization may be near simultaneous processes, driven by a combination of fluid depressurization and cooling (e.g. [69]).

Conversely, using data from melt inclusions in volcanic phenocrysts, Blundy *et al.* [103] argue that the Cl- and Cu-rich fluids responsible for PCD formation are unlikely to contain sufficient sulfur because of

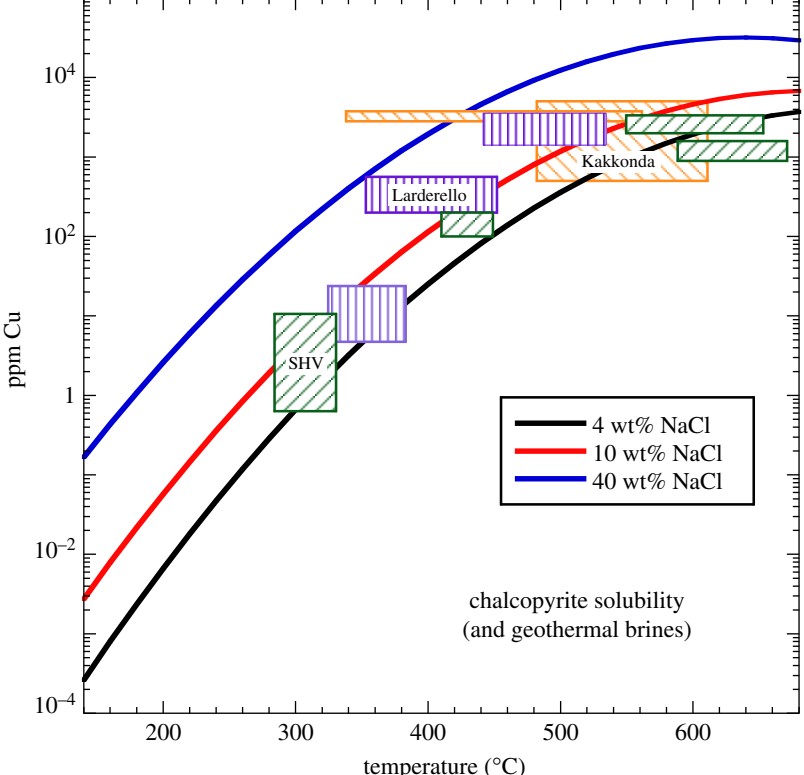

**Figure 5.** Solubility of chalcopyrite (as ppm Cu in solution) in hydrothermal solutions as a function of temperature based on equation (10.3) for three representative salinities (4, 10 and 40 wt% NaCl). The figure is based on an empirical fit to solubility curves presented in Fig. 17e of Kouzmanov & Pokrovski [24] for pyrite–magnetite–haematite-saturated fluids with pH = 5 at 0.3 to 1 kbar pressure. Also shown are ranges of homogenization temperatures and Cu contents for brine inclusions from geothermal wells from Montserrat (SHV), Larderello and Kakkonda (data from table 1).

the different degassing behaviour of Cl and S. In response to this sulfur deficiency, Blundy *et al.* [103], following Hattori & Keith [102], proposed that sulfur is delivered at a later, ore-forming stage, precipitating sulfides through reaction of $SO_2$ gas with metal-rich brines. Based on their observations of sulfide mineralization at Pinatubo volcano, The Philippines, and Bingham Canyon, Hattori & Keith [102] proposed that the source of this sulfur was mafic magma. Blundy *et al.* [103] presented experimental evidence to support such a proposal.

Although ore minerals are predominantly sulfides, it is not essential for all sulfur in the parent magmatic fluid to be in its reduced state. Disproportionation of $SO_2$ to form $S^{2-}$ and $S^{6+}$ is a potent means to create sulfides by reaction with a metal-bearing solution, especially in the presence of $Ca^{2+}$ ions when anhydrite ($CaSO_4$) is an additional product phase [42]. Such reactions are to be expected in transcrustal systems where $SO_2$-rich fluids from deeper in a system can interact with shallow-stored, Cl-rich fluids in the same system. Coexistence of anhydrite and metal sulfides is common both in PCDs (e.g. [6,42,104]) and in geothermal wells, including Kakkonda [105]. Supply of deep sulfur is also consistent with the 'excess sulfur' phenomenon at many volcanoes, whereby the supply of sulfur to the atmosphere exceeds that which could have been dissolved in the erupted magma [106]. If additional sulfur is required for ore-formation, then there may be a time lag between phase separation and sulfide precipitation (e.g. [103]).

Regardless of the details of the sulfur source, a main driver for sulfide mineral precipitation from fluids is cooling, as solubility of both chalcopyrite and bornite is strongly temperature dependent [24]. Sulfide solubility also depends on fluid salinity, pH, redox state and sulfur fugacity. For example, chalcopyrite solubility in a pyrite–magnetite–haematite-buffered, 40 wt% $NaCl_{eq}$ fluid with pH = 5 at 0.3–1.0 kbar falls from 2 wt% Cu at 550°C to 2 ppm at 200°C (figure 5; and [24]). This explains the very low Cu content of cool, neutralized, low-salinity geothermal fluids. Ultimately the solubility of chalcopyrite depends on a solubility product of $Cu^+$, $Fe^{3+}$ and $S^{2-}$ ions in solution. Thus, for different intensive parameters and/or buffer assemblages, the solubility of chalcopyrite, expressed in terms of ppm Cu in solution, will change. For example, at fixed salinity (10 wt% $NaCl_{eq}$), pH (5), redox (magnetite–haematite buffer) and temperature (400°C), but in the absence of pyrite, an increase in the

fluid concentration of $H_2S$ from 0.017 to 0.1 mol kg$^{-1}$ will reduce the Cu content of chalcopyrite-saturated liquids from 160 to 23 ppm ([24], fig. 14d).

In terms of our analyses of fluid inclusions from Montserrat, Larderello and Kakkonda (table 1), the solubility relationships in figure 5 may account for the overall trends in Cu content with temperature. However, measured Cu concentrations lie below those of the relevant 40 wt% NaCl$_{eq}$ solubility curve. One explanation is that the brines were not saturated with a pyrite–magnetite–haematite assemblage at the time of trapping, such that the extrapolated solubility curves are not appropriate for the pH-$f$O$_2$ conditions of the studied brines. Conversely, the chemical evolution of the brines may reflect successive episodes of phase separation during protracted volatile fluxing [70], with copper partitioning into brines at conditions below chalcopyrite saturation, potentially coupled with dilution by meteoric water. As the analysed brine inclusions do not necessarily represent a single suite of co-trapped fluids, it is not possible to discriminate between these alternatives without further chemical information, for example, from stable isotopes. Our observations emphasize the need for experimentally calibrated thermodynamic models for high-temperature chalcopyrite solubility in high-ionic-strength fluids. This is an area of active research.

# 8. Sub-volcanic brine lenses

The feeder zones (conduits) of volcanoes, through which hot, saline magmatic fluids pass, comprise hypabyssal intrusions (dykes and sills), magma-filled conduits (the likely origin of many stocks), pyroclastic material and fractured and brecciated rocks, typically with associated hydrothermal alteration and quartz veins (e.g. [107,108]). High porosities (e.g. greater than or equal to 20 vol% under Unzen volcano; [109]) are a characteristic of such systems, consistent with reduced P-wave velocities at depths less than or equal to 8 km beneath some active volcanoes (e.g. [110]). Average porosity of both dense and fragmental rocks decreases with increasing depth, but cores and well-logs show that active geothermal systems can maintain porosities (matrix plus fracture) of 10 to 30 vol% down to depths of 2.5 km (figure 6). Fragmental materials are typically the most porous; other lithologies, such as vapour-phase-altered pyroclastic materials, are resistant to densification by physical and chemical processes and may preserve significant primary porosity. Sub-volcanic conduits with associated damage zones, and fragmental infills related to prior intrusions, eruptions and steam explosions, have very high porosities (figure 6), that can provide efficient pathways for both fluid flow and accumulation. The presence of fractures may lead to a significant mismatch between relatively high porosity at a reservoir scale and relatively low porosity on a drill-core scale, as evidenced by data and modelling at Kakkonda (figure 6). Precipitation from (and dissolution by) ascending fluids will modify the original porosity structure in the feeder zone.

To investigate the potential of sub-volcanic conduits as a means of focusing and storing magmatic fluids, Afanasyev et al. [70] performed hydrodynamic models of discharge through a sub-volcanic domain of a 738°C supercritical aqueous fluid (4 wt% NaCl) from a magma body located at 7 km depth. Afanasyev et al. [70] considered explicitly the effect of a volcanic conduit region composed of high permeability, fractured rock. The porosity–depth distribution used by Afanasyev et al. [70] throughout the modelled domain is presented in figure 6 for comparison with the available natural data. For the reference model scenario (total fluid flux of $3 \times 10^9$ kg yr$^{-1}$ through a 0.33 km radius conduit of permeability $k \leq 10^{-14}$ m$^2$), Afanasyev et al. [70] presented calculations for degassing durations of up to 250 kyr. The permeability-depth distribution adopted by Afanasyev et al. [70] results in an effective transition from hydrostatic to lithostatic pressures at depths of 5–6 km and temperatures of 440–520°C, corresponding to the transition between hydrothermal convection and conductive regimes described above. A region of active hydrothermal convection develops in the upper 3 km of their modelled domain. Thus the model configuration approximates that of typical geothermal systems.

It is instructive to adapt the simulations of Afanasyev et al. [70] to White Island volcano (New Zealand), a plausible modern analogue to the scenario being explored. Hedenquist et al. [14] report a White Island flux of $3.5 \times 10^5$ kg day$^{-1}$ of SO$_2$ via a fumarolic fluid containing, on average, 3.5 wt% SO$_2$ at temperatures up to 800°C. The corresponding total fluid flux is $3.7 \times 10^9$ kg yr$^{-1}$, a value similar to that used by Afanasyev et al. [70]. The Cu content of fluids discharged from White Island, based on the reported Cu/SO$_2$ ratio, is 30 ppm. This is probably a minimum value for the original magmatic fluid, due to sub-surface deposition of Cu beneath the edifice [14] as well as phase separation of Cu-rich brines. The total duration of degassing from White Island is not known—for the purposes of the calculations reported here, we consider only the 10 kyr case. Note that this is a

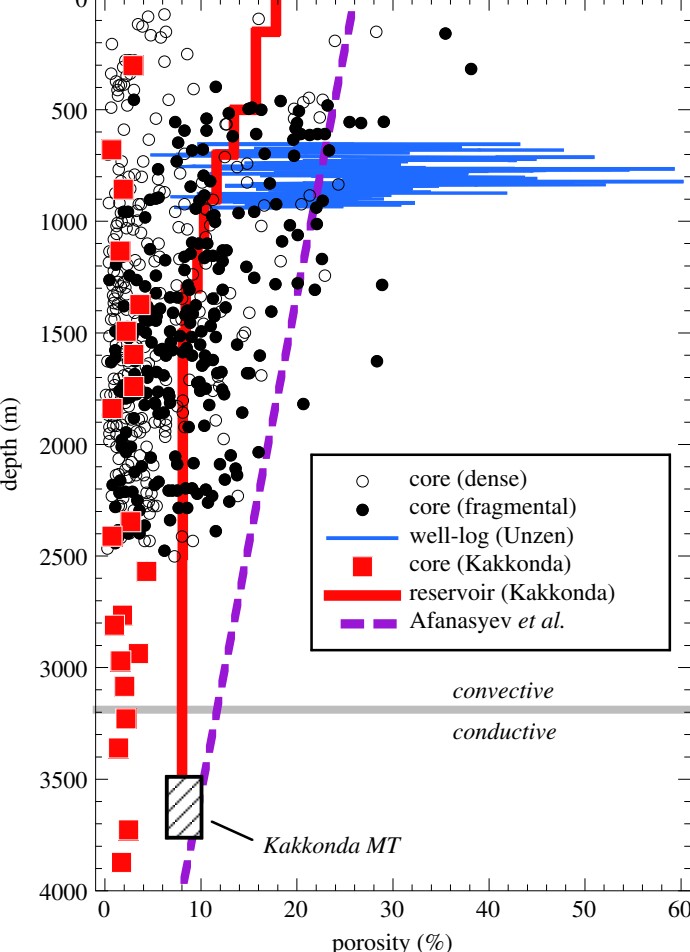

**Figure 6.** Porosity–depth relationships in volcanic-hosted geothermal reservoirs, and volcanic conduits. Geothermal well drill-core data (open and filled black symbols) are redrafted from Fig. 46.14 of Stimac *et al.* [71], and distinguish between dense (e.g. lavas, intrusives) and fragmental (breccias, tuffs, volcaniclastics) rock types. Blue line shows well-log data for the Unzen conduit taken from Fig. 2 of Ikeda *et al.* [109]. Red squares show porosities of individual drill-core rock samples from Kakkonda [111]; solid red line shows the depth-porosity relationship adopted in the Kakkonda reservoir model of McGuiness *et al.* [112]. In this model, porosity is set at a fixed value in a succession of horizontal layers, decreasing from 18.3% in the top layer to 8.5% in the bottom layer, based on the sonic and density log results of Kakkonda well WD-1 series and on an analysis of the electric logs of other Kakkonda wells reported by Sakagawa *et al.* [113]. The shaded box labelled *Kakkonda MT* is the range of calculated porosities for the approximately 3.7 km deep, 0.63 to 1.0 S m$^{-1}$ conductor at Kakkonda [80], as calculated in the text. The boundary between the hydrothermal convective zone and thermal conductive zone in WD-1a well is located at approximately 3.2 km [79] and shown with the grey line. Purple dashed line is the depth-porosity relationship adopted in the brine-lens model of Afanasyev *et al.* [70].

relatively short degassing period in the context of the long-term growth and differentiation of crustal magmatic systems. Despite the overall similarity of modelled conditions to those at White Island, we recognize that fluid fluxes and metal contents may vary significantly from one volcanic system to another. For that reason, our calculations are designed merely to be illustrative.

Afanasyev *et al.*'s [70] calculations predict accumulation of relatively dense (*ca* 1350 kg m$^{-3}$) hypersaline liquids at approximately 2 km depth in annular lenses surrounding the conduit (figure 7a), in close agreement with the calculations of Weis *et al.* [68] and Weis [69], despite some differences in modelling approaches. Both models concern flows of NaCl–H$_2$O fluid that account for phase transitions and multi-phase equilibria, and similar governing equations are used. The main difference concerns the modelling of permeability. Instead of the dynamic permeability model of Weis [69], we simulate the brittle–ductile transition with the static depth-dependent profile for permeability described above. Afanasyev *et al.* [70] show that phase transitions in NaCl–H$_2$O result in fluid focusing and brine lens formation even in a static permeability field.

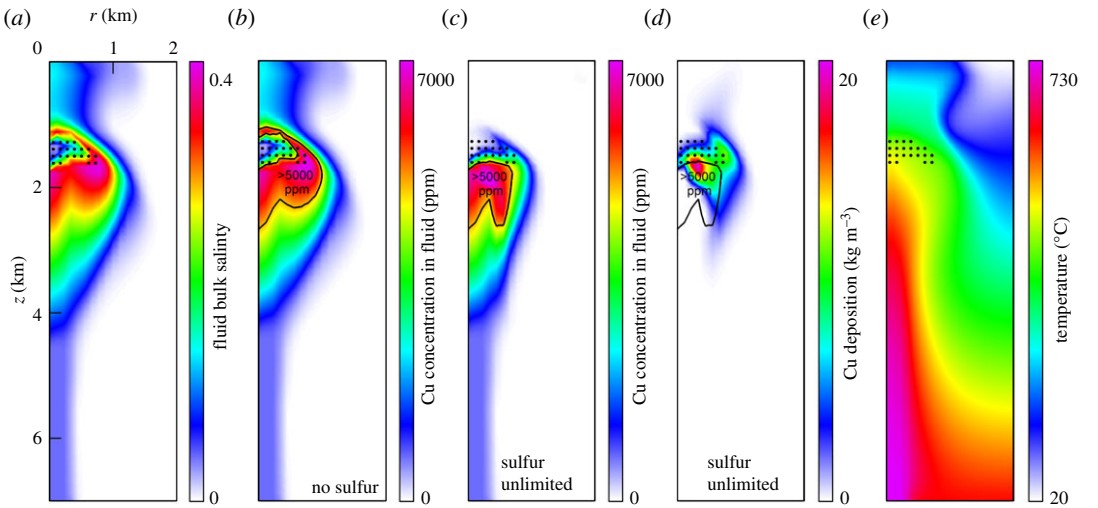

**Figure 7.** Modelled distribution of salinity, Cu and temperature in the sub-volcanic region for the reference scenario of Afanaysev *et al.* [70] after 10 000 years of degassing in terms of (from left to right): (*a*) bulk fluid salinity expressed as molar fraction NaCl$_{eq}$; (*b*) concentration of dissolved Cu (as ppm Cu) for the no-sulfide case; (*c*) concentration of dissolved Cu (as ppm Cu) for the unlimited-sulfide case; (*d*) precipitated Cu for the unlimited-sulfide case, expressed as kg Cu per m$^3$ of rock; and (*e*) temperature (°C). The conduit radius is 0.33 km, centred on the vertical axis; input flux of fluid (738°C, 4 wt% NaCl$_{eq}$, 700 ppm Cu) is $3 \times 10^9$ kg yr$^{-1}$ originating from 7 km depth. Black dots in all panels indicate halite precipitation. The volume containing hypersaline liquids with greater than 5000 ppm Cu is outlined in black in (*b*)–(*d*). Panels have identical vertical and horizontal scales, and are axisymmetric about the left-hand axis.

A direct comparison of our models and those of Weis [69] is provided in electronic supplementary material, figure S2. The depth and shape of modelled brine lenses echo the original proposal of Fournier [67,72] that brine accumulates below the brittle–ductile transition, which serves as a permeability barrier. However, the numerical models predict further that the hypersaline liquid lens is capped by halite precipitation that occludes pore space and restricts further upward movement. Precipitation of other solutes, such as SiO$_2$, may also help to confine the brine lens, although this has not yet been explored in detail in our models. Halite precipitation has been described from ore deposits [56,114,115] and geothermal reservoirs [116,117] and is consistent with the evidence of halite saturation in some of the brine fluid inclusions in table 1.

In a parametric analysis, Afanasyev *et al.* [70] showed that the hypersaline liquid lens is not well developed for lower temperature fluid inputs (less than 700°C) or for very wide ($r > 1$ km) conduits, but is enhanced for narrow conduits, and hotter, more saline input fluids. Modelled lenses persist for up to 1 Myr after degassing ends, eventually sinking, spreading out, and being diluted and dissipated by any meteoric water convection at the margins [70]. This situation resembles that in many geothermal reservoirs, including Kakkonda [79,112], where the meteoric and magmatic fluid systems are efficiently isolated from one another despite active hydrothermal convection.

# 9. Geophysical observations

Geophysics affords considerable insight into the architecture and dynamics of crustal magma systems [118]. Techniques include potential fields (electrical conductivity, gravity, magnetics), seismic wave speeds and tomography, and geodesy (ground- and satellite-based). Increasingly these techniques are being applied to individual volcanic centres to elucidate magma plumbing. Although the images differ from volcano to volcano, a unifying theme of most geophysical surveys of volcanoes is the presence of relatively large volumes of reduced seismic wave speed and enhanced electrical conductivity at depths greater than or equal to 2 km consistent with large distributed volumes of partial melt and/or magmatic fluids, rather than a single large-liquid-filled magma vat. Real-time geodetic surveys indicate that such systems are dynamic on timescales of years to decades [119,120], probably due to the relative mobility of fluids (e.g. [121]). Such images are consistent with the upper reaches of transcrustal magmatic systems (figure 1). However, the non-uniqueness of geophysical techniques, especially when applied in isolation, makes it

difficult to resolve relatively small system features, and to establish the abundance, composition and distribution of the fluid phase (melt, hypersaline liquid, gas).

From the perspective of sub-volcanic brine lenses, electrical conductivity has the greatest potential as an imaging and mapping tool. Electromagnetic surveys, including magnetotelluric (MT) and time-domain (TDEM), have been conducted at numerous active and dormant volcanoes, including Mount Fuji, Japan [122]; Taal, The Philippines [123]; Kusatsu-Shirane, Japan [124]; Uturuncu, Bolivia [125]; Unzen, Japan [126–128]; Taupo Volcanic Zone, New Zealand [129–131]; and Mount St Helens, USA [132,133]. These surveys consistently reveal prominent, electrically conductive (greater than or equal to $1 \, \text{S m}^{-1}$) regions at depths greater than or equal to 2 km, interpreted by Afanasyev *et al.* [70] to represent hypersaline liquid accumulations. Although this is the prevailing interpretation of the authors of these many studies, other interpretations are possible and have been advanced. For example, conductive lenses may be smectite clays, although these are stable only at less than approximately 250°C [71], temperatures well below those expected at such depths in the active volcanic systems imaged. Moreover, clay layers, where present, e.g. Rotokawa [129], Kusatsu-Shirane [124] or Montserrat [92], tend to be laterally persistent and lie at depths less than 1 km, with forms resembling the clay caps of geothermal systems. On the other hand, the conductive lenses might be magma, although the lower electrical conductivity of silicate melt compared with hypersaline liquid would require high melt fractions (greater than 50%) at relatively shallow depths to attain the same bulk conductivity. Such high melt fraction is commonly inconsistent with other geological evidence, such as shear wave speeds (e.g. [125,128]). Finally, the conductor may be composed, in part, of electrically conductive minerals such as sulfides or magnetite, which can have conductivities as high or higher than brines (e.g. [134]). However, conductive minerals would need to be fully connected through the rock volume in order to generate the observed anomalies; the requisite wetting behaviour is more consistent with a fluid conductor phase.

Notably, the calculated depth, geometry and electrical conductivity of the modelled and imaged hypersaline liquid lenses are very similar [70]. In a number of MT studies, the conductor at 2 km depth extends to the middle or lower crust (e.g. [125,129–133]) suggesting connectivity to a deeper magma source. In those cases, the upper regions of the conductor would represent fluids exsolved from ascending hydrous magmas (± precipitated sulfides), creating a vertical, conductive continuum from regions of hydrous partial melt to accumulated fluids, as shown schematically in figure 1. MT surveys alone would not be able to resolve this transition.

Despite the potential ambiguities of MT images, the example of Kakkonda provides some important confirmatory evidence for the existence of brine lenses, as, uniquely, Kakkonda is a location where MT images are supported by direct fluid sampling. Well WD-1a penetrated the periphery of the main conductive anomaly as imaged by Uchida *et al.* [135]. Conductivity at WD-1a well bottom, where temperature is greater than equal or to 500°C, is in the range $0.10–0.16 \, \text{S m}^{-1}$ (MT profiles Line-C and Line-1 of [135]). The core of the conductor body lies approximately 1.5 km to the southeast of WD-1a at approximately the same depth (2–4 km) and temperature, but with a conductivity of 0.63 to less than $1 \, \text{S m}^{-1}$. The measured porosity of the brine-hosting core from 3727 m depth is 2.4% [111].

Afanasyev *et al.* [70] adopt a formulation of Archie's Law for calculation of the electrical conductivity of a porous medium containing intergranular saline fluid. For the case where the pore space is filled exclusively by fluid (i.e. without halite, for which there is no evidence at Kakkonda) the conductivity of the rock is given by

$$\sigma_r = \frac{\sigma_l \phi^m}{\alpha}, \tag{9.1}$$

where $\sigma_r$ is the electrical conductivity of the fluid-saturated rock, $\sigma_l$ is the electrical conductivity of the fluid, calculated at the pressure–temperature–salinity of interest using the model of Sinmyo & Keppler [136], $\phi$ is the porosity, $m$ is the cementation exponent and $\alpha$ is the tortuosity factor. Values of $m = 1.9$ and $\alpha = 0.6$ were adopted by Afanasyev *et al.* [70] and are retained here for consistency.

The calculated conductivity of 39 and 55 wt% $NaCl_{eq}$ brine at 1 kbar and 500°C is 53 and $69 \, \text{S m}^{-1}$, respectively. For a porous medium ($\phi = 0.024$; [111]) saturated with these two brines the calculated rock conductivity, from equation (9.1), is 0.08 and $0.10 \, \text{S m}^{-1}$, respectively, in excellent agreement with the MT data for WD-1a well bottom. It is possible to calculate the porosity in the core of the conductor, assuming that the brine has the same salinity range (39–55 wt% $NaCl_{eq}$), pressure (depth) and temperature. To obtain the MT value of $0.63 \, \text{S m}^{-1}$ requires porosities in the range 6.4 to 7.5 vol%. To attain a conductivity of $1 \, \text{S m}^{-1}$, a value at the upper bound of what is imaged at Kakkonda [135], but in keeping with conductivity anomalies beneath many volcanoes, the required porosity is 8.2 to

9.5 vol%. The calculated range of porosity values is consistent with those used in the model of Afanasyev et al. [70] at these depths (10 vol% at 3.5 km), and with the McGuinness et al. [112] reservoir model for Kakkonda (figure 6). We conclude that in the one location where hypersaline fluid accumulation has been proven by drilling, the electrical conductivity data can be adequately explained by brine at porosities of several per cent. For reference, the measured permeability of core from WD-1a well bottom (3727 m depth) is $10^{-16}$ m$^2$ [111], a value intermediate between Afanasyev et al.'s [70] model conduit ($10^{-15}$ m$^2$) and matrix ($10^{-18}$ m$^2$) permeabilities at this depth.

# 10. Copper accumulation in sub-volcanic brine lenses

To assess the potential for metal accumulation in sub-volcanic brine lenses we have incorporated copper transport into the hydrodynamic simulations of Afanasyev et al. [70] by including partitioning of Cu between vapour and liquid of differing salinity, and the solubility of chalcopyrite. In the light of the importance of S$^{2-}$ ions for copper ore mineral precipitation our new model considers the fate of both sulfide-sufficient and sulfide-free, Cu-bearing magmatic fluids as they discharge through a high-permeability and porosity conduit, undergo phase separation and cool. We note that Weis et al. [68] also incorporated Cu solubility and partitioning into their models, concluding that Cu accumulates below a halite cap, within a region of elevated salinity, similar to the hypersaline liquid lens in figure 7a.

## 10.1. Methods

We augment the hydrodynamic model equations of Afanasyev et al. [70] with the following conservation equation for modelling Cu transport

$$\frac{\partial}{\partial t}\left(\phi\sum_{i=g,l}\rho_i c_i s_i + (\phi_0 - \phi)\rho_{Cu}\right) + \mathrm{div}\left(\sum_{i=g,l}\rho_i c_i w_i\right) = 0, \tag{10.1}$$

where $\phi$ is the porosity, $\phi_0$ is the initial porosity prior to Cu precipitation, $\rho$ is the density, $c$ is the Cu mass fraction, $s$ is the saturation, $w$ is Darcy's velocity, and subscripts $g$, $l$ and $Cu$ refer to parameters of the gas, liquid and chalcopyrite (solid) phases, respectively. In using equation (10.1), we assume that, providing reduced sulfur is available, Cu forms a separate solid phase (e.g. chalcopyrite) that precipitates independently of halite. The copper sulfide mineral density $\rho_{Cu}$ is set to 4100 kg m$^{-3}$.

The total concentration of Cu, $c_t$, in gas and liquid phases is given by

$$c_t = \sum_{i=g,l}\rho_i c_i s_i \bigg/ \sum_{i=g,l}\rho_i s_i. \tag{10.2}$$

Copper partitioning between vapour and liquid phases is calculated in line with the experimental data in Fig. 15 of Kouzmanov & Pokrovski [24]

$$\log\frac{m_g}{m_l} = 3.866 \log\frac{\rho_g}{\rho_l}, \tag{10.3}$$

where $m$ is the number of Cu moles per 1 kg of fluid in the corresponding phase. This expression is similar to that of Tattitch and Blundy [25] wherein copper partitioning scales with the salinity ratio of liquid and vapour.

To constrain reasonable upper bounds on brine Cu contents requires an expression for the equilibrium solubility of Cu, $c_{eq}$, at brine lens conditions. As noted above, $c_{eq}$ depends on a number of factors, including intensive parameters and the mineral assemblage that buffers $f$H$_2$S, and $f$O$_2$. In the absence of a robust thermodynamic model at elevated temperatures, we have chosen to simply interpolate the pyrite–magnetite–haematite-saturated solubility data in Fig. 17e of Kouzmanov & Pokrovski [24] to yield an empirical relationship for $c_{eq}(T,x)$ as a function of temperature and fluid salinity $x$:

$$\log(c_{eq}) = a_{00} + a_{10}\ln(x) + a_{01}T + a_{20}[\ln(x)]^2 + a_{11}\ln(x)T + a_{02}T^2, \tag{10.4}$$

where $T$ is in °C, $x$ is weight fraction NaCl$_{eq}$ in the liquid, $a_{00} = -2.616$, $a_{10} = 1.741$, $a_{01} = 0.02567$, $a_{20} = 0.07463$, $a_{11} = -2.117 \times 10^{-5}$, $a_{02} = -0.00153$. The resulting expression, for three representative salinities, is shown in figure 5. Copper is completely dissolved in hypersaline liquid if $c_t < c_{eq}(T,x)$ and chalcopyrite precipitates when $c_t \geq c_{eq}(T,x)$. Further experiments are required to refine these solubility

relationships at high temperature (greater than 400°C) and salinity; equation (10.4) is simply a useful starting point for a plausible set of brine lens conditions.

Calculations assume that pH of the fluid is 5, consistent with rock-buffering via hydrolysis reactions involving K-feldspar and muscovite. For a one molal NaCl solution (5.8 wt% $NaCl_{eq}$) at 225°C these reactions define pH = 5 ± 0.5 [137]. Hypersaline liquids recovered at depth at 500°C from Kakkonda, have pH in the range 3–5 [77,78]. If pH deviates from 5 then chalcopyrite solubility will be modified. For a unit decrease in pH below 5, all other parameters constant, solubility increases by an order of magnitude; for a unit increase in pH solubility decreases by the same factor [24]. Modelled pH evolution of the fluids in response to rock reaction would be a future development, as would modifications to the buffer assemblage. However, comparison of the calculated solubility curves with our analyses of fluid inclusions from geothermal drill-core (figure 5 and table 1) suggests that equation (10.4) provides a reasonable approximation to natural systems that contain sufficient $S^{2-}$ to precipitate chalcopyrite.

Numerical simulation of Cu transport is done using the method of splitting by physical processes. Each time step consists of two separate steps A and B. In Step A transport of the primary NaCl–$H_2O$ fluid is simulated for a given porosity $\phi$ and permeability, and new distributions of $P$, $T$, $\rho_i$, $s_i$, $w_i$ are evaluated (see [70]). In Step B, these distributions are frozen and Cu transport is calculated according to equations (10.1)–(10.4). At the end of Step B if $c_t$ in a grid block is higher than the equilibrium value (5) then the fraction $c_t$-$c_{eq}$ of Cu is precipitated and thereby $c_t$ is reduced to $c_{eq}$. Based on the amount of precipitated Cu the porosity $\phi$ and the permeability distributions are modified. The calculation then continues with Step A at the next time step.

Calculations were run with the same initial conditions as the reference model of Afanasyev *et al.* [70] in terms of fluid flux, permeability, porosity etc. (figure 7a). The input fluid contains 700 ppm Cu, a conservative value based on ID fluid inclusions from Bingham Canyon [63,65] and comparable to the 500 ppm Cu used by Weis *et al.* [68]. As noted above, this value exceeds that in the White Island fumarole gas (30 ppm), which has been lowered by sub-surface reactions and Cu mineral precipitation. Sulfide content of the input fluid is deemed either: (i) sufficient to precipitate chalcopyrite when its solubility is attained according to equation (10.4); or (ii) zero, in which case no Cu precipitation can occur. Our results, therefore, represent end-member cases where Cu precipitation is (i) maximized or (ii) minimized at the expense of Cu retained in solution. We do not consider explicitly sulfide formation via the Ca-mediated disproportionation of $SO_2$ proposed by Mavrogenes & Blundy [42], nor variations in mineral buffer assemblage.

## 10.2. Results

Model results for the reference scenario are presented in terms of bulk fluid salinity (figure 7a), Cu dissolved in hypersaline liquid for the no-sulfide (figure 7b) and unlimited-sulfide (figure 7c) cases, Cu contained in precipitated solids (unlimited-sulfide case only, figure 7d), and the temperature field (figure 7e). In panel (b–e), the region containing hypersaline liquids with greater than or equal to 5000 ppm Cu is outlined. Results are shown at 10 000 years after the onset of fluid discharge (figure 7a). For both precipitated and dissolved Cu, concentrations are highest in an annulus centred on the conduit at 1.5 to 2 km depth. The shape and depth of the precipitated Cu annulus is similar to that of Weis *et al.* [68], despite their use of a different chalcopyrite solubility expression and modelling approach (electronic supplementary material, figure S2)

The precipitated Cu annulus (figure 7d) is more restricted in extent than the Cu-rich liquid annulus (figure 7b,c), reflecting higher temperatures near the conduit. The highest bulk contents of precipitated Cu, approximately 20 kg m$^{-3}$ of rock, are contained within an inclined region located at 1.5 km depth 700 m from the centre of the conduit (figure 7d). Very little Cu is precipitated within the conduit itself. These bulk Cu contents equate to maximum potential ore grades of 0.75 wt% Cu for rock with 2650 kg m$^{-3}$ density. This grade is comparable to many economic PCD mines. Precipitated Cu contents decrease rapidly away from this region reflecting temperature decrease. The mass of solid Cu (as chalcopyrite) increases almost linearly with degassing duration, as the liquid lens becomes saturated and more and more chalcopyrite precipitates. Thus ore grades of approximately 1.5 wt% Cu would require degassing for greater than or equal to 20 kyr at the reference flux.

The Cu-in-liquid annulus (figure 7b,c) is slightly broader and deeper than for precipitated Cu, extending from the axis of symmetry out to 1 km radial distance. The no-sulfide annulus (figure 7b) is slightly broader than the unlimited-sulfide annulus (figure 7c). The highest hypersaline liquid Cu contents (greater than or equal to 7000 ppm) occur at the outermost periphery of the annulus where

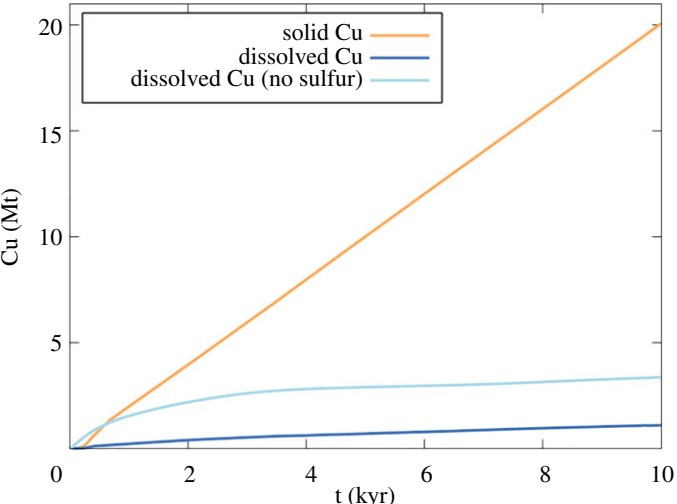

**Figure 8.** Time evolution (up to 10 000 years) of total mass of dissolved (orange line) and precipitated Cu (dark blue line) for unlimited-sulfide case, and dissolved Cu (pale blue line) for the no-sulfide case. Models are for the reference scenario in figure 7. The total Cu input to the modelled domain is 21 Mt. Copper masses are integrated across the reference volume defined in the text; there is very little stored Cu in the modelled domain beyond this volume. Note the tendency towards steady-state mass of dissolved Cu after 10 000 years.

temperature exceeds 400°C (figure 7e) for both the no-sulfide and unlimited-sulfide cases. These modelled Cu contents are consistent with high-salinity fluid inclusion analyses from PCDs (e.g. [24,63,65]). Note that, for the unlimited-sulfide case, contours of Cu-in-liquid concentration do not map simply onto isotherms, because salinity (in addition to temperature) also controls Cu solubility (figure 5). In the no-sulfide case, the highest Cu contents in the hypersaline liquid lens are controlled not by chalcopyrite solubility, but by the dynamics of phase separation and partitioning. They are therefore more sensitive to the model parameters, such as input fluid Cu content, fluid flux and porosity–permeability characteristics of the conduit region than for the unlimited-sulfide case.

Total Cu input to the system in 10 000 years is 21 Mt for the modelled fluid flux and Cu content. For the unlimited-sulfide case, the total mass of Cu retained in the modelled domain includes both precipitated and dissolved forms. For a reference volume with radius 1.5 km extending from the surface to 3 km depth the mass of precipitated Cu is 20 Mt after 10 000 years (figure 8). The mass of Cu dissolved in hypersaline liquid at this time (approx. 1 Mt) is less by a factor of 20. Thus, almost no copper is lost to the atmosphere via the low-salinity exhaust vapour, and very little is stored at low concentration outside the reference volume. For the no-sulfide case, there is no precipitated Cu, but the total mass dissolved (approx. 3.5 Mt after 10 000 years; figure 8) is less than the total Cu retained in the unlimited-sulfide case, due to significant Cu loss to the atmosphere, at a time-averaged flux of 4800 kg day$^{-1}$. For comparison, copper fluxes to the atmosphere of 200 to 20 000 kg day$^{-1}$ have been recorded at several subduction-related basalt volcanoes [4]; at White Island andesite volcano the copper flux is approximately 300 kg day$^{-1}$ [14]. The range in Cu flux between volcanoes probably reflects variation in the initial fluid content of Cu and availability of reduced sulfur, as well as the efficiency of Cu sequestering by phase separation, sulfide precipitation, and mixing with meteoric water beneath the volcanic edifice. Our no-sulfide and unlimited-sulfide cases probably provide bounds on the situation beneath active volcanoes.

The annular form of the modelled high-Cu regions, both precipitated and dissolved (figure 7c–e), is reminiscent of the three-dimensional form and dimension of ore shells in several PCDs, e.g. Bajo de la Alumbrera, Argentina [47], Bingham Canyon, USA [138], Batu Hijau, Indonesia [139]. This is consistent with the notion that hypersaline liquid accumulation is related to ore formation, as previously suggested by, *inter alia*, Fournier [72], Weis *et al.* [68], Weis [69] and Blundy *et al.* [103]. The size and copper endowment of the modelled high-Cu regions for the unlimited-sulfide case (approx. 20 Mt) are broadly similar to these world-class PCDs, whereas the endowment for the no-sulfide case is substantially smaller. This comparison emphasizes the importance of the availability of sulfide (as well as Cu) in generating PCDs.

As discussed above, a central question in PCD research is the extent to which a single magmatic fluid can provide both sulfide and copper, and, if so, under what circumstances. Needless to say, a sulfide-poor copper-

rich brine lens will not become a large PCD without the subsequent addition of sulfur, as noted by Blundy *et al.* [103]. It is well established in volcano studies that the mass of sulfur discharged from active volcanoes frequently exceeds that dissolved in erupted magma [106]. One explanation for this 'excess sulfur' problem is that significant volumes of unerupted magma contribute to sulfur discharge through the volcanic edifice. Degassing from a transcrustal magma system, as shown in figure 1, provides a ready means to supply additional sulfur. The requirement for substantial quantities of sulfur to form sulfide-rich ores may explain the relative scarcity of world-class PCDs compared with the apparent ubiquity of electrically conductive regions beneath many volcanoes. If such lenses do represent accumulations of metal-rich hypersaline liquids, absent an adequate supply of sulfide they may eventually become diluted and dissipated through groundwater interaction, with low preservation potential.

In our simulations, the conduit region remains at high temperature (figure 7*e*) because degassing has been ongoing for 10 000 years. When degassing stops, the system cools; in the unlimited-sulfide case further Cu-sulfide precipitates. Thus, over time the Cu-sulfide ore body grows at the expense of the Cu-rich hypersaline liquid lens. Sulfide precipitation can occur in the no-sulfide case only if the Cu-bearing brine lens interacts with subsequent pulses of sulfur-rich fluid from deeper in the system. Where this fluid contains predominantly $SO_2$, sulfide precipitation may occur by the Ca-mediated disproportionation reaction described above. In this case, brine accumulation and lens growth is decoupled in time from sulfide ore formation, as suggested by Blundy *et al.* [103].

Working against brine lens growth are dilution and removal of liquid by convecting groundwater, as suggested by modelling [70,140]. Nonetheless, hypersaline liquid lenses appear to be a persistent feature of volcanoes even several hundred thousand years after the end of degassing (e.g. Uturuncu, [125]). Development of a quasi-steady state system after more than 10 000 years (e.g. Figure 8) relates to a balance between the influx of new magmatic fluid, entrainment of hypersaline liquids into the associated hydrothermal system, and discharge of Cu to the atmosphere. It is likely that this balance will vary from volcano to volcano and depend to large extent on the physical separation of the magmatic liquid reservoir and the convecting hydrothermal system.

# 11. Implications for *in situ* mining

Copper mining exploits ore deposits that have long since cooled and exhumed to the (near) surface. Consequently, the bulk of the Cu in a hypogene PCD is in the form of solid sulfides with vestiges of Cu-bearing hypersaline liquids confined to tiny fluid inclusions in the host rocks. Because of relatively low ore grade (less than 1 wt% Cu) mining entails large-volume extraction of Cu-poor rock in open pits or underground mines. Extracted sulfide ore must then be crushed, concentrated and smelted, prior to dissolution and electrowinning to generate high-purity copper cathode. Thus, copper production from hypogene ore is costly, in terms of energy [141], and environmentally impactful, in terms of excavation, hazardous reagents, disposal of mine tailings, generation of rock dust and smelter emissions [142].

The possibility that sub-volcanic hypersaline liquid lenses contain substantial dissolved Cu raises an alternative, *in situ* approach to mining, namely direct extraction to the surface of hot, Cu-bearing liquids via boreholes that penetrate the brine lens. Our reference model results indicate that the highest dissolved Cu concentrations are found in rocks at a radial distance of approximately 1 km from the volcanic conduit and depths of approximately 2 km (figure 7*b,c*). Dissolved Cu concentrations are highest when the transporting fluid is poor in reduced sulfur. Temperatures in this region are very high (greater than or equal to 400°C, figure 7*e*), but within the range of the latest generation of geothermal targets. For example, the Iceland Deep Drilling Project geothermal well IDDP-2 at Reykjanes penetrated 426°C volcanic rock at depths of 4.5 km [143], the Venelle-2 well at Lardarello reached 504°C at 2.8 km [144], and the Kakkonda WD-1a well reached 500°C at 3.7 km [79]. There is emerging interest in drilling even deeper, hotter systems to extract power from supercritical fluids associated with young magmatic systems [73,74,145–147]. For example, the Japan Beyond Brittle Project at Naruko volcano will target 350–500°C fluids stored in ductile reservoir rocks at 3–5 km depth [148,149]. A 1 S m$^{-1}$ conductivity, vertically elongate body (C1) beneath Naruko extends from the lower crust to approximately 4 km depth [150]. This structure is consistent with a body of melt at depth that transitions to saline fluids at shallower levels.

The potential of hot hypersaline liquids as a source of metals (e.g. Mn, Zn, Pb) and silica has long been recognized [151–157], but their attempted exploitation has a chequered history (e.g. [158–162]).

Demand for lithium has led to a recent resurgence of interest in metal recovery from high-salinity geothermal fluids [163].

Provided that deep wells into hot rock can be completed successfully, the difficulty of recovering Cu-bearing fluids faces two key problems. The first is the relatively low contents of Cu (and other metals) in conventional geothermal fluids. Indeed, Neupane & Wendt [164], in their assessment of US geothermal mineral resources, state 'the presence of low mineral content in the brine could be prohibitive for extraction'. The available data for geothermal fluids (e.g. [165]) support this statement. However, almost all geothermal fluids analysed so far are mixtures of magmatic and meteoric fluids with quite low salinities (average approx. 1 wt% $NaCl_{eq}$; [71]) recovered from actively convecting hydrothermal systems at temperatures below 360°C, where solubility of copper minerals is low (figure 5). The lack of suitably hot, solute-rich, hypersaline magmatic liquids, with the exception of Kakkonda, is largely a consequence of intentionally avoiding such regions when drilling for geothermal resources. Recovering metal-rich hypersaline liquids requires tapping brine lenses at significantly higher temperatures than is conventional in geothermal fields, and at depths below the brittle–ductile transition, remote from dilution by convecting meteoric water.

A second challenge is scaling and corrosion of the well-bore in response to decompression and cooling of solute-rich, acidic fluids (e.g. [166]). It is for this reason that most geothermal production wells avoid solute-rich liquid lenses in favour of low-salinity, high-enthalpy vapour [74]. Scales recovered from a number of geothermal wells are extremely Cu-rich, due to precipitation of sulfides (and tellurides). For example, at Kakkonda, variable temperature histories in both the hypersaline liquid reservoir and during the sampling process itself resulted in Cu-Zn-Pb-Fe rich sulfide/oxide scales with a wide range of chalcophile metal ratios [80] and up to 14 wt% Cu [86]. Despite this variability, the record of deposition of metals in the scales, combined with a transition from Pb-rich to Zn-rich to Cu-rich scales with increasing temperature, is consistent with a high-temperature (greater than 500°C) hypersaline reservoir at Kakkonda that, at depth, may contain dissolved metals at significant concentrations, as suggested by the fluid inclusion estimates in table 1.

The propensity for scaling will be exacerbated when extracted fluids are even hotter and more solute-rich than in conventional geothermal wells. There have been a number of attempts to develop fluid extraction methods to minimize scaling by modifying the chemistry and temperature of the extracted fluid (e.g. [167–169]). Reducing scaling of silica, an important additional component of geothermal fluids, has been a particular priority (e.g. [170–172]). Further development of such methods will lead to more efficient recovery of metals from deep, hypersaline fluid reservoirs.

Finally, drilling into hot rock and extracting metal-bearing fluids represents a major technical challenge because of uncertainties in the permeability and porosity structure of the reservoir, and the highly corrosive nature of the fluids themselves. This is largely because there has been very little drilling into deep portions of geothermal systems with accumulations of high-temperature magmatic fluids lying below the brittle–ductile transition. Drill-core samples recovered from such wells will tend to underestimate the *in situ* porosity and permeability, as cooling and reduction in fluid pressure cause mineral precipitation and closing of fractures around the well-bore. Also, where hypersaline magmatic fluids accumulate there may be dynamic permeability variations with spatial and temporal changes in brittle versus ductile host rock behaviour (e.g. [69]). Data on matrix porosity and permeability from traditional geothermal reservoirs are often proprietary or not well documented, and there are few published attempts to reconcile electrical conductivity and reservoir porosity. Our attempts to do this at Kakkonda, as presented above, do support the existence of hypersaline fluids in reservoir rocks with less than or equal to 10 vol% porosity, and permeability of the order $10^{-16} m^2$.

It is important to understand sub-volcanic porosity–permeability relationships because they control how much metal-bearing hypersaline fluids can be stored, and the extent to which they will flow into the borehole with or without additional reservoir stimulation (e.g. [149,173]). Porosity–permeability relationships also influence the response of the reservoir to fluid extraction, notably variations in bottom-hole pressures. Loss of control of pressure in the well-bore can lead, for example, to blowouts of the type that occurred at Wairakei Geothermal Field, New Zealand, in the 1960s [174]; a result of rapid, uncontrolled decompression of hot geothermal fluids. Although there are fundamental differences between the 1960s drilling operation at Wairakei and drilling to extract metal-bearing hypersaline fluids, it is clear that careful monitoring and control of well-bore and reservoir pressures during drilling, use of dense drilling muds, and careful design of well-bore cements and casings are fundamental to the safe operation of *in situ* mining. Corrosion-resistant well-bore casing (or coating) materials are particularly important, and an area of active research (e.g. [175–177]). These technical challenges require further investigation by drilling engineers and materials scientists, which is beyond the scope of this article.

## 12. Prospectives

The economic potential of metal-bearing sub-volcanic hypersaline fluids hinges on a number of unknown parameters that require further evaluation. These include the salinity and composition of putative magmatic brines (including base metals and deleterious elements, such as arsenic and mercury), their abundance and distribution beneath volcanoes, and their extractability. At present, we can only speculate, based on hydrodynamic models and limited geological and geophysical data, on what such lenses would look like and where they would lie. Uncertainty extends to a variety of technical challenges associated with extraction and recovery. In these regards detailed reservoir characterization and careful selection of drilling sites, alongside technical developments, are key to future development of the concept. The limited data from a few deep magmatic geothermal exploration sites shows how underrepresented hypersaline magmatic brines are in the general record of geothermal fluids, largely as a result of the preferential targeting of fluids from the convective hydrothermal regime for conventional geothermal power production.

Reservoir characterization requires integration of a variety of geophysical techniques including MT (to identify conductive bodies), gravity (to constrain the distribution of low-density fluids versus dense sulfides), geodesy (to monitor ground inflation and deflation in real time) and seismology (to characterize porosity–depth relationships). Integration of different types of geophysical data is necessary to resolve some of the potential ambiguities that come with a single method. Few active volcanoes are sufficiently characterized using these different techniques, although there is growing interest in applying joint geophysical inversions to address problems of sub-volcanic magma storage at a number of active volcanoes: Laguna del Maule, Chile [178]; Mount St Helens, USA [179,180]; Uturuncu, Bolivia [181]; Santorini, Greece [182]. There is also an argument for scientific drilling directly into sub-volcanic regions to better understand their magmatic architecture [183,184]. There is considerable commonality in the objectives of these drilling proposals, e.g. melt distribution and connectivity, fluid/melt chemistry, temperature and dynamics, and the key unknown parameters around the proposed exploitation of sub-volcanic hypersaline fluid lenses. Direct drilling of potential target volcanoes is undoubtedly the best method to reduce some of the uncertainties inherent in the *in situ* extraction of metal-rich brines, not least to establish their presence in the volumes predicted by our models. Drilling would also allow for recovery and chemical characterization of any such fluids.

## 13. Conclusion

Sub-volcanic regions can trap hypersaline liquids that form on phase separation of ascending supercritical magmatic fluids. There is permissive evidence for the existence of brine lenses from geophysical studies of the roots of active or dormant volcanoes, for example from electrical conductivity surveys. However, these geophysical signals have ambiguities, and further work is required to resolve the contributions to elevated electrical conductivity due to brines, sulfide minerals and silicate melts, and combinations thereof.

Hydrodynamic calculations show that Cu concentrations in dynamic, hypersaline liquid lenses beneath active or dormant volcanoes may be as high as 7000 ppm at temperatures over 400°C. For our reference model, after 10 000 years of degassing the hypersaline liquid lens contains 1.0 to 1.4 Mt of dissolved Cu, depending on the availability of reduced sulfur. These expectations are consistent with limited measurements of high Cu content in fluid inclusions from geothermal drill-core (table 1), together with analyses of recovered hypersaline fluids at Kakkonda. In order to become potential economic metal resources, hot, metal-rich brines would need to be extracted efficiently without significant reservoir clogging or well-bore scaling. The extent to which such a resource could be exploited depends on the technological challenges and costs of drilling into hot reservoir rocks and recovering hypersaline liquid. If Cu can be extracted in solution form, Cu processing costs may be greatly reduced. Similarly, the co-production of geothermal energy to power fluid extraction and processing would confer additional economic and environmental benefit (e.g. [162]). We have focused only on Cu; the presence of other metals in magmatic fluids, including Li, Zn, Pb, Au and Ag [4] allows for significant co-recovery. We propose, on the basis of recent developments in our understanding and geophysical imaging of crustal magmatic systems, that mining of sub-volcanic metal-bearing brines may be a fruitful avenue for addressing the impending growth in demand for certain metal resources that will arise during the transition to a low-carbon economy. The economic

potential of such brines hinges critically on the geophysical challenge of locating them and the technological challenge of extracting them to the surface. Regardless of the economic outcomes, direct drilling into hot, sub-volcanic systems is increasingly feasible technically and will provide invaluable scientific insights into both magmatic and ore-forming processes.

Data accessibility. Model results are available from the following link: (https://doi.org/10.5061/dryad.0cfxpnw0w). Further data are provided in the electronic supplementary material [185].

Authors' contributions. J.B. devised the original brine lens mining concept in 2016. It was subsequently refined through discussions with all other authors. B.T. performed the fluid inclusion metal content calculations, microthermometry and analyses. A.A. performed the hydrodynamic simulations; O.M. and I.U. post-processed the results to include copper. All authors helped to draft the final manuscript, gave final approval for publication and agree to be held accountable for the work performed therein.

Competing interests. We declare we have no competing interests.

Funding. J.B. acknowledges funding through a Royal Society Research Professorship (RP\R1\201048); A.A. and O.M. acknowledge funding from Russian Science Foundation under grant no. 16-17-10199. Samples from Larderello were kindly provided by P. Fulignati (University of Pisa).

Acknowledgements. We thank A. Stinton (University of West Indies) for help in obtaining drill-core samples from Montserrat MON-1 well, financial and technical support for which were provided by: Government of Montserrat; UK Foreign, Commonwealth and Development Office, Natural Environment Research council; British Geological Survey; Iceland Drilling and ISOR Iceland Geosurvey; and Baker Hughes. We thank B. Kunz and F. Jenner for assistance with LA-ICPMS analyses, G. Pokrovski for help with the chalcopyrite solubility calculations, J. Stimac for a primer on geothermal systems, and S. Gatehouse, S. Simmons, J. Hedenquist, J. Lowenstern, J. Mavrogenes and N. White for reviews of the manuscript. Analytical work was supported through a research grant from BHP.

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
