## [Peer Review File · Royal Society Open Science]

Review History

RSOS-202192.R0 (Original submission)

Review form: Reviewer 1 (Gregor Borg)

Is the manuscript scientifically sound in its present form?

Yes

Are the interpretations and conclusions justified by the results?

Yes

Is the language acceptable?

Yes

Do you have any ethical concerns with this paper?

No

Have you any concerns about statistical analyses in this paper?

No

Recommendation?

Accept with minor revision (please list in comments)

Comments to the Author(s)

Dear Authors,

My congratulations, this is the best-written manuscript I ever received to review, both content- and language-wise! There are very few minute suggestions for modifications of some phrasings (see Appendix A). It would be nice if you at least mention the other elements that might be recovered (you do that in your conclusions for some) but also for the unwanted "nasties" (e.g. Hg, As). Good luck with your further work on this subject, Gregor Borg

Review form: Reviewer 2 (Jacob Lowenstern)

Is the manuscript scientifically sound in its present form?

No

Are the interpretations and conclusions justified by the results?

No

Is the language acceptable?

Yes

Do you have any ethical concerns with this paper?

No

Have you any concerns about statistical analyses in this paper?

No

Recommendation?

Major revision is needed (please make suggestions in comments)

Comments to the Author(s)

The manuscript is interesting and provides a plausible (?) view of Cu mobility through Cu-rich arc-intrusive environments to create important Cu porphyry deposits. The authors envision that hypersaline brines may contain thousands of ppm Cu and accumulate as liquids for sufficiently long periods of time that they could potentially be "mined" for their metals. It is submitted as a perspective, which is appropriate. It is not a review in that it makes little effort to cite a lot of relevant older literature that first explored the importance of brines in porphyry ore deposits and geothermal systems. It's not a research article in that it extends far into the speculative realm, with relatively little self-skepticism about the practicality of the ideas expoused.

I'll also start by saying that I'm a big proponent of further exploration of this high-temperature magma-hydrothermal interface. There's SO much we don't know, and what we learn there will be of great importance for understanding ore deposits, tapping energy from high temperature environments, and learning how to better monitor volcanic systems. And any article that promotes such exploration, and highlights the fascinating scientific dilemmas we face in this environment, is a paper that I'd like to see published.

Having said that, I think there are some problems with the current manuscript.

The paper's introduction envisions a future where humans can avoid the hundreds of near-surface ore-grade Cu porphyry deposits to instead extract Cu from much deeper, high-temperature intrusions that might be actively creating Cu-rich brines. At this point in time, it is

extremely expensive to drill even a single hole in a 2-4 km-deep high-temperature environment... one that would sample only a tiny volume of liquid brine relative to the entire intrusion-related "liquid-lode" that would contain the envisioned 1.4 Mt of copper. Moreover, once syphoned off, any cooling of the fluid would likely cause precipitation of both the ore and gangue (e.g., silica), reducing the ability of any tapped fluids to flow out of the well. The authors pass this off as an issue for the chemical engineers to solve, but it's a lot more daunting that they admit. . How can you suck all the liquid brine in the cupola up to the surface with only a few wells, or are the permeabilities envisioned to be sufficiently low that the intrusion would behave like an oil field?

In the end, it comes down to: IF the common geophysical anomalies (MT) are actually revealing brines at the tops of intrusions, and IF such brines remain sufficiently S-deficient and high-temperature that they still contain high dissolved Cu, then maybe they COULD be tapped as metal ore IF we can figure out how to prevent the liquid from crystallizing AND IF we can drill into such environments cheaply and IF we can somehow suck the liquid out of this ductile, hot environment to the surface so that we wouldn't have to drill dozens or hundreds of holes. I'm sorry, but this just seems a bit too much, even for a perspective paper.

I just don't think that this concept is really credible. It comes across more speculative than a typical scientific article, but with the veneer that it provides an important insight for our hoped-for carbon-free energy future.

To be publishable, I think the authors need to be a bit more honest with the reader ...but to do so, may sufficiently undermine the premise of the article, that a) these metal-rich brines are as common as implied, and b) that economic recovery is plausible and advantageous compared with our current mines.

Below I list a variety of comments that I think are worth addressing.

It wasn't clear to me how the model in this paper for Cu porphyry generation differs from Weis et al. (2012) or what it is adding that is new. Is this more an effort to reproduce aspects of their model so that you can predict the Cu concentration of the hydrosaline liquids?

Though the paper is well written and with good organization, it's not always clear what you're aiming for, particularly with respect to the Cu porphyry model. Is the goal of the paper to stress the importance of transcristal systems in Cu porphyry generation, or is the purpose to demonstrate the Cu-rich saline fluids are sitting there "ripe-for-the-picking." At times, it's not clear where you are heading, or why. Is the purpose of this paper REALLY just to promote the idea of brine-dissolved Cu for ore generation? Or is it to review some of the authors' other papers on Cu porphyries?

You also seem conflicted about the importance of S. It's clear that a lot of people have argued that the fluids streaming through the ore shell contain a lot of S, either in a separate vapor phase or in supercritical fluids. But that would seem to undermine the argument that the hypersaline brines are sitting there awaiting sufficient S or temperature reduction to allow for precipitation of Cu sulfides. To me, the stability of Cu in the brines without precipitation would be very temporary.

The geophysics of course in another conundrum. These anomalies can be explained by saline fluids, by clays, by sulfides, and partly by temperature variations. It's much more exciting to interpret them as saline brines, but we need to admit that there are (I believe) zero examples of such anomalies that have been confirmed to be hypersaline brines by actual drilling. So this article is built on speculation after speculation. Could these anomalies be showing saline, Cu-bearing brines? Yes. Some could be. But I doubt a high percentage.

I agree that a concept of transcrustal magmas is consistent with the large-scale scavenging of metals and volatiles from large crustal magma bodies to small near-surface ore concentrations. But this concept of Cu and Mo porphyries has been with us for decades. There are hundreds of studies that promote the concept that fluxing of volatiles from deep mafic magmas combined with convective degassing of intermediate magmas are critical to ore formation. There's a rich literature on alkaline magmas providing Cu. There's a literature on sulfide saturation with later breakdown and release of Cu. As the authors point out, though, it is a challenge to tie together this long-term whole-crust view of the magmatic system, with the geological evidence for fracturing, brecciation, and mineralization within a much shorter time scale. It appears that a long-term process of metal accumulation is ultimately finalized and preserved relatively quickly.

I fully agree the rapid crystallization of Cu-sulfides during cooling of hydrothermal fluid could result in high Zn/Cu ratios in fluid inclusions at Kakkonda as well as minimal Cu in the extracted fluids. But the problem here, is you can't have it both ways... if indeed Cu is dumped out of solution during cooling, then it won't stay in solution in lenses of brine sitting in rocks straddling the ductile/brittle transition (~400°C).

Other comments:

Line 66: The central point of Hedenquist and Lowenstern (1994).

Line 83: Worth Looking at Stanton (1994) Ore Elements in Arc Lavas, which is a phenomenal summary of metals concentrations in magmas focused.

Line 112: There are lots of good references for alteration around intrusions, especially around porphyries.

Line 230: If you add CO₂ into the mix, then it's pretty much impossible to have a truly supercritical situation. You'll get a CO₂-H₂O vapor and a liquid. That liquid may then later separately unmix to give you a hypersaline phase and a vapor, the latter or which could probably mix with the early exsolved CO₂-H₂O vapor. My main point is that in a lot of magmas at P<3000 bars, the presence of CO₂ will mean that Cl-bearing magmas will saturate with more than one H₂O-CO₂-HCl-NaCl fluid (not even accounting for the S!). The relevance of subcritical behavior was pointed out by Shinohara et al. (1989), Metrich and Rutherford (1992) and Lowenstern (1994). Roedder's 1984 book made clear the importance of brines in Cu porphyries, and the relevance of unmixing, albeit at lower temperatures.

Line 290: Why mention subduction zones here. How is this relevant? Geothermal systems are also found in a lot of non-arc-settings. Olkaria, Iceland, The Geysers, Salton Buttes.

Line 373: In reality, the discussion didn't really prove that Cu was enriched in these tapped high-T geothermal fluids.

Line 398: A lot may come down to whether there's sulfate in the brines as well.

Line 436; There are also a lot of high Vp anomalies under volcanoes, often linked to crystallized intrusions.

Line 480: Not clear what you're really doing with the model, and why, and how this model is designed to be different than what Weis et al. (2012) were doing.

Line 488: Cloke and Kesler (1979) was the critical paper on recognizing the presence of halite.

Line 490: Fournier (1999) in Economic Geology was an entire paper on the importance of lenses of brines in intrusions. And that paper followed up on another classic paper he wrote in 1987. https://pubs.usgs.gov/pp/1987/1350/pdf/chapters/pp1350_ch55.pdf 1387. Both are widely cited. This topic is not new.

Line 495: That would seem to imply that there's no incursion of a <400°C hydrothermal system for over a million years....?

Line 736: This ignores the fact that the best ore is supergene. Where the sulfide ore is oxidized, dissolved and re-precipitated into a high concentration ore tied to a paleo-water table.

Line 898: It's not clear to me how any recent developments have changed our understanding here. We've known about lenses of brines for 40 years relative to porphyries. We drilled into this environment at Kakkonda in the late 90s. The new modeling, first done by Weis et al. (2012) provided insights that the brines might be located in a small volume, and have relatively high Cu, but whether that Cu would remain un-precipitated for any significant amount of time is speculative. As is the thought that we could mine this liquid at a price even close to what we can get from surface deposits. The tie to MT surveys is still quite speculative as well.

Jake Lowenstern, USGS

Decision letter (RSOS-202192.R0)

Dear Professor Blundy

The Editors assigned to your paper RSOS-202192 "The Economic Potential of Metalliferous Sub-volcanic Brines" have now received comments from reviewers and would like you to revise the paper in accordance with the reviewer comments and any comments from the Editors. Please note this decision does not guarantee eventual acceptance.

[Note that the Associate Editor recommendation was to accept your paper after minor revision, but he did recommend that careful attention be paid to the second reviewers comments in particular and as Subject Editor I felt that the combined recommendations of Associate Editor and reviewers were most consistent with a decision of major revision required.]

Please submit your revised manuscript and required files (see below) no later than 21 days from today's (ie 22-Apr-2021) date. Note: the ScholarOne system will 'lock' if submission of the

revision is attempted 21 or more days after the deadline. If you do not think you will be able to meet this deadline please contact the editorial office immediately.

on behalf of Professor Wolfgang Maier (Associate Editor) and Peter Haynes (Subject Editor)
openscience@royalsociety.org

Associate Editor Comments to Author (Professor Wolfgang Maier):

This is a very interesting and stimulating paper, a view that was shared by one of the reviewers. However, the second reviewer had a more critical perspective and provided a lot of constructive suggestions. While I feel this paper should be published, addressing the second reviewer's comments would be beneficial to the community. This includes discussing the uncertainties inherent in some of the data (notably geophysics!) and their interpretations, and also the prospects of ultimate mining (including the risks and costs of drilling). The reviewer summarised the challenge in his statement quite succinctly: "IF the common geophysical anomalies (MT) are actually revealing brines at the tops of intrusions, and IF such brines remain sufficiently S-deficient and high-temperature that they still contain high dissolved Cu, then maybe they COULD be tapped as metal ore IF we can figure out how to prevent the liquid from crystallizing AND IF we can drill into such environments cheaply and IF we can somehow suck the liquid out of this ductile, hot environment to the surface so that we wouldn't have to drill dozens or hundreds of holes". I personally am a bit more optimistic; Sure, we presently could not mine the brine at a price close to that of surface deposits, but this will ultimately have to be balanced by the environmental costs of surface mining. This could be a very stimulating discussion. Please also address reviewer Lowenstern's request for a more comprehensive review of the past literature on porphyry copper deposits.
Wolf Maier

Reviewer comments to Author:

Reviewer: 1

Comments to the Author(s)

Dear Authors,

my congratulations, this is the best-written manuscript I ever received to review, both content- and language-wise! There are very few minute suggestions for modifications of some phrasings. It would be nice if you at least mention the other elements that might be recovered (you do that in your conclusions for some) but also for the unwanted "nasties" (e.g. Hg, As). Good luck with your further work on this subject, Gregor Borg

Reviewer: 2

Comments to the Author(s)

The manuscript is interesting and provides a plausible (?) view of Cu mobility through Cu-rich arc-intrusive environments to create important Cu porphyry deposits. The authors envision that hypersaline brines may contain thousands of ppm Cu and accumulate as liquids for sufficiently long periods of time that they could potentially be “mined” for their metals. It is submitted as a perspective, which is appropriate. It is not a review in that it makes little effort to cite a lot of relevant older literature that first explored the importance of brines in porphyry ore deposits and geothermal systems. It’s not a research article in that it extends far into the speculative realm, with relatively little self-skepticism about the practicality of the ideas expounded.

I’ll also start by saying that I’m a big proponent of further exploration of this high-temperature magma-hydrothermal interface. There’s SO much we don’t know, and what we learn there will be of great importance for understanding ore deposits, tapping energy from high temperature environments, and learning how to better monitor volcanic systems. And any article that promotes such exploration, and highlights the fascinating scientific dilemmas we face in this environment, is a paper that I’d like to see published.

Having said that, I think there are some problems with the current manuscript.

The paper’s introduction envisions a future where humans can avoid the hundreds of near-surface ore-grade Cu porphyry deposits to instead extract Cu from much deeper, high-temperature intrusions that might be actively creating Cu-rich brines. At this point in time, it is extremely expensive to drill even a single hole in a 2-4 km-deep high-temperature environment... one that would sample only a tiny volume of liquid brine relative to the entire intrusion-related “liquid-lode” that would contain the envisioned 1.4 Mt of copper. Moreover, once syphoned off, any cooling of the fluid would likely cause precipitation of both the ore and gangue (e.g., silica), reducing the ability of any tapped fluids to flow out of the well. The authors pass this off as an issue for the chemical engineers to solve, but it’s a lot more daunting that they admit. . How can you suck all the liquid brine in the cupola up to the surface with only a few wells, or are the permeabilities envisioned to be sufficiently low that the intrusion would behave like an oil field?

In the end, it comes down to: IF the common geophysical anomalies (MT) are actually revealing brines at the tops of intrusions, and IF such brines remain sufficiently S-deficient and high-temperature that they still contain high dissolved Cu, then maybe they COULD be tapped as metal ore IF we can figure out how to prevent the liquid from crystallizing AND IF we can drill into such environments cheaply and IF we can somehow suck the liquid out of this ductile, hot environment to the surface so that we wouldn’t have to drill dozens or hundreds of holes. I’m sorry, but this just seems a bit too much, even for a perspective paper.

I just don’t think that this concept is really credible. It comes across more speculative than a typical scientific article, but with the veneer that it provides an important insight for our hoped-for carbon-free energy future.

To be publishable, I think the authors need to be a bit more honest with the reader ...but to do so, may sufficiently undermine the premise of the article, that a) these metal-rich brines are as common as implied, and b) that economic recovery is plausible and advantageous compared with our current mines.

Below I list a variety of comments that I think are worth addressing.

It wasn't clear to me how the model in this paper for Cu porphyry generation differs from Weis et al. (2012) or what it is adding that is new. Is this more an effort to reproduce aspects of their model so that you can predict the Cu concentration of the hydrosaline liquids?

Though the paper is well written and with good organization, it's not always clear what you're aiming for, particularly with respect to the Cu porphyry model. Is the goal of the paper to stress the importance of transcrustal systems in Cu porphyry generation, or is the purpose to demonstrate the Cu-rich saline fluids are sitting there "ripe-for-the-picking." At times, it's not clear where you are heading, or why. Is the purpose of this paper REALLY just to promote the idea of brine-dissolved Cu for ore generation? Or is it to review some of the authors' other papers on Cu porphyries?

You also seem conflicted about the importance of S. It's clear that a lot of people have argued that the fluids streaming through the ore shell contain a lot of S, either in a separate vapor phase or in supercritical fluids. But that would seem to undermine the argument that the hypersaline brines are sitting there awaiting sufficient S or temperature reduction to allow for precipitation of Cu sulfides. To me, the stability of Cu in the brines without precipitation would be very temporary.

The geophysics of course in another conundrum. These anomalies can be explained by saline fluids, by clays, by sulfides, and partly by temperature variations. It's much more exciting to interpret them as saline brines, but we need to admit that there are (I believe) zero examples of such anomalies that have been confirmed to be hypersaline brines by actual drilling. So this article is built on speculation after speculation. Could these anomalies be showing saline, Cu-bearing brines? Yes. Some could be. But I doubt a high percentage.

I agree that a concept of transcrustal magmas is consistent with the large-scale scavenging of metals and volatiles from large crustal magma bodies to small near-surface ore concentrations. But this concept of Cu and Mo porphyries has been with us for decades. There are hundreds of studies that promote the concept that fluxing of volatiles from deep mafic magmas combined with convective degassing of intermediate magmas are critical to ore formation. There's a rich literature on alkaline magmas providing Cu. There's a literature on sulfide saturation with later breakdown and release of Cu. As the authors point out, though, it is a challenge to tie together this long-term whole-crust view of the magmatic system, with the geological evidence for fracturing, brecciation, and mineralization within a much shorter time scale. It appears that a long-term process of metal accumulation is ultimately finalized and preserved relatively quickly.

I fully agree the rapid crystallization of Cu-sulfides during cooling of hydrothermal fluid could result in high Zn/Cu ratios in fluid inclusions at Kakkonda as well as minimal Cu in the extracted fluids. But the problem here, is you can't have it both ways... if indeed Cu is dumped out of solution during cooling, then it won't stay in solution in lenses of brine sitting in rocks straddling the ductile/brittle transition (~400°C).

Other comments:

Line 66: The central point of Hedenquist and Lowenstern (1994).

Line 83: Worth Looking at Stanton (1994) Ore Elements in Arc Lavas, which is a phenomenal summary of metals concentrations in magmas focused.

Line 112: There are lots of good references for alteration around intrusions, especially around porphyries.

Line 230: If you add CO₂ into the mix, then it's pretty much impossible to have a truly supercritical situation. You'll get a CO₂-H₂O vapor and a liquid. That liquid may then later separately unmix to give you a hypersaline phase and a vapor, the latter of which could probably mix with the early exsolved CO₂-H₂O vapor. My main point is that in a lot of magmas at P<3000 bars, the presence of CO₂ will mean that Cl-bearing magmas will saturate with more than one H₂O-CO₂-HCl-NaCl fluid (not even accounting for the S!). The relevance of subcritical behavior was pointed out by Shinohara et al. (1989), Metrich and Rutherford (1992) and Lowenstern (1994). Roedder's 1984 book made clear the importance of brines in Cu porphyries, and the relevance of unmixing, albeit at lower temperatures.

Line 290: Why mention subduction zones here. How is this relevant? Geothermal systems are also found in a lot of non-arc-settings. Olkaria, Iceland, The Geysers, Salton Buttes.

Line 373: In reality, the discussion didn't really prove that Cu was enriched in these tapped high-T geothermal fluids.

Line 398: A lot may come down to whether there's sulfate in the brines as well.

Line 436: There are also a lot of high V_p anomalies under volcanoes, often linked to crystallized intrusions.

Line 480: Not clear what you're really doing with the model, and why, and how this model is designed to be different than what Weis et al. (2012) were doing.

Line 488: Cloke and Kesler (1979) was the critical paper on recognizing the presence of halite.

Line 490: Fournier (1999) in Economic Geology was an entire paper on the importance of lenses of brines in intrusions. And that paper followed up on another classic paper he wrote in 1987. https://pubs.usgs.gov/pp/1987/1350/pdf/chapters/pp1350_ch55.pdf 1387. Both are widely cited. This topic is not new.

Line 495: That would seem to imply that there's no incursion of a <400°C hydrothermal system for over a million years....?

Line 736: This ignores the fact that the best ore is supergene. Where the sulfide ore is oxidized, dissolved and re-precipitated into a high concentration ore tied to a paleo-water table.

Line 898: It's not clear to me how any recent developments have changed our understanding here. We've known about lenses of brines for 40 years relative to porphyries. We drilled into this environment at Kakkonda in the late 90s. The new modeling, first done by Weis et al. (2012) provided insights that the brines might be located in a small volume, and have relatively high Cu, but whether that Cu would remain un-precipitated for any significant amount of time is speculative. As is the thought that we could mine this liquid at a price even close to what we can get from surface deposits. The tie to MT surveys is still quite speculative as well.

Jake Lowenstern, USGS

===PREPARING YOUR MANUSCRIPT===

one version identifying all the changes that have been made (for instance, in coloured highlight, in bold text, or tracked changes);
 a 'clean' version of the new manuscript that incorporates the changes made, but does not highlight them. This version will be used for typesetting if your manuscript is accepted.

===PREPARING YOUR REVISION IN SCHOLARONE===

- Any electronic supplementary material (ESM).
- If you are requesting a discretionary waiver for the article processing charge, the waiver form must be included at this step.
- If you are providing image files for potential cover images, please upload these at this step, and inform the editorial office you have done so. You must hold the copyright to any image provided.
- A copy of your point-by-point response to referees and Editors. This will expedite the preparation of your proof.

- Ensure that your data access statement meets the requirements at <https://royalsociety.org/journals/authors/author-guidelines/#data>. You should ensure that you cite the dataset in your reference list. If you have deposited data etc in the Dryad repository, please include both the 'For publication' link and 'For review' link at this stage.
- If you are requesting an article processing charge waiver, you must select the relevant waiver option (if requesting a discretionary waiver, the form should have been uploaded at Step 3 'File upload' above).
- If you have uploaded ESM files, please ensure you follow the guidance at <https://royalsociety.org/journals/authors/author-guidelines/#supplementary-material> to include a suitable title and informative caption. An example of appropriate titling and captioning may be found at https://figshare.com/articles/Table_S2_from_Is_there_a_trade-off_between_peak_performance_and_performance_breadth_across_temperatures_for_aerobic_scope_in_teleost_fishes_/3843624.

Author's Response to Decision Letter for (RSOS-202192.R0)

See Appendix B.

Decision letter (RSOS-202192.R1)

Dear Professor Blundy,

It is a pleasure to accept your manuscript entitled "The Economic Potential of Metalliferous Sub-volcanic Brines" in its current form for publication in Royal Society Open Science. The comments of the Editors are included at the foot of this letter.

You can expect to receive a proof of your article in the near future. Please contact the editorial office (openscience@royalsociety.org) and the production office (openscience_proofs@royalsociety.org) to let us know if you are likely to be away from e-mail

contact – if you are going to be away, please nominate a co-author (if available) to manage the proofing process, and ensure they are copied into your email to the journal.

on behalf of Professor Wolfgang Maier (Associate Editor) and Peter Haynes (Subject Editor)
openscience@royalsociety.org

Associate Editor Comments to Author (Professor Wolfgang Maier):

Dear Jon and co,

Thank you for addressing the criticisms and suggestions of the reviewers convincingly and comprehensively. I have recommended that this manuscript can now be accepted as is. I feel this will be a very worthy addition to the journal that the research community will find interesting and stimulating, particularly in this time of increasing recognition of the immense challenges of the green energy transition.

Regards

Appendix A**ROYAL SOCIETY
OPEN SCIENCE****The Economic Potential of Metalliferous Sub-volcanic Brines**

Journal:	Royal Society Open Science
Manuscript ID	RSOS-202192
Article Type:	Perspective
Date Submitted by the Author:	10-Dec-2020
Complete List of Authors:	Blundy, Jon; University of Oxford, Department of Earth Sciences Afanasyev, Andrey; Moscow State University Melnik, Oleg; Moscow State University Tattitch, Brian; University of Bristol Sparks, RSJ; University of Bristol, Department of Earth Sciences Rust, Alison; University of Bristol, Earth Sciences Utkin, Ivan; Moscow State University
Subject:	Volcanology < EARTH SCIENCES
Keywords:	Copper mining, Volcanoes, Ore deposits, Geothermal energy
Subject Category:	Earth and Environmental Science

Author-supplied statements

Relevant information will appear here if provided.

Ethics

Does your article include research that required ethical approval or permits?:

This article does not present research with ethical considerations

Statement (if applicable):

CUST_IF_YES_ETHICS :No data available.

Data

It is a condition of publication that data, code and materials supporting your paper are made publicly available. Does your paper present new data?:

Yes

Statement (if applicable):

Model results are available from the following link:

<https://doi.org/10.5061/dryad.0cfxpnw0w>

<https://datadryad.org/stash/share/QLIGPzubrGmggiApy-a66CQQf-JmD59LEKxktGIm75g>.

Conflict of interest

I/We declare we have no competing interests

Statement (if applicable):

CUST_STATE_CONFLICT :No data available.

Authors' contributions

This paper has multiple authors and our individual contributions were as below

Statement (if applicable):

Blundy developed original concept, designed research project, wrote first draft of manuscript. Afanasyev developed and ran computer MUFITS code. Melnik assisted in interpretation of computer models. Tattitch helped to refine original concept, analysed fluid inclusions, provided insights from Kakkonda system. Sparks assisted in project design. Rust assisted in interpretation of reservoir properties. Utkin assisted with computer model results and inter-model comparisons. All authors gave final approval for publication and agree to be held accountable for the work performed therein

The Economic Potential of Metalliferous Sub-volcanic Brines

Jon Blundy^{1*}, Andrey Afanasyev³, Oleg Melnik³, Brian Tattitch², Steve Sparks²,

Alison Rust², Ivan Utkin³

¹Department of Earth Sciences, University of Oxford, South Parks Road, Oxford

OX1 3AN, UK

²School of Earth Sciences, University of Bristol, Wills Memorial Building, Bristol

BS8 1RJ, UK

³Institute of Mechanics, Moscow State University, 1 Michurinsky Prospekt,

Moscow 119192, Russia

*Author for correspondence: jonathan.blundy@earth.ox.ac.uk

Keywords: Copper mining, Volcanoes, Ore deposits, Geothermal energy

Submitted to Royal Society Open Science

**18 Abstract**

The transition to a low-carbon economy will create significantly increased
demand for a wide range of metals, notably copper, which is extensively used in
electricity generation and transmission, and in electric vehicles. Increased
demand will require new approaches to copper exploration and extraction, with
a particular emphasis on sustainability. Conventional copper mining entails
energy-intensive extraction of relatively low-grade (≤ 1.5 wt% Cu) ore from
large-volume open pits or underground mines and subsequent ore refining. Most
copper derives ultimately from high-temperature, aqueous fluids that exsolve
from magmas, typically in subduction-zone settings. Ore formation, in such cases,
involves phase separation of ascending magmatic fluids to form copper-bearing
hypersaline liquids (or 'brines') and subsequent precipitation of copper sulfides
through cooling and chemical reactions. Electrical conductivity surveys of many
subduction-related volcanoes reveal conductive bodies at around 2 km depth
with properties that resemble lenses of hypersaline liquid hosted in porous rock.
Here we build upon emerging concepts in crustal magmatism to explore the
potential of sub-volcanic brine lenses as an *in situ* source of copper and other
metals. Using hydrodynamic simulations we show that 10,000 years of sub-
volcanic magma degassing can generate a Cu-rich (≥ 7000 ppm) brine lens
containing up to 1.4 Mt Cu in a rock volume of a few km³ at ~2 km depth. Copper
contents are greatest in the hot (≥ 400 °C) core of the brine lens. Direct extraction
of metal-rich brines could represent a novel development in metal resource
extraction that obviates the need for solid-ore processing, and generates
geothermal power as a by-product.

Introduction

Our modern world is reliant on natural resources, from steel for
construction to rare earth elements for high-tech devices. As we transition from
fossil fuel-dominated to more sustainable economies, demand for certain natural
resources will rise dramatically. In a world powered by wind, sun and tides, for
example, copper demand will increase more than five-fold, surpassing known
global reserves well before 2100 (Schipper et al., 2018; Valenta et al., 2019). The
rise in demand for 'critical metals' (e.g. lithium, scandium, cobalt, rare earths)
will be even greater (Grandell et al., 2016). As existing reserves become depleted
and new deposits ever harder to find, it is unclear how the extra demand can be
met; recycling alone will be insufficient (Schipper et al., 2018).

Discovering natural mineral resources is, ultimately, a geological problem.
The overwhelming majority of non-ferrous metals derive from igneous processes
associated with magmatism. Non-ferrous metal ores can be viewed as extreme
end-products of igneous geochemical cycles that begin in the Earth's mantle and
conclude with discharge of hot magmas and gases at the surface. Volcanic
volatile phases are particularly efficient transporters of metals as evidenced by
epithermal ore deposits and the chemistry of fumarole gases. Some active,
subduction-related basaltic volcanoes discharge metal-rich gases with a time-
averaged flux to the atmosphere of more than 10^4 kg/day of copper and zinc,
along with a slew of other metals (e.g. silver, tungsten, indium, tin, lead,
molybdenum) at fluxes of up to 1000 kg/day (Edmonds et al., 2018). This
observation emphasises the centrality of igneous processes to formation of
mineral resources, and raises the possibility of recovering metals directly from
modern volcanic fluids as an alternative to mining ancient solid ore deposits.

Here we discuss the potential for *in situ* mining of hot, metalliferous volcanic
fluids¹, with a particular emphasis on copper, in light of recent developments in
our understanding of the dynamics and architecture of crustal magmatic
systems.

15 73 **Porphyry Copper Deposits**

Porphyry copper deposits (PCDs) provide ~75% of the world's copper,
plus significant molybdenum and gold (Sillitoe 2010). PCDs are typically
associated with subduction-related magmatism involving oxidised, H₂O-rich
silicic magmas, themselves derived by differentiation of mantle-derived, hydrous
basalt magmas (Richards 2015). Magmatic differentiation within the crust
elevates contents of dissolved H₂O and other volatiles, including important
metal-complexing ligands, such as Cl⁻ and HS⁻, while imparting geochemical
signatures characteristic of differentiation at lower- or mid-crustal pressures
(Annen et al., 2006; Loucks 2014; Chiaradia & Caricchi, 2017). Unless the
parental basalts are unusually Cu-rich, for which there is little evidence
(Chiaradia 2014), a key process for Cu enrichment and ore formation is efficient
extraction of Cu by hydrous fluids exsolved from evolved magmas as they ascend
and crystallise. Such fluids are associated with active dacitic, andesitic and
basaltic andesite volcanoes (e.g. Giggenbach, 1992; Hedenquist et al. 1993;
Hedenquist & Lowenstern, 1994; Taran et al., 1995) both during and between

¹ We use the term *fluid* to describe any flowing geological phase, including volcanic gas, low-salinity vapour, hypersaline liquid and silicate melt. *Supercritical fluids* are those beyond the critical end point in volatile-dominated systems, i.e. at pressures above phase separation into liquid and vapour. Magmatic *volatiles* include the species H₂O, CO₂, SO₂, H₂S, NaCl, HCl and other halides. The process of volatile release from silicate melts is described broadly as *degassing* regardless of the exact state of the fluid released.

eruptions of magma (Shinohara, 2008; Christopher et al 2015). These fluids are
predominantly H₂O with lesser quantities of CO₂, SO₂, H₂S and halogens and a
wide range of trace metals (Williams-Jones & Heinrich, 2005; Simmons & Brown,
2007).

The relationship between magmatic fluids involved in sub-surface PCD
formation and those discharged directly to the atmosphere is complex. Phase
changes associated with fluid decompression and cooling can modify the
chemistry of the original fluid exsolved from magma. For example, phase
separation of H₂O-NaCl fluids into coexisting low (vapour) and high (liquid)
density fluids can have a profound effect on metal contents. To form an ore
deposit, some component of the original magmatic fluid must become trapped
and cool within the crust, depositing its metal load in the form of ore minerals.
Copper mineralisation typically takes the form of copper-bearing sulfides, such
as chalcopyrite and bornite. Reduced sulfur (S²⁻) is therefore a key ingredient in
the formation of PCDs. Similarly, the primacy of the chloride (Cl⁻) ligand in
transporting Cu (Candela and Holland 1984; Simon et al. 2006; Frank et al. 2011;
Zajacz et al. 2011; Kouzmanov and Pokrovsky 2012; Tattitch and Blundy 2017)
requires that ore-forming magmatic fluids are saline, that is, they contain
chloride, often expressed in terms of an equivalent amount of sodium chloride
(NaCl_{eq}). Characteristic and extensive rock alteration haloes around PCDs,
containing a variety of secondary minerals, such as sericite mica and clays, testify
to the highly reactive nature of hot magmatic fluids when they come into contact
with cooler igneous rocks. The reactant fluids responsible for alteration include
both primary, hot ore-forming fluids and cooler, acidic exhaust gases produced
once mineralisation has occurred. Heated external water (groundwater,

seawater) may also mix with magmatic fluids and participate in hydrothermal
reactions. The eventual style of alteration is a complex interplay between the
temperature and composition of these fluids and their ascent path. Formation of
extensive, hydrothermally altered 'lithocaps' above PCDs results from reactions
involving fluids of different provenance (Sillitoe, 2010).

PCDs are spatially associated with hypabyssal intrusions emplaced at
depths of 1-4 km (Sillitoe 2010) that provide focused pathways for magmatic
fluids exsolved deeper in the system (Cloos 2001). These depths correspond to
the upper reaches of magma reservoirs (or 'chambers') that underlie active
volcanoes (e.g. Huber et al., 2019). The hypabyssal intrusions themselves are
sourced from these magma reservoirs; some intrusions may reach the surface
and represent conduits for volcanic or phreatomagmatic eruptions, lava dome
extrusions or brecciation events. Solidified shallow magma reservoirs take the
form of plutons with a carapace of thermally metamorphosed, and variously
hydrothermally altered, country rock.

40 41 130 **Transcrustal Magmatic Systems**

PCDs form in the upper crust and hitherto the focus of attention with
respect to the magmatic sources of the ore-forming fluids has been a large
shallow magma chamber (e.g. Cloos, 2003; others). However, the paradigm of
crustal magmatism is evolving, and it has become clear that processes pertinent
to the origin of magmas and fluids extend to much greater depths. Rather than
large, long-lived vats of melt-rich magma, shallow magma reservoirs are
increasingly viewed as an ephemeral expression of much larger, vertically-
extensive, and longer-lived magma systems in which partially molten rocks or

'mushes' dominate (Cashman et al., 2017; Sparks et al., 2019). Such systems may
extend throughout the crust as a continuous connected system or as discrete
separated reservoirs that become interconnected during magma ascent. The
architecture of transcrustal magmatic systems has some important implications
for ore formation.

In the emerging transcrustal concept (Figure 1) magmas and fluids stored
in and released from shallow sub-volcanic reservoirs represent the time-
integrated products of processes operating over a significant vertical extent; they
reflect protracted sequences of magmatic differentiation and fluid exsolution
over a wide pressure-temperature range. This is particularly important in the
case of magmatic volatile species because of the strongly pressure-sensitive
nature of their solubility in silicate melts (Lesne et al., 2011; Burgisser et al.,
2015). Consequently, the composition of the magmatic volatile phase exsolved
from magmas is a sensitive function of the pressure (depth) at which degassing
occurs.

Numerical models of mushy 'hot-zone' systems (Bergantz, 1989; Annen &
Sparks, 2002; Annen et al., 2006; Solano et al., 2012; Karakas & Dufek, 2015;
Blundy & Annen, 2016; Jackson et al., 2019) indicate that they typically form
over periods of hundreds of thousands to several million years through the
sequential emplacement of mantle-derived magmas into the mid- or lower crust.
Over time temperatures and melt fractions in the hot zone increase, eventually
forming a substantial, vertically-extensive body of partially molten rock. The
deep-seated, partially-molten nature of hot zones beneath a thick insulating roof
confers a longevity that is not possible for shallow, liquid-rich magma chambers
that are limited due to thermal constraints (Annen et al., 2015). Such shallow

large magma chambers are now considered ephemeral and associated with
episodes of relatively high magma flux from the hot zone into the upper crust.

Hot zones are not static; they are dynamic mixtures of relatively dense
solids and buoyant melts and fluids. Reaction of ascending melts and fluids with
their surrounding rocks, including partially solidified ancestral magmas, is
considered to be an important process in chemical evolution of magmas (e.g.
Solano et al., 2012; Jackson et al., 2019) and fluids (e.g. Mavrogenes & Blundy,
2017). Reactive flow enables melts and fluids generated deeper in the system
(i.e., at higher pressure) to encounter and react with shallow-stored (lower
pressure) components of the same magmatic system. The intrinsic gravitational
instability of mush systems means that melt and fluid ascent can be either
percolative (steady state) or catastrophic (Sparks et al., 2019). Release of melts
from hot zones may lead to eruption at the surface or formation of ephemeral
shallow magma chambers and shallow-level plutons, themselves constructed
incrementally over extended periods of time (Annen, 2009). The highly reactive
nature of fluids derived from deeper parts of a magmatic system has a number of
implications for ore deposits (e.g. Mavrogenes & Blundy, 2017), as discussed
further below

Fluids released from hot zones, transport heat and fluid-mobile elements,
including copper, upwards. The ascent of melt and fluid from hot zones may be
decoupled in time and space (Christopher et al., 2015). The endowment of the
world's most economic PCDs (>10 Mt contained Cu) requires that copper be
extracted efficiently from significant volumes of magma, an observation broadly
consistent with transcrustal magmatic systems rather than discrete shallow
plutons (Chiaradia & Caricchi, 2017). The abundance of brecciation and veining

features in PCDs (Sillitoe, 2010), and high-resolution radiometric evidence that
relatively short-lived pulses of mineralisation occur at the end of relatively long-
lived magmatic episodes (e.g. Barra et al., 2013; Chelle-Michou et al., 2014;
Tapster et al., 2016; Buret et al., 2016), suggest that catastrophic, large-scale fluid
release events are a key step in PCD formation. Thermal modelling of
incrementally-assembled igneous bodies shows that fluid release can occur in
relatively brief pulses against a background of longer-lived magma accumulation
and crystallisation (Schöpa et al., 2017; Chiaradia & Caricchi, 2017). Ultimately
the upward flux of melt and fluids is controlled by the evolving permeability of
the reservoirs in which they form (Degruyter et al., 2019). Currently, this is not a
well-understood aspect of the multiphase, mushy systems that are thought to
underlie volcanoes.

**Magmatic volatiles**

The composition of magmatic fluids depends on the relative solubilities of
the key volatile species in silicate melts that in turn depend on pressure.
Generally speaking, fluids released deeper in the system are richer in CO₂ than
those released at shallower levels, due to the greater solubility of H₂O than CO₂
(e.g. Burgisser et al., 2015). Thus, fluids released from magmas at depth in the
crust will differ from the fluids in equilibrium with magma stored at shallow
depth, with abundant opportunities for chemical reaction, including flushing of
shallow magmatic systems with deep-derived CO₂, for which there is abundant
petrological evidence (e.g. Caricchi et al., 2018). The behaviour of minor fluid
species (e.g. NaCl, HCl, SO₂, H₂S) is more complex and, in the case of sulfur
species, dependent on redox state. For example, in their experimental study of

degassing of oxidised basalt magma Lesne et al. (2011) show that magmatic
chlorine is retained in melts to lower pressures than SO₂.

The volcanic fluids emitted from active basaltic volcanoes are dominated
by H₂O (±CO₂) and have relatively low salinity (<1 wt% NaCl_{eq}; Aiuppa et al.,
2009; Edmonds et al, 2018); their composition at the surface closely resembles
that of the fluid at the point of shallow exsolution. The situation is different for
magmatic fluids that exsolve from evolved, more chlorine-rich silicic magmas at
greater depths where higher salinity (>1 wt% NaCl_{eq}) fluids are generated.
Thermodynamics of the NaCl-H₂O system (Henley and McNabb 1978; Driesner
and Heinrich 2005) define a two-phase region of hypersaline liquid (often
referred to as 'brine') and low-salinity vapour. The two-phase region is delimited
by a solvus whose location depends on temperature and pressure (Figure 2). At
the pressure and temperature of initial volatile exsolution from H₂O-rich silicic
magmas, most magmatic fluids likely exist as a single, super-critical fluid of
intermediate density (salinity ~2 to 12 wt% NaCl_{eq}; Kouzmanov and Pokrovski
2012) due to volatile saturation at pressures above the NaCl-H₂O solvus (Cline
and Bodnar 1991). However, direct exsolution of high salinity fluids from some
silicic magmas is also known to occur (Kasai et al. 1998a,b; Kodera et al. 2014;
Webster et al. 2015; Fulignati 2017, 2018). Fluid inclusions in PCDs support
involvement of parental, single-phase intermediate-density (ID) fluids of modest
salinity (4-10 wt% NaCl_{eq}; Audetat et al. 2008; Landtwing et al. 2005; Klemm et
al. 2008; Seo et al. 2012).

Although primary intermediate density fluids typically contain ≤2000 ppm
Cu (Kouzmanov and Pokrovsky 2012), the proclivity of copper (and many other
metals) for the chloride ligand means that the separated hypersaline liquid phase

becomes strongly metal-enriched. Upon phase separation, Cu partitions into the
liquid, in almost direct proportion to the salinity ratio between liquid and vapour
(Kouzmanov and Pokrovsky 2012; Tattitch and Blundy 2017). This ratio
depends on the pressure and temperature of phase separation (Figure 2), thus
higher Cu contents are found in higher salinity liquids formed at lower pressures
and higher temperatures. Globally, Cu contents of hypersaline fluid inclusions
from PCDs range from 3 ppm to 3 wt% with salinities of 25 to 73 wt% NaCl_{eq}
(Kouzmanov and Pokrovski 2012). The upper end of this range is considerably
more Cu-rich than the original fluid that exsolved from the ~~parent~~ silicic magma,
a testament to the Cu-sequestering ability of brine. The low salinity vapour
counterpart that discharges to the atmosphere is correspondingly depleted in
metals.

Numerical models of magmatic fluid flow through the shallow crust (Weis
et al. 2012; Weis 2015; Afanasyev et al. 2018) predict that the dense, hypersaline
liquid phase becomes physically isolated from its low-density vapour
counterpart, and can be retained at depth in rock pore space, such as
intergranular pores, cavities and fractures. Thus a significant fraction of the
original metal budget of silicic volcanoes can become trapped within fluids in
porous rocks at depth, rather than be discharged to the atmosphere or retained
in the cooled crystallized mass. These considerations raise the possibility of the
existence of metal-rich hypersaline liquid reservoirs beneath active volcanoes
with attendant economic potential. Although few, if any, major PCD-like deposits
are likely to underlie modern volcanoes, the ubiquity of active volcanoes may
partly counterbalance their small size justifying a careful evaluation of their
economic potential.

An important consideration in PCD formation is whether Cu-bearing
fluids contain adequate reduced sulfur to form an ore body. Using data from melt
inclusions in volcanic phenocrysts, Blundy et al. (2015) argues that the Cl- and
Cu-rich fluids responsible for PCD formation are unlikely to contain sufficient
sulfur because of the different degassing behaviour of Cl and S, as described
above. In response to this sulfur deficiency, Blundy et al (2015) propose that
sulfur is delivered at a later, ore-forming stage, precipitating sulfides through
reaction with metal-rich brines. They presented experimental evidence to
support this proposal. Mavrogenes & Blundy (2017) used experiments to explore
the possibility that SO₂ could also drive metal precipitation from brines, via Ca-
mediated disproportionation reactions that produce both S²⁻ and anhydrite
(CaSO₄). Such reactions are to be expected in transcristal systems where fluids
from deeper in a system can interact with shallow-stored fluid in the same
transcristal system. Conversely, Seo et al. (2012) use analyses of S in fluid
inclusions from Bingham Canyon PCD to argue that ample sulfur, of unknown
redox state, is available in stoichiometric proportion to form chalcopyrite with
dissolved Cu and Fe in the same inclusions, such that there is no need for any
later addition of sulfur from deeper in the system. Unfortunately, sulfur is
notoriously hard to measure precisely in fluid inclusions by standard mass
spectrometric methods, and the sulfur budget of ore-forming fluids remains a
critical unknown in our understanding of PCD formation. We return to this
uncertainty below.

**Metals in Geothermal Systems**

The heat and fluids associated with shallow magmatic systems are often
harnessed through several forms of geothermal energy production. Geothermal
energy is extracted from a range of volcano types, but a high proportion of
systems in operation today are associated with subduction zone volcanoes.
Conventional geothermal wells are drilled into the upper reaches of magmatic
hydrothermal systems, and provide insights into the composition and
distribution of fluid phases. The fluids accessed in geothermal systems are
typically dominated by a variety of non-magmatic fluids, including sea-water and
meteoric water, variably modified by heating, water-rock interactions, phase
separation and mixing, with limited chemical input from magmatic derived fluids
(Stimac et al. 2015)².

A lesser number of high temperature or “supercritical” geothermal systems
(Dobson et al., 2017; Agostinetti et al., 2017) utilize wells that do access high-
temperature (>400°C) accumulations of magma-derived fluids. In a few of the
deepest wells, drilled into young, hot magmatic systems, samples of true
magmatic hypersaline liquids have been recovered directly. For example, at
Larderello (Italy) Cathelinau et al. (1994) and Fulignati (2018) report quartz-
hosted hypersaline liquid inclusions (30-70 wt% NaCl_{eq}) recovered from granite
host rocks at depths of 3-4 km and temperatures of 425-650 °C, consistent with
measured down-hole temperatures. Active accumulations of hypersaline liquids
(39 to 55 wt% NaCl_{eq}) have been identified at depths in excess of 3.5 km and
temperatures above 500 °C from the Kakkonda (Japan) geothermal field (Kasai et
al. 1998a,b). Recovered fluids contain 3400 ppm Zn and 1200 ppm Pb, with

²Although it is common in many of these geothermal systems to refer to any fluid
with total dissolved solids (TDS) >0.1 to 1 wt% as “brine”, these are distinct from
the truly hypersaline liquids that are our focus here.

estimates of >5000 ppm Zn in the original (~55wt% NaCl_{eq}) hypersaline liquid
(Kasai et al. 1998a,b, Uchida et al. 1998). Copper, however, is at concentrations
below detection in the recovered hypersaline fluids; this is thought to be due to
rapid Cu-sulfide precipitation during cooling and decompression of fluids in the
wellbore prior to sampling (Kasai et al. 1998a). Measurements of fluid inclusions
from this system indicate temperatures above 500 °C and salinities above 50
317 wt% NaCl_{eq} (Sasaki et al. 1995) consistent with sampling conditions of the well-
318 bore fluids. Evidently, fluid inclusions from drill-core provide reliable insights
into the temperature and chemistry of fluids in a deep geothermal well.

There are currently no direct trace metal analyses of deep magmatic fluid
inclusions from Kakkonda or Larderello. However, insights into their chemistry
can be gained from published petrographic studies. Detailed examination of
quartz-hosted fluid inclusion images from Kakkonda (Sasaki et al. 1998;
Fujimoto et al. 2000) and Larderello (Fulignati 2018) reveals that they contain
abundant opaque daughter minerals indicative of substantial chalcophile metal
concentrations in the fluids, similar to those observed in metalliferous
hypersaline liquid inclusions from experiments (Frank et al., 2011; Tattitch and
Blundy 2017) and from ore deposits (e.g. Ulrich et al. 1999) (Figure 3a-e). From
the images we can estimate the bulk composition of the fluid inclusions using
pixel maps and volume calculations for cubic halite crystals, tetrahedral or cubic
sulfides (chalcopyrite or pyrite/pyrrhotite) and spherical bubbles, compared to
the remaining liquid volume of the inclusion. For fluids with known salinities and
Cu contents (Figure 3a,b), a combination of volume calculations with densities
for observed solid phases recovers salinities (45-60 wt% NaCl_{eq}) and Cu contents
(5000-7000 ppm) that lie within 1 sigma of the average for the range measured

by laser-ablation inductively-coupled mass spectrometry (LA-ICPMS) on the
same inclusions. This gives us confidence in our methodology when applied to
images of fluid inclusions with unknown Cu concentrations. Assuming, on the
basis of crystal shapes, that the opaque phases from Kakkonda and Larderello
are sulfides, either chalcopyrite (34% Cu) or pyrite/pyrrhotite that is saturated
in chalcopyrite (~3.5% Cu), we obtain estimated Cu concentrations with an
upper range of 2500-5000 ppm for Kakkonda (Figure 3c,d) and 1500-2500 ppm
for Larderello (Figure 3e,f). No explicit measure of the uncertainty on these Cu
estimates is possible. However, any large variations in inclusion thickness would
create a mismatch between the calculated inclusion salinity and that reported.
Our estimated values are not meant to define accurately the Cu concentration,
but simply to show that fluid inclusion petrography is consistent with Cu
concentrations at levels similar to those observed in magmatic brines from
mineralized systems.

Direct analyses of fluid inclusions from high-temperature, magma-sourced
geothermal systems, such as those described here, by LA-ICPMS can provide
much more accurate constraints of the contents of Cu and other metals. To date,
very few such analyses have been reported in the literature. One set of fluid
inclusion LA-ICPMS data is available for moderate temperature (~400°C)
Kakkonda hypersaline liquids (Sasaki et al. 1998). Measured Zn/Cu \approx 7 for some
of the assemblages. Combined with other direct measurements of Zn, this yields
Cu concentrations from ~500 ppm to >1000 ppm in the undisturbed high-
temperature hypersaline liquid reservoir. These values are likely minima given
the evidence for substantial precipitation of Cu, and later Zn, as the fluid
migrates through its host rocks and cools (e.g. Sasaki et al. 1998; Fujimoto et al.

2000). Copper loss from fluids through precipitation of sulfides was further
exacerbated during well-sampling at Kakkonda (Kasai et al. 1998a, 2003),
consistent with the lack of measurable Cu in fluid samples extracted to the
surface (Kasai et al. 2003; Sasaki et al. 1998; Fujimoto et al. 2000), and the
occurrence of abundant Cu-rich well-bore scales (Yanagisawa et al. 2000). Scales
collected from variable depths in drill holes #13, #19, and #21 contained
chalcocite, Cu-Fe arsenides, sphalerite, galena and pyrite; each of the scales also
contains significant but variable amounts of silica-alumina phases, sulfates and
carbonates (Fujimoto et al. 2000). All of these mineral phases are likely to have
precipitated directly from hot, ascending fluids, testifying to their metal-rich
character. Cu contents of the analysed scales are in the range 0.27 to 14 wt% Cu,
with highly variable Zn/Cu ratios of 0.002 to 74 (Yanagisawa et al. 2000).

These two examples of deep, hot geothermal drilling suggest that high-
temperature, magmatic hypersaline liquid lenses can have significant base metal
concentrations, including Cu, consistent with the composition of sub-volcanic
hypersaline liquids observed in porphyry ore-forming environments. In other
geothermal reservoirs, such as Geysers (USA; Moore and Gunderson 1995) and
Mori (Japan; Muramatsu and Komatsu 1999), mixing of meteoric water or
seawater with magmatic fluids likely reduces metal contents, although there are,
as yet, no direct trace metal analyses of fluid inclusions to test this possibility.

The presence of reduced sulfur is critical to formation of ore minerals, yet
gauging the content and redox state of sulfur in magmatic fluids faces
considerable analytical challenges. Consequently, the sulfur budget of ore-
forming fluids remains a critical unknown in our understanding of PCD
formation. Limited available data for intermediate density (ID) fluid inclusions

from PCDs indicate sulfur contents in the range 200 ppm to 3 wt% (Kouzmanov
and Pokrovski 2012) albeit with analytical uncertainties of at least ± 20 -50%
relative (e.g. Seo et al, 2012). In almost all ID fluid inclusion analyses assembled
by Kouzmanov and Pokrovski (2012) sulfur is stoichiometrically in excess of that
required to form chalcopyrite (CuFeS_2). If primary magmatic, saline fluids
contain sufficient reduced sulfur to precipitate sulfides, then phase separation
and ore mineralisation may be near simultaneous processes, driven by a
combination of fluid depressurisation and cooling (e.g. Weis 2015).

Conversely, using data from melt inclusions in volcanic phenocrysts,
Blundy et al. (2015) argue that the Cl- and Cu-rich fluids responsible for PCD
formation are unlikely to contain sufficient sulfur because of the different
degassing behaviour of Cl and S, as described above. In response to this sulfur
deficiency, Blundy et al (2015) propose that sulfur is delivered at a later, ore-
forming stage, precipitating sulfides through reaction with metal-rich brines.
They presented experimental evidence to support this proposal. Sulfur supply
from deeper parts of a magmatic system is to be expected in transcrustal systems
of the type described above, and is consistent with the 'excess sulfur'
phenomenon at many volcanoes, whereby the atmospheric supply of sulfur
exceed that which could have been dissolved in the erupted magma (Wallace &
Edmonds, 2011).

If ore-forming fluids are sulfur-poor, such that additional sulfur is required
via supply of SO_2 or H_2S from deeper in the magmatic system, then there may be
a time lag between phase separation and sulfide precipitation (e.g. Blundy et al.
2015). Although ore minerals are predominantly sulfides, it is not essential for
all sulfur in the fluid to be in the reduced state. Disproportionation of SO_2 to form

S^{2-} and S^{6+} is a potent means to create sulfides by reaction with a metal-bearing
solution, especially in the presence of Ca^{2+} ions when anhydrite ($CaSO_4$) is an
additional precipitate (Mavrogenes & Blundy, 2017). Coexistence of anhydrite
and metal sulfides is common both in PCDs (e.g. Gustafson & Hunt, 1975; Sillitoe,
2010; Mavrogenes & Blundy, 2017) and in geothermal wells, including Kakkonda
(Sasaki et al., 2008).

Regardless of the details of the sulfur source, a main driver for sulfide
mineral precipitation from fluids is cooling, as solubility of both chalcopyrite and
bornite is strongly temperature dependent, at least to temperatures below 400
420 °C where experimental data are available (Kouzmanov and Pokrovski. 2012).
Sulfide solubility also depends on fluid salinity, pH, redox state and sulfur
fugacity. For example, chalcopyrite solubility in a pyrite-magnetite-hematite
buffered, 40 wt% $NaCl_{eq}$ fluid with pH=5 at 0.3-1.0 kbar falls from 2 wt% Cu at
550 °C to 2 ppm at 200 °C (Fig. 4; and Kouzmanov and Pokrovski 2012). This
explains the very low Cu content of cool, neutralised, low-salinity geothermal
fluids. Unfortunately, metal sulfide solubilities in saline fluids above 400 °C are
not presently well constrained experimentally.

45 429 **Sub-volcanic brine lenses**

The feeder zones (conduits) of volcanoes, through which hot, saline
magmatic fluids pass, comprise hypabyssal intrusions (dykes and sills),
pyroclastic material and fractured and brecciated rocks, typically with associated
hydrothermal alteration and quartz veins (e.g. Stasiuk et al. 1996; Goto et al.
2008). High porosities (e.g. ≥ 20 vol% under Unzen volcano; Ikeda et al. 2008) are
a characteristic of such systems, consistent with reduced P-wave velocities at

depths ≤ 8 km beneath active arc volcanoes (e.g. Kiddle et al. 2010). Average
porosity of both dense and fragmental rocks decreases with increasing depth,
but cores and well logs show that active geothermal systems can maintain
porosities (matrix plus fracture) of 10 to 30 vol% down to depths to 2.5 km
(Figure 5), with fragmental materials typically the most porous; other lithologies,
such as vapour-phase-altered pyroclastic materials, are notably resistant to
densification by physical and chemical processes and may preserve significant
primary porosity. Sub-volcanic conduits with associated damage zones related
to prior intrusions, eruptions and steam explosions have very high porosities
(Fig. 5), and provide efficient pathways for both fluid flow and accumulation. The
presence of fractures can lead to a significant mismatch between relatively high
porosity at a reservoir scale and relatively low porosity on a drill-core scale, as
evidenced by data and modelling at Kakkonda (Fig. 5). It should be noted also
that precipitation from (and dissolution by) ascending fluids will modify the
original porosity structure in the feeder zone.

To investigate the potential of sub-volcanic conduits as a means of
focussing and storing magmatic fluids, Afanasyev et al. (2018) performed
hydrodynamic models of discharge through a sub-volcanic domain of a 738°C
supercritical aqueous fluid (4 wt% NaCl) from a magma body located at 7 km
depth. Afanasyev et al. (2018) considered explicitly the effect of a volcanic
conduit region composed of high permeability, fractured rock. The porosity-
depth distribution used by Afanasyev et al. (2018) throughout the modelled
domain is presented in Fig. 5 for comparison to the available natural data. For
the reference scenario (total fluid flux of 3×10^9 kg/yr through a 0.33 km radius

conduit of permeability $k \leq 10^{-14} \text{ m}^2$) Afanasyev et al. (2018) presented
calculations for degassing durations of up to 250 kyr.

It is instructive to adapt the simulations of Afanasyev et al. (2018) to White
Island volcano (New Zealand), a plausible modern analogue to the scenario being
explored. Hedenquist et al. (1993) report a White Island flux of $3.5 \times 10^5 \text{ kg/day}$
of SO_2 via a fumarolic fluid containing, on average, 3.5 wt% SO_2 at temperatures
up to 800 °C. The corresponding total fluid flux is $3.7 \times 10^9 \text{ kg/yr}$, a value similar
to that used by Afanasyev et al. (2018). The Cu content of fluids discharged from
White Island, based on the reported Cu/ SO_2 ratio, is 30 ppm. This is likely a
minimum value for the original magmatic fluid, due to sub-surface deposition of
Cu beneath the edifice (Hedenquist et al. 1993) as well as phase separation of Cu-
rich brines. The total duration of degassing from White Island is not known – for
the purposes of the calculations reported here we consider only the 10 kyr case.
Note that this is a relatively short degassing period in the context of the long-
term growth and differentiation of crustal magmatic systems, consistent with the
notion of pulsatory fluid release discussed above. Despite the overall similarity
of modelled conditions to those at White Island, we recognise that fluid fluxes
and metal contents may vary significantly from one volcanic system to another.
For that reason our calculations are designed to be illustrative rather than
definitive.

Afanasyev et al.'s (2018) calculations predict accumulation of relatively
dense (c. 1350 kg m^{-3}) hypersaline liquids at ~2 km depth in annular lenses
surrounding the conduit (Figure 6a), in close agreement with the calculations of
Weis et al. (2012) and Weis (2015) despite some differences in the modelling
approaches. A direct comparison of our models and those of Weis (2015) is

provided in Supplementary Figure 1. The numerical models predict that the
hypersaline liquid lens is capped by halite precipitation that occludes pore space
and restricts further upward movement. Halite precipitation has been described
from ore deposits (Kodera et al. 2014; Lecumberri-Sanchez et al 2015) and
geothermal reservoirs (Hesshaus et al. 2013; Nitschke et al. 2017). In a
parametric analysis Afanasyev et al. (2018) show that the hypersaline liquid lens
is not well developed for lower temperature fluid inputs (<700 °C) or for very
wide ($r > 1$ km) conduits, but is enhanced for narrow conduits, and hotter, more
saline input fluids. Modelled lenses persist for up to 1 Myr after degassing ends,
eventually sinking, spreading out, and being diluted and dissipated by meteoric
water convection (Afanasyev et al. 2018).

49630
**Geophysical Observations**

Geophysics affords considerable insight into the architecture and dynamics
of crustal magma systems (Magee et al., 2018). Techniques include potential
fields (electrical conductivity, gravity), seismic wave speeds and tomography,
and geodesy (ground- and satellite-based). Increasingly these techniques are
being applied to individual volcanic centres to elucidate magma plumbing.
Although the images differ from volcano to volcano, a unifying theme of most
geophysical surveys of volcanoes is the presence of relatively large volumes of
reduced seismic wave speed and enhanced electrical conductivity at depths >1
506 km consistent with large distributed volumes of partial melt and/or magmatic
fluids rather than a single large-liquid-filled magma vat. Real-time geodetic
surveys indicate that such systems are dynamic on timescales of years to
decades (Pritchard & Simons, 2004; Biggs et al., 2014), likely due to the relative

mobility of fluids (e.g. Gottsmann et al., 2017). Unfortunately, a shortcoming of
all geophysical techniques, especially when applied in isolation, is their non-
uniqueness, especially when it comes to resolving relatively small features, and
establishing the abundance, composition and distribution of the fluid (melt, gas)
phase.

From the perspective of sub-volcanic brine lenses, electrical conductivity
has the greatest potential as an identification and mapping tool. Electromagnetic
surveys, including magnetotelluric (MT) and time-domain (TDEM), have been
conducted at several active and dormant subduction-related volcanoes: e.g.
Mount Fuji, Japan (Aizawa et al. 2005); Taal, Philippines (Yamaya et al. 2013);
Kusatsu-Shirane, Japan (Nurhasan et al. 2006); Uturuncu, Bolivia (Comeau et al.
2016); Unzen, Japan (Srigutomo et al. 2008, Komori et al. 2013, Trihadini et al.
2019); Taupo Volcanic Zone, New Zealand (Bertrand et al., 2015, Heise et al
2015, Hill et al. 2015); and Mount St. Helens, USA (Hill et al. 2009; Bedrosian et
al. 2018). These consistently show prominent, electrically conductive (≥ 1 S/m)
regions at depths ≥ 2 km, interpreted by Afanasyev et al. (2018) to represent
hypersaline liquid accumulations. However, other interpretations are possible,
and have been advanced by the authors of some of these studies. For example,
conductive lenses may be smectite clays, yet these are stable only below ~ 250 °C
(Stimac et al. 2015), temperatures well below those expected at such depths in
the active volcanic systems imaged. Moreover, such layers, where present, e.g.
Rotokawa (Bertrand et al. 2015), Kusatsu-Shirane (Nurhasan et al. 2006) or
Montserrat (Ryan et al., 2013), tend to be laterally persistent and lie at depths < 1
533 km. Conversely, the conductive lenses might be magma, although the lower
electrical conductivity of silicate melt compared to hypersaline liquid would

require high melt fractions (>50%) at relatively shallow depths, which is
commonly inconsistent with other geological evidence (e.g. Comeau et al. 2016;
Trihadini et al. 2019). Finally, the conductor may be composed, in part, of
conductive sulfide minerals, although these would need to be fully connected in
order to generate the observed anomalies, behaviour that is more consistent
with a fluid, rather than solid, conductor phase. Notably, the calculated depth,
geometry and electrical conductivity of the modelled and imaged hypersaline
liquid lenses are very similar (Afanaysev et al. 2018). In a number of MT studies
the conductor at 2 km depth extends to the middle or lower crust (e.g. Hill et al.
2009; Comeau et al. 2016; Bertrand et al., 2015, Heise et al 2015, Hill et al. 2015;
Bedrosian et al. 2018) suggesting connectivity to a deeper magma source. In
those cases, the upper regions of the conductor would represent fluids exsolved
from ascending hydrous magmas (\pm precipitated sulfides), creating a vertical,
conductive continuum from regions of hydrous partial melt to accumulated
fluids, as shown schematically in Figure 1. MT surveys alone would not be able to
resolve this transition.

**Copper Accumulation in Sub-volcanic Brine Lenses**

To assess the potential for metal accumulation in sub-volcanic brine lenses
we have incorporated copper transport into the hydrodynamic simulations of
Afanaysev et al. (2018) by including partitioning of Cu between vapour and
liquid of differing salinity, and the solubility of chalcopyrite in such fluids. In light
of the importance of S^{2-} ions for copper ore mineral precipitation our new model
considers the fate of both sulfur-sufficient and sulfur-free, Cu-bearing magmatic
fluids as they discharge through a high-permeability and porosity conduit,

undergo phase separation and cool. We note that Weis et al. (2012) also
incorporated Cu solubility and partitioning into their models, concluding that Cu
accumulates below a halite cap, within a region of elevated salinity, similar to the
hypersaline liquid lens in Fig. 6a.

Methods. We augment the hydrodynamic model equations of Afanasyev et al.
(2018) with the following conservation equation for modelling Cu transport

$$\frac{\partial}{\partial t} \left(\phi \sum_{i=g,l} \rho_i c_i s_i + (\phi_0 - \phi) \rho_{Cu} \right) + \text{div} \left(\sum_{i=g,l} \rho_i c_i \mathbf{w}_i \right) = 0 \quad (1)$$

where ϕ is the porosity, ϕ_0 is the initial porosity prior to Cu precipitation, ρ is the
density, c is the Cu mass fraction, s is the saturation, \mathbf{w} is the Darcy's velocity, and
subscripts g , l and Cu refer to parameters of the gas, liquid and chalcopyrite
(solid) phases respectively. In using Eq. (1) we assume that, providing reduced
sulfur is available, Cu forms a separate solid phase (e.g. chalcopyrite) that
precipitates independently of halite. The copper sulfide mineral density ρ_{Cu} is set
to 4100 kg/m³.

The total concentration of Cu, c_t , in gas and liquid phases is given by

$$c_t = \sum_{i=g,l} \rho_i c_i s_i / \sum_{i=g,l} \rho_i s_i \quad (2)$$

Copper partitioning between vapour and liquid phases is calculated in line
with the experimental data in Fig.15 of Kouzmanov and Pokrovski (2012)

$$\log \frac{m_g}{m_l} = 3.866 \log \frac{\rho_g}{\rho_l} \quad (3)$$

where m is the number of Cu moles per 1 kg of fluid in the corresponding phase.

This expression is similar to that of Tattitch and Blundy (2017) wherein copper

partitioning scales with the salinity ratio of liquid and vapour.

To obtain the equilibrium solubility of Cu, c_{eq} , we interpolate the solubility

data in Fig.17e of Kouzmanov and Pokrovski (2012) to yield an empirical

relationship for $c_{eq}(T,x)$ as a function of temperature and fluid salinity x :

$$586 \quad \log(c_{eq}) = a_{00} + a_{10} \ln(x) + a_{01}T + a_{20} [\ln(x)]^2 + a_{11} \ln(x)T + a_{02}T^2 \quad (4)$$

where T is in °C, x is weight fraction NaCl_{eq} in the liquid, $a_{00}=-2.616$, $a_{10}=1.741$,

$a_{01}=0.02567$, $a_{20}=0.07463$, $a_{11}=-2.117 \times 10^{-5}$, $a_{02}=-0.00153$. The resulting

expression, for three representative salinities, is shown in Fig. 4. Copper is

completely dissolved in hypersaline liquid if $c_t < c_{eq}(T,x)$ and chalcopyrite

precipitates when $c_t \geq c_{eq}(T,x)$. Further experiments are required to refine these

solubility relationships at high temperature (>400 °C) and salinity; Eq. (4) is

simply a useful starting point.

Calculations assume that the pH of the fluid is 5, consistent with rock-

buffering via hydrolysis reactions involving K-feldspar and muscovite. For a one

molal NaCl solution (5.8 wt% NaCl_{eq}) at 225°C these reactions define $\text{pH} = 5 \pm 0.5$

(Yardley 2005). Hypersaline liquids recovered at depth at 500°C from Kakkonda,

have pH in the range 3-5 (Kasai et al. 1998a,b). If pH deviates from 5 then

chalcopyrite solubility will be modified. For a unit decrease in pH below 5, all

other parameters constant, solubility increases by an order of magnitude; for a

unit increase in pH solubility decreases by the same factor (Kouzmanov and

Pokrovski 2012). Modelled pH evolution of the fluids in response to rock

reaction would be a future development.

Numerical simulation of Cu transport is done using the method of splitting
by physical processes. Each time step consists of two separate steps A and B. In
Step A transport of the primary NaCl-H₂O fluid is simulated for a given porosity ϕ
and permeability, and new distributions of P , T , ρ_i , s_i , w_i are evaluated (see
Afanasyev et al. 2018). In Step B, these distributions are frozen and Cu transport
is calculated according to Eqs. (1)-(4). At the end of Step B if c_t in a grid block is
higher than the equilibrium value (4) then the fraction $c_t - c_{eq}$ of Cu is precipitated
and thereby c_t is reduced to c_{eq} . Based on the amount of precipitated Cu the
porosity ϕ and the permeability distributions are modified. The calculation then
continues with Step A at the next time step.

Calculations were run with the same initial conditions as the reference
model of Afanasyev et al. (2018) in terms of fluid flux, permeability, porosity etc
(Fig. 6a). The input fluid contains 700 ppm Cu, a conservative value based on ID
fluid inclusions from Bingham Canyon (Landtwing et al. 2005; Seo et al. 2012)
and comparable to the 500 ppm Cu used by Weis et al. (2012). As noted above,
this value exceeds that in the White Island fumarole gas (30 ppm), which has
been lowered by sub-surface reactions and Cu mineral precipitation. Sulfide
content of the input fluid is deemed either: (i) sufficient to precipitate
chalcopyrite when its solubility is attained according to Eq. (4); or (ii) zero, in
which case no Cu precipitation can occur. Our results therefore represent end-
member cases where Cu precipitation is (i) maximised or (ii) minimised at the
expense of Cu retained in solution. We do not consider explicitly sulphide
formation via the Ca-mediated disproportionation of SO₂ proposed by
Mavrogenes & Blundy (2017).

Results. Model results for the reference scenario are presented in terms of bulk
fluid salinity (Fig. 6a), Cu dissolved in hypersaline liquid for no-sulfur (Fig. 6b)
and unlimited-sulfur (Fig. 6c) cases, Cu contained in precipitated solids
(unlimited-sulfur case only, Fig. 6d), and the temperature field (Fig. 6e). In
panels (b) to (e) the region containing hypersaline liquids with ≥ 5000 ppm Cu is
outlined. Results are shown at 10,000 years after the onset of fluid discharge
(Fig. 6a). For both precipitated and dissolved Cu, concentrations are highest in an
annulus centred on the conduit at 1.5 to 2 km depth. The shape and depth of
precipitated Cu annulus is similar to that of Weis et al (2012), despite their use of
a different chalcopyrite solubility expression and modelling approach
(Supplementary Figure 1)

The precipitated Cu annulus (Fig. 6d) is more restricted in extent than the
Cu-rich liquid annulus (Fig. 6b,c), reflecting higher temperatures near the
conduit. The highest bulk contents of precipitated Cu, ~ 20 kg/m³ of rock, are
contained within an inclined region located at 1.5 km depth 700 m from the
centre of the conduit (Fig. 6d). Very little Cu is precipitated within the conduit
itself. These bulk Cu contents equate to maximum potential ore grades of 0.75
646 wt% Cu for rock with 2650 kg/m³ density. This grade is comparable to many
economic PCD mines. Precipitated Cu contents decrease rapidly away from this
region reflecting temperature decrease. The mass of solid Cu (as chalcopyrite)
increases almost linearly with degassing duration, as the liquid lens becomes
saturated and more and more chalcopyrite precipitates. Thus ores grades of ~ 1.5
651 wt% Cu would require degassing for ≥ 20 kyr at the reference flux.

The Cu-in-liquid annulus (Fig. 6b,c) is slightly broader and deeper than for
precipitated Cu, extending from the axis of symmetry out to 1 km radial distance.

The no-sulfur annulus (Fig. 6b) is slightly broader than the unlimited-sulfur
annulus (Fig. 6c). The highest hypersaline liquid Cu contents (≥ 7000 ppm) occur
at the outermost periphery of the annulus where temperature exceeds $400\text{ }^{\circ}\text{C}$
(Fig. 6e) for both the no-sulfur and unlimited-sulfur cases. These modelled Cu
contents are consistent with high-salinity fluid inclusion analyses from PCDs (e.g.
Landtwing et al. 2005; Seo et al. 2012; Kouzmanov and Pokrovski 2012). Note
that, for the unlimited-sulfur case, contours of Cu-in-liquid concentration do not
map simply onto isotherms, because salinity (in addition to temperature) also
controls Cu solubility (Fig. 4). In the no-sulfur case, the highest Cu contents in the
hypersaline liquid lens are controlled not by chalcopyrite solubility, but by the
dynamics of phase separation and partitioning. They are therefore more
sensitive to the model parameters, such as input fluid Cu content, fluid flux and
porosity-permeability characteristics of the conduit region.

Total Cu input to the system in 10,000 years is 21 Mt for the modelled fluid
flux and Cu content. For the unlimited-sulfur case, the total mass of Cu retained
in the modelled domain includes both precipitated and dissolved forms. For a
reference volume with radius 1.5 km extending from the surface to 3 km depth
the mass of precipitated Cu is 20 Mt after 10,000 years (Fig. 7). The mass of Cu
dissolved in hypersaline liquid at this time (~ 1 Mt) is less by a factor of 20. Thus,
almost no copper is lost to the atmosphere via the low-salinity exhaust vapour,
and very little is stored at low concentration outside the reference volume. For
the no-sulfur case, there is no precipitated Cu, but the total mass dissolved (~ 3.5
676 Mt after 10,000 years; Fig. 7) is less than the total Cu retained in the unlimited-
677 sulfur case, due to significant Cu loss to the atmosphere, at a time-averaged flux
of 4800 kg/day. For comparison, copper fluxes to the atmosphere of 200 to

20,000 kg/day have been recorded at several subduction-related basalt
volcanoes (Edmonds et al. 2018); at White Island andesite volcano the copper
flux is ~300 kg/day (Hedenquist et al. 1993). The range in Cu flux between
volcanoes likely reflects variation in the initial fluid content of Cu and reduced
sulfur, as well as the efficiency of Cu sequestering by phase separation, sulfide
precipitation, and mixing with meteoric water beneath the volcanic edifice. Our
no-sulfur and unlimited sulfur cases likely provide bounds on the situation
beneath active volcanoes.

The annular form of the modelled high-Cu regions, both precipitated and
dissolved (Fig. 6c-e), is reminiscent of the 3-D form and dimension of ore shells
in several PCDs, e.g. Bajo de la Alumbrera, Argentina (Buret et al. 2016), Bingham
Canyon, USA (Redmond et al. 2004), Batu Hijau, Indonesia (Arif and Baker 2004).
This is consistent with the notion that hypersaline liquid accumulation is related
to ore formation, as previously suggested by Weis et al. (2012), Weis (2015) and
Blundy et al., (2015). The size and copper endowment of the modelled high-Cu
regions for the unlimited sulfur case (~20 Mt) are broadly similar to these
world-class PCDs, whereas the endowment for the no-sulfur case is substantially
smaller. This comparison emphasises the importance of the availability of sulfur
(as well as Cu) in generating PCDs. A central question in PCD research is the
extent to which a single magmatic fluid can provide both of these ingredients
and, if so, under what circumstances. Needless to say, a sulfur-poor copper-rich
brine lens will not become a large PCD without the subsequent addition of sulfur,
as noted by Blundy et al. (2015). It is well-established in volcano studies that the
mass of sulfur discharged from active volcanoes frequently exceeds that
dissolved in erupted magma (Wallace and Edmonds, 2011). One explanation for

[revised manuscript text omitted]

of the latest generation of geothermal targets. For example, the Iceland Deep
Drilling Project geothermal well IDDP-2 at Reykjanes penetrated 426 °C volcanic
rock at depths of 4.5 km (Friðleifsson et al. 2017), the Venelle-2 well at
Lardarello, Italy, reached 504°C at 2.8 km (Bertani et al. 2018), and the Kakkonda
WD-1 well, Japan, reached 500 °C at 3.7 km (Saito et al. 1997; Ikeuchi et al. 1998,
Muraoka et al. 1998). There is emerging interest in drilling even deeper, hotter
systems to extract power from supercritical fluids associated with young
magmatic systems (Elders et al. 2014; Scott et al., 2015; Tsuchiya et al. 2016;
Agostinetti et al. 2017; Dobson et al. 2017). For example, the Japan Beyond
Brittle Project at Naruko volcano will target 350-500 °C fluids at 3-5 km depth
(Muraoka et al. 2014; Watanabe et al. 2017).

The potential of hot hypersaline liquids as a source of metals (e.g. Mn, Zn,
Pb) and silica has long been recognised (Kimura 1953; Kennedy 1961; White et
al. 1963; Skinner et al. 1967; Ellis 1968; Werner 1970; Barnea 1979), but their
attempted exploitation has a chequered history (e.g. Maimoni 1982; Duyvesteyn
1992; Gallup 1998; Clutter 2000; Bourcier et al. 2005). Demand for lithium has
led to a recent resurgence of interest in metal recovery from high-salinity
geothermal fluids (Harrison 2014).

Provided that deep wells into hot rock can be completed successfully, the
difficulty of recovering Cu-bearing fluids faces two key problems. The first is the
relatively low contents of Cu (and other metals) in conventional geothermal
fluids. Indeed, Neupane and Wendt (2017), in their assessment of US geothermal
mineral resources, state “the presence of low mineral content in the brine could
be prohibitive for extraction”. The available data for geothermal fluids (e.g.
Yardley and Bodnar 2013) support this statement. However, almost all

geothermal fluids analysed so far are mixtures of magmatic and meteoric fluids
with quite low salinities (average ~1 wt% NaCl_{eq}; Stimac et al., 2015) recovered
at temperatures below 360 °C, where solubility of copper minerals is low (Fig. 4).
The lack of suitably hot, solute-rich, hypersaline magmatic liquids encountered
in geothermal wells, with the exception of the Kakkonda example, are largely a
consequence of intentionally avoiding such regions when drilling for geothermal
resources. Recovering metal-rich hypersaline liquids requires tapping brine
lenses at significantly higher temperatures than is conventional in geothermal
fields.

A second challenge is scaling and corrosion of the wellbore in response to
decompression and cooling of solute-rich, acidic fluids (e.g. Reyes et al. 2002). It
is for this reason that most geothermal production wells avoid solute-rich liquid
lenses in favour of low-salinity, high-enthalpy vapour or supercritical fluids
(Agostinetti et al. 2017). Scales recovered from a number of geothermal wells are
extremely Cu-rich, due to precipitation of sulfides (and tellurides). For example,
at Kakkonda, variable temperature histories in both the hypersaline liquid
reservoir and during the sampling process itself resulted in Cu-Zn-Pb-Fe rich
sulfide/oxide scales with highly variable chalcophile metal ratios (Uchida et al.
1998) and Cu contents up to 14 wt% (Yanasigawa et al. (2000). Nonetheless, the
record of deposition of metals in the scales, combined with a transition from Pb-
rich to Zn-rich to Cu-rich scales with increasing temperature, is consistent with a
high temperature (>500 °C) hypersaline reservoir at Kakkonda that, at depth,
may contain dissolved metals at significant concentrations. The style of scaling
with depth is a consequence of the strong temperature dependence of mineral
solubility, and the propensity for scaling will be exacerbated when extracted

fluids are even hotter and more solute-rich. There have been a number of
attempts to develop fluid extraction methods to minimise scaling by modifying
the chemistry and temperature of the extracted fluid (e.g. Harrar et al. 1979;
Gallup, 1992; Baba et al. 2015). Reducing scaling of silica, an important
additional component of geothermal fluids has been a particular priority (e.g.
Patterson, 2009; Gallup et al., 2003; Premuzic et al., 1997). Further development
of such methods will lead to more efficient recovery of metals from deep,
hypersaline fluid reservoirs.

Finally, drilling into hot rock and extracting metal-bearing fluids represents
a major technical challenge because of uncertainties in the permeability and
porosity structure of the reservoir. This is partly because there has been very
little drilling into deep portions of geothermal systems with accumulations of
high-temperature magmatic fluids. Drill-core samples recovered from such wells
will tend to underestimate the in-situ porosity and permeability as cooling and
reduction in fluid pressure cause mineral precipitation and closing of fractures
around the well-bore (cf. Fig. 5). Also, where hypersaline magmatic fluids
accumulate there may be dynamic permeability variations with spatial and
temporal changes in brittle versus ductile host rock behaviour (e.g. Weis, 2015).
Even data on matrix porosity and permeability from traditional geothermal
reservoirs are often proprietary or not well documented. However, it is
important to understand sub-volcanic porosity-permeability relationships
because they control how much metal-bearing hypersaline fluid can be stored,
and the extent to which they will flow into the borehole with and without
additional reservoir stimulation (e.g. Eggertsson et al, 2020). Porosity-
permeability relationships also influence the response of the reservoir to fluid

[revised manuscript text omitted]

Hydrodynamic calculations show that Cu concentrations in dynamic,
hypersaline liquid lenses beneath active or dormant volcanoes may be as high as
7000 ppm at temperatures over 400 °C. For our reference model, after 10,000
885 years of degassing the hypersaline liquid lens contains 1.0 to 1.4 Mt of dissolved
Cu, depending on the availability of reduced sulfur. These expectations are
consistent with observations of high Cu content in fluid inclusions from natural
samples. Such hot, metal-rich liquids would need to be extracted efficiently
without significant reservoir clogging or well-bore scaling. The extent to which
such a resource could be exploited economically depends on the technological
challenges and costs of drilling into hot reservoir rocks and recovering
hypersaline liquid. If Cu can be extracted in solution form, Cu processing costs
may be greatly reduced. Similarly, the co-production of geothermal energy to
power fluid extraction and processing would confer additional economic and
environmental benefit (e.g.. Bourcier et al., 2005). We have focussed only on Cu;
the presence of other metals in magmatic fluids (including Li, Zn, Pb, Au and Ag;
Edmonds et al. 2018) also allows for significant co-recovery. We propose, on the
basis of recent developments in our understanding of crustal magmatic systems,
that mining of sub-volcanic metal-bearing brines may be a fruitful avenue for
addressing the impending growth in demand for certain metal resources that
will arise during the transition to a low-carbon economy.

**References**

- Afanasyev A, Blundy J, Melnik O, Sparks S (2018) Formation of magmatic brine
lenses via focussed fluid-flow beneath volcanoes. *Earth Planet Sci Lett*
486:119-128.
- Agostinetti NP, Licciardi A, Piccinini D, Mazzarini F, Musumeci G, Saccorotti G,
Chiarabba C (2017) Discovering geothermal supercritical fluids: a new
frontier for seismic exploration. *Sci Rep* 7:14592.
- Aiuppa, A., Baker, D.R. and Webster, J.D., 2009. Halogens in volcanic systems.
*Chemical Geology*, 263(1-4), pp.1-18.
- Aizawa K, Yoshimura R, Oshiman N, Yamazaki K, Uto T, Ogawa Y, Tank SB, Kanda
914 W, Sakanaka S, Furukawa Y, Hashimoto T (2005) Hydrothermal system
beneath Mt. Fuji volcano inferred from magnetotellurics and electric self-
potential. *Earth Planet Sci Lett* 235:343-355.
- Annen C. (2009) From plutons to magma chambers: Thermal constraints on the
accumulation of eruptible silicic magma in the upper crust. *Earth and*
*Planetary Science Letters*. 284(3-4):409-16.
- Annen, C., Blundy, J.D., Sparks, R.S.J., 2006. The genesis of intermediate and silicic
magmas in deep crustal hot zones. *Journal of Petrology*, 47, 505-539
- Annen, C., Blundy, J.D., Leuthold, J. and Sparks, R.S.J., 2015. Construction and
evolution of igneous bodies: Towards an integrated perspective of crustal
magmatism. *Lithos*, 230, 206-221.
- Arif J, Baker T (2004) Gold paragenesis and chemistry at Batu Hijau, Indoneisa:
implications for gold-rich porphyry copper deposits. *Mineral Deposita*,
39:523-535.

Audétat A, Pettke T, Heinrich CA, Bodnar RJ (2008) The composition of
magmatic-hydrothermal fluids in barren and mineralized intrusions. *Econ*
*Geol* 103:877-908.
Baba A, Demir MM, Koç GA, Tuğcu C (2015) Hydrogeological properties of hyper-
saline geothermal brine and application of inhibiting siliceous scale via pH
modification. *Geothermics* 53:406-412.
Barnea J (1979) Geothermal minerals - the neglected minerals. *Geotherm Energy*
*Mag*, 7:12.
Bedrosian PA, Peacock JR, Bowles-Martinez E, Schultz A, Hill GJ (2018) Crustal
inheritance and a top-down control on arc magmatism at Mount St. Helens.
*Nat Geosci* 11:865-870.
Bertani R, Büsing H, Buske S, Dini A, Hjelstuen M, Luchini M, Manzella A, Nybo R,
Rabbel W, Serniotti L (2018) The first results of the DESCRAMBLE project.
*Proc, 43rd Workshop on Geothermal Reservoir Engineering, Stanford*
*University, Stanford, California, February 12-14, 2018, SGP-TR-213*
Bertrand EA, Caldwell TG, Bannister S, Soengkono S, Bennie SL, Hill GJ Heise W,
2015. Using array MT data to image the crustal resistivity structure of the
southeastern Taupo Volcanic Zone, New Zealand. *J Volcanol Geotherm Res*,
305:63-75.
Biggs, J., Ebmeier, S.K., Aspinall, W.P., Lu, Z., Pritchard, M.E., Sparks, R.S.J. and

[revised manuscript text omitted]

1093 Hooft, E.E., Nomikou, P., Toomey, D.R., Lampridou, D., Getz, C., Christopoulou,
1094 M.E., O'Hara, D., Arnoux, G.M., Bodmer, M., Gray, M. and Heath, B.A., 2017.
Backarc tectonism, volcanism, and mass wasting shape seafloor morphology

in the Santorini-Christiana-Amorgos region of the Hellenic Volcanic Arc.
*Tectonophysics*, 712, pp.396-414.
Huber C, Townsend M, Degruyter W, Bachmann O (2019) Optimal depth of
subvolcanic magma chamber growth controlled by volatiles and crust
rheology. *Nature Geosci* 12:762-768.
Ikeda R, Kajiwarara T, Omura K, Hickman S (2008) Physical rock properties in and
around a conduit zone by well-logging in the Unzen Scientific Drilling Project,
Japan. *J Volcanol Geotherm Res* 175:13-19.
Kasai K, Sakagawa Y, Komatsu R, Sasaki M, Akaku K, Uchida T (1998a) The origin
of hypersaline liquid in the Quaternary Kakkonda granite, sampled from well
WD-1a, Kakkonda geothermal system, Japan. *Geothermics* 27:631-645.
Kasai K, Sakagawa Y, Miyazaki S, Akaku K, Uchida T (1998b) Supersaline and
metal-rich brine obtained from the Quaternary Kakkonda Granite by NEDO
WD-1a in the Kakkonda geothermal field, Japan. *Mineral Deposita* 33:298-301.
Kennedy AM (1961) The recovery of lithium and other minerals from
geothermal water at Wairakei. In: *Proc UN Conference on New Sources of*
*Energy*, Rome, Italy (pp. 21-31).
Kiddle EJ, Edwards BR, Loughlin SC, Petterson M, Sparks RSJ, Voight B (2010)
Crustal structure beneath Montserrat, Lesser Antilles, constrained by
xenoliths, seismic velocity structure and petrology. *Geophys Res Lett* 37(19).
Kimura K (1953) On the utilization of hot springs in Japan. *Proc Seventh Pacific*
*Science Congress*, New Zealand, 2:500.
Kiser, E., Palomeras, I., Levander, A., Zelt, C., Harder, S., Schmandt, B., Hansen, S.,
Creager, K. and Ulberg, C., 2016. Magma reservoirs from the upper crust to the

Moho inferred from high-resolution Vp and Vs models beneath Mount St.
Helens, Washington State, USA. *Geology*, 44(6), pp.411-414.
Klemm LM, Pettke T, Heinrich CA (2008) Fluid and source magma evolution of
the Questa porphyry Mo deposit, New Mexico, USA. *Mineral Deposita* 43:533.
Koděra P, Heinrich CA, Wälle M, Lexa J (2014) Magmatic salt melt and vapor:
Extreme fluids forming porphyry gold deposits in shallow subvolcanic
settings. *Geology*, 42:495-498.
Komori S, Kagiya T, Utsugi M, Inoue H, Azuhata I (2013) Two-dimensional
resistivity structure of Unzen Volcano revealed by AMT and MT surveys. *Earth*
*Planets Space* 65:759-766.
Kouzmanov K, Pokrovski GS (2012) Hydrothermal controls on metal distribution
in porphyry Cu (-Mo-Au) systems. *Econ Geol Spec Publ* 16:573-618
Landtwing MR, Pettke T, Halter WE, Heinrich CA, Redmond PB, Einaudi MT,
Kunze K (2005) Copper deposition during quartz dissolution by cooling
magmatic-hydrothermal fluids: the Bingham porphyry. *Earth Planet Sci Lett*
235:229-243.
Lecumberri-Sanchez P, Steele-MacInnis M, Weis P, Driesner T, Bodnar RJ (2015)
Salt precipitation in magmatic-hydrothermal systems associated with upper
crustal plutons. *Geology*, 43:1063-1066.
Lesne, P., Kohn, S., Blundy, J., Witham, F., Botcharnikov, R., Behrens, H., 2011.
Experimental simulation of closed-system degassing in the system basalt-H₂O-
CO₂-S-Cl. *Journal of Petrology* 52: 1737-1762
Loucks RR (2014) Distinctive composition of copper-ore-forming arc magmas.
*Austral J Earth Sci* 61:5-16.

Lowenstern, J.B., Sisson, T.W. and Hurwitz, S., 2017. Probing magma reservoirs
to improve volcano forecasts. *Eos*, 98.
Magee, C., Stevenson, C.T., Ebmeier, S.K., Keir, D., Hammond, J.O., Gottsmann, J.H.,
Whaler, K.A., Schofield, N., Jackson, C.A., Petronis, M.S. and O'Driscoll, B., 2018.
Magma plumbing systems: a geophysical perspective. *Journal of Petrology*,
59(6), pp.1217-1251.
Maimoni A (1982) Minerals recovery from Salton Sea geothermal brines: a
literature review and proposed cementation process. *Geothermics* 11:239-
258.
Mavrogenes, J., Blundy, J., 2017. Crustal sequestration of magmatic sulfur dioxide.
*Geology* **45**, 211-214
McGuinness, M., White, S., Young, R., Ishizaki, H., Ikeuchi, K. and Yoshida, Y., 1995.
A model of the Kakkonda geothermal reservoir. *Geothermics*, 24:1-48.
Moore JN, Gunderson RP (1995) Fluid inclusion and isotopic systematics of an
evolving magmatic-hydrothermal system. *Geochim Cosmochim Acta* 59:3887-
3907.
Muramatsu Y, Komatsu R (1999) Microthermometric evidence for the formation
of Ca-rich hypersaline brine and CO₂-rich fluid in the Mori Geothermal
Reservoir, Japan. *Resource Geology* 49:27-37.
Muraoka H, Asanuma H, Tsuchiya N, Ito T, Mogi T, Ito H (2014) The Japan
Beyond-Brittle Project. *Scientific Drilling* 17:51-59.
Muraoka H, Uchida T, Sasada M (1998) Deep geothermal resources survey
program: igneous, metamorphic and hydrothermal processes in a well
encountering 500 °C at 3729 m depth, Kakkonda, Japan. *Geothermics* 27:507-
534.

Neupane G, Wendt DS (2017). Assessment of mineral resources in geothermal
brines in the US. In: Proc 42nd Workshop on Geothermal Reservoir
Engineering, Stanford University, Stanford, CA, USA pp 13-15.
Nitschke F, Held S, Himmelsbach T, Kohl T (2017) THC simulation of halite
scaling in deep geothermal single well production. *Geothermics* 65:234-243.
Northey, S., Haque, N. and Mudd, G., 2013. Using sustainability reporting to
assess the environmental footprint of copper mining. *Journal of Cleaner*
*Production*, 40, pp.118-128.
Nurhasan, Ogawa Y, Ujihara N et al (2006) Two electrical conductors beneath
Kusatsu-Shirane volcano, Japan, imaged by audiomagnetotellurics, and their
implications for the hydrothermal system. *Earth Planets Space* 58:1053-1059.
Palacios, J-L., Abadias, A., Valero, A., Valero, A. and Reuter, M., 2019. The energy
needed to concentrate minerals from common rocks: The case of copper ore.
*Energy*, 181, pp.494-503.
Patterson, M.C., 2006. Geothermal brines- high value mineral extraction. In *Sohn*
*International Symposium; Advanced Processing of Metals and Materials Volume*
*6: New, Improved and Existing Technologies: Aqueous and Electrochemical*
*Processing* 6, 579-588).
Premuzic, E.T., Lin, M.S., Jin, J.Z. and Hamilton, K., 1997. Geothermal waste
treatment biotechnology. *Energy sources* 19, 9-17.
Pritchard, M.E. and Simons, M., 2004. An InSAR-based survey of volcanic
deformation in the central Andes. *Geochemistry, Geophysics, Geosystems*, 5(2).
Pritchard, M.E., de Silva, S.L., Michelfelder, G., Zandt, G., McNutt, S.R., Gottsmann,
1192 J., West M.E., Blundy, J., Christensen, D.H., Finnegan, N.J., Minaya, E., Sparks,
R.S.J., Sunagua, M., Unsworth, M.J., Alvizuri, C., Comeau, M.J., del Potro, R., Diez,

1194 M., Farrell, A., Henderson, S.T., Jay, J.A., Lopez, T., Legrand, D., Naranjo, J.A.,
McFarlin, H., Muir, D., Perkins, J.P., Spica, Z., Wilder, A., Ward, K.M., 2018.
Synthesis: PLUTONS: Investigating the relationship between pluton growth
and volcanism in the Central Andes. *Geosphere* **14**, 3
Redmond PB, Einaudi MT, Inan EE, Landtwing MR, Heinrich CA (2004) Copper
deposition by fluid cooling in intrusion-centered systems: New insights from
the Bingham porphyry ore deposit, Utah. *Geology*, 32:217-220.
Reyes AG, Trompeter WJ, Britten K, Searle J (2002) Mineral deposits in the
Rotokawa geothermal pipelines, New Zealand. *J Volcanol Geotherm Res*
119:215-239.
Richards JP (2015) The oxidation state, and sulfur and Cu contents of arc
magmas: implications for metallogeny. *Lithos*, 233:27-45.
Ryan, G.A., Peacock, J.R., Shalev, E. and Rugis, J., 2013. Montserrat geothermal
system: A 3D conceptual model. *Geophysical Research Letters*, 40(10),
pp.2038-2043.
Saito S, Sakuma S, Uchida T (1997) Frontier geothermal drilling operations
succeed at 500 °C BHST. In: SPE/IADC drilling conference. Society of
Petroleum Engineers.
Sakagawa, Y., Aoyama, K., Ikeuchi, K., Takahashi, M., Kato, O., Doi, N., Tosha, T.,
Ominato, T. and Koide, K., 2000, May. Natural state simulation of the
Kakkonda geothermal field, Japan. In *Proceedings of World Geothermal*
*Congress, Kyushu-Tohoku, Japan* (pp. 2839-2844).
Sasaki M, Fujimoto K, Sawaki T, Tsukamoto H, Muraoka H, Sasada M, Ohani T,
Yagi M, Kurosawa M, Doi N, Kato O, Kasai K, Komatsu R, Muramatsu Y (1998)
Characterization of a magmatic/meteoric transition zone at the Kakkonda

geothermal system, northeast Japan. *Water-Rock Interaction*, Arehart &
Hulston (Eds), p. 483-486
Sasaki, M., Fujimoto, K., Tsukamoto, H., Sawaki, T., Sasada, M., Kurosawa, M.,
Yagi, M., Muramatsu, Y., Kato, O., Komatsu, R. and Kasai, K., 2003. Geochemical
features of vein anhydrite from the Kakkonda geothermal system, Northeast
Japan. *Resource Geology*, 53(2), pp.127-142.
Schipper, B.W., Lin, H.C., Meloni, M.A., Wansleeben, K., Heijungs, R. and van der
Voet, E., 2018. Estimating global copper demand until 2100 with regression
and stock dynamics. *Resources, Conservation and Recycling*, 132, pp.28-36.
Schöpa, A., Annen, C., Dilles, J.H., Sparks, R.S.J., Blundy, J.D., 2017. Magma
Emplacement Rates and Porphyry Copper Deposits: Thermal Modeling of the
Yerington Batholith, Nevada. *Economic Geology* **112**, 1653-1672
Scott S, Driesner T, Weis P (2015) Geologic controls on supercritical geothermal
resources above magmatic intrusions. *Nat Comm* 6:7837.
Seo JH, Guillong M, Heinrich CA (2012) Separation of molybdenum and copper in
porphyry deposits: The roles of sulfur, redox, and pH in ore mineral
deposition at Bingham Canyon. *Econ Geol* 107:333-356.
Shinohara, H., 2008. Excess degassing from volcanoes and its role on eruptive
and intrusive activity. *Reviews of Geophysics*, 46(4).
Sillitoe RH (2010) Porphyry copper systems. *Econ Geol* 105:3-41.
Simmons, S.F. and Brown, K.L., 2007. The flux of gold and related metals through
a volcanic arc, Taupo Volcanic Zone, New Zealand. *Geology*, 35(12), pp.1099-
1102.

Simon AC, Pettke T, Candela PA, Piccoli PM, Heinrich CA (2006) Copper
partitioning in a melt–vapor–brine–magnetite–pyrrhotite assemblage.
*Geochim Cosmochim Acta* 70:5583-5600.
Singer, B.S., Andersen, N.L., Le Mével, H., Feigl, K.L., DeMets, C., Tikoff, B., Thurber,
C.H., Jicha, B.R., Cardona, C., Córdova, L. and Gil, F., 2014. Dynamics of a large,
restless, rhyolitic magma system at Laguna del Maule, southern Andes, Chile.
*GSA today*, 24(12), pp.4-10.
Skinner BJ, White DE, Rose HJ, Mays RE (1967) Sulfides associated with the
Salton Sea geothermal brine. *Econ Geol* 62:316-330.
Sparks RSJ, Annen C, Blundy JD, Cashman KV, Rust AC, Jackson MD (2019)
Formation and dynamics of magma reservoirs *Philosophical Transactions of*
*the Royal Society A* **377**: 20180019
Srigutomo W, Kagiya T, Kanda W, Munekane H, Hashimoto T, Tanaka Y, Utada
H, Utsugi M (2008) Resistivity structure of Unzen Volcano derived from time
domain electromagnetic (TDEM) survey. *J Volcanol Geotherm Res* 175:231–
240
Stasiuk MV, Barclay J, Carroll MR, Jaupart C, Ratté JC, Sparks RSJ, Tait SR (1996)
Degassing during magma ascent in the Mule Creek vent (USA). *Bull Volcanol*
58:117-130.
Stimac J, Goff F, Goff CJ, 2015. Intrusion-Related Geothermal Systems. In:
Sigurdsson H et al. (Eds.), *The Encyclopedia of Volcanoes*, pp. 799–822.
Tapster, S., Condon, D.J., Naden, J., Noble, S.R., Petterson, M.G., Roberts, N.M.W.,
Saunders, A.D. and Smith, D.J., 2016. Rapid thermal rejuvenation of high-
crystallinity magma linked to porphyry copper deposit formation; evidence

from the Koloula Porphyry Prospect, Solomon Islands. *Earth and Planetary*
*Science Letters*, 442, pp.206-217.
Taran, Y.A., Hedenquist, J.W., Korzhinsky, M.A., Tkachenko, S.I. and Shmulovich,
1269 K.I., 1995. Geochemistry of magmatic gases from Kudryavy volcano, Iturup,
Kuril Islands. *Geochimica et Cosmochimica Acta*, 59(9), pp.1749-1761.
Tattitch BC, Blundy JD (2017) Cu-Mo partitioning between felsic melts and
saline-aqueous fluids as a function of $X_{\text{NaCl}_{\text{eq}}}$, f_{O_2} , and f_{S_2} . *Amer Mineral*
102:1987-2006.
Triahadini A, Aizawa K, Teguri Y, Koyama T, Tsukamoto K, Muramatsu D, Chiba
1275 K, Uyeshima M (2019) Magnetotelluric transect of Unzen graben, Japan:
conductors associated with normal faults. *Earth Planets Space* 71:28.
Tsuchiya N, Yamada R, Uno M (2016) Supercritical geothermal reservoir
revealed by a granite–porphyry system. *Geothermics*, 63:182-194.
Uchida T, Akaku, K, Yanagisawa N, Kamenesonono H, Sasaki M, Miyazaki S, Doi U
(1998) Deep Geothermal Resources Survey Project in the Kakkonda
Geothermal Field. *Energy Sources* 20:763-77
Ulberg, C.W., Creager, K.C., Moran, S.C., Abers, G.A., Thelen, W.A., Levander, A.,
Kiser, E., Schmandt, B., Hansen, S.M. and Crosson, R.S., 2020. Local source Vp
and Vs tomography in the Mount St. Helens region with the iMUSH broadband
array. *Geochemistry, Geophysics, Geosystems*, 21(3), 2019GC008888.
Ulrich T, Gunther D, Heinrich CA (1999) Gold concentrations of magmatic brines
and the metal budget of porphyry copper deposits, *Nature* 399:676-679
Wallace, P.J. and Edmonds, M., 2011. The sulfur budget in magmas: evidence
from melt inclusions, submarine glasses, and volcanic gas emissions. *Reviews*
*in Mineralogy and Geochemistry*, 73(1), pp.215-246.

Watanabe N, Numakura T, Sakaguchi K, Saishu H, Okamoto A, Ingebritsen SE,
Tsuchiya N (2017) Potentially exploitable supercritical geothermal resources
in the ductile crust. *Nat Geosci* 10:140.
Webster JD, Vetere F, Botcharnikov RE, Goldoff B, McBirney A, Doherty AL
(2015) Experimental and modeled chlorine solubilities in aluminosilicate
melts at 1 to 7000 bars and 700 to 1250 °C: Applications to magmas of
Augustine Volcano, Alaska. *Amer Mineral*, 100:522-535.
Weis P (2015) The dynamic interplay between saline fluid flow and rock
permeability in magmatic-hydrothermal systems. *Geofluids* 15:350-371.
Weis P, Driesner T, Heinrich CA (2012) Porphyry-copper ore shells form at
stable pressure-temperature fronts within dynamic fluid plumes. *Science*,
338:1613-1616.
Werner HH (1970) Contribution to the mineral extraction from supersaturated
geothermal brines Salton Sea Area, California. *Geothermics* 2:1651-1655.
White DE, Anderson ET, Grubbs DK (1963) Geothermal brine well: mile-deep
drill hole may tap ore-bearing magmatic water and rocks undergoing
metamorphism. *Science* 139:919-922.
Williams-Jones, A.E. and Heinrich, C.A., 2005. 100th Anniversary special paper:
vapor transport of metals and the formation of magmatic-hydrothermal ore
deposits. *Economic Geology* 100:1287-1312.
Yamaya Y, Alanis PKB, Takeuchi A et al (2013) A large hydrothermal reservoir
beneath Taal Volcano (Philippines) revealed by magnetotelluric resistivity
survey: 2D resistivity modeling. *Bull Volc* 75:729.

Yanagisawa, N., Fujimoto, K. and Hishi, Y., 2000. Sulfide scaling of deep-
geothermal well at Kakkonda geothermal field in Japan. Proceedings of the
World Geothermal Congress 2000:1969-1974.
Yardley BW (2005) Metal concentrations in crustal fluids and their relationship
to ore formation. *Econ Geol* 100:613-632.
Yardley BW, Bodnar RJ (2014) Fluids in the continental crust. *Geochem*
*Perspectives* 3:1-2.
Zajacz Z, Seo JH, Candela PA, Piccoli PM, Tossell JA (2011) The solubility of
copper in high-temperature magmatic vapors: a quest for the significance of
various chloride and sulfide complexes. *Geochim Cosmochim Acta* 75:2811-
2827.

**Acknowledgements**

1327 A.A. and O.M. acknowledge funding from Russian Science Foundation under
1328 grant # 16-17-10199. We thank G. Pokrovski for help with the chalcopyrite
solubility calculations. We thank J. Stimac for a primer on the architecture of
geothermal systems and S. Simmons, J. Hedenquist and N. White for critical
reviews of an earlier version of this manuscript.

**Authors' contributions**

JB developed the original brine lens mining concept in 2016. It was subsequently
refined through discussions with all other authors. BT performed the fluid
inclusion metal content calculations. AA performed the hydrodynamic
simulations; OM and IU adapted the results to include copper. All authors helped

to draft the final manuscript, gave final approval for publication and agree to be
held accountable for the work performed therein.

**Figure Captions**

Figure 1. Schematic of a transcrustal magmatic system showing features
described in the text. Note that proportion of melt and dimensions of crystals are
exaggerated for clarity.

Figure 2. Phase relations in the system NaCl-H₂O calculated using SOWAT
(Driesner and Heinrich 2007). Solid lines show isothermal projections of the
solvus that describe coexisting hypersaline liquid and low-salinity vapour. Upon
decompression a single-phase fluid of intermediate density will undergo phase
separation. The compositions of coexisting phases depend on pressure and
temperature; the compositions of coexisting vapour and hypersaline liquid at
1100 bars and 700 °C are shown for reference by the horizontal blue line. An
illustrative supercritical magmatic input fluid with 4 wt% NaCl_{eq} (as used in the
models) is shown by the vertical grey bar. The halite saturation field is shown in
red.

Figure 3: Three-dimensional phase maps of experimental and natural metal-
bearing, hypersaline fluid inclusions. Maps were generated from
photomicrographs of hypersaline liquid inclusions from: (a) experiments
(Tattitch and Blundy 2017); (b) a porphyry Cu deposit (Bajo de la Alumbrera;
Ulrich et al. 1999); and two deep geothermal systems (c,d) Lardarello (Fulignati
et al. 2018) and (e,f) Kakkonda (Fujimoto et al. 2000; Sasaki et al. 1998). All

inclusions show large halite daughter minerals due to high salinity, along with
numerous opaque daughter sulphide minerals. Liquid inclusions (a) and (b)
come from experimental systems with positive confirmation of chalcopyrite
daughter minerals and known Cu contents. Pixel maps of the inclusions yield
volume estimates for halite (red), sylvite (orange), and opaque daughter crystals
along with total volume estimates based on assuming elliptical inclusion
geometries and/or thicknesses given by bubble size and shape. Salinity estimates
for all inclusions based on volume estimates roughly match those reported from
fluid inclusion microthermometry. Estimates of the possible volume of
chalcopyrite (yellow) (34% Cu) or Cu-bearing Fe-sulfide (brown) (~3.5%) have
been compared to total fluid mass to determine the range of Cu concentrations
(in ppm). For comparison, copper concentrations from LA-ICPMS analyses of (a)
and (b) are given in square brackets. No explicit measure of uncertainty is
considered but large variations in inclusion thickness from those modelled
would result in unrealistic deviations in inclusion salinity from the measured
values.

Figure 4. Solubility of chalcopyrite (as ppm Cu in solution) in hydrothermal
solutions as a function of temperature based on equation (4) for three
representative salinities. The figure is based on an empirical fit to solubility
curves presented in Fig. 17e of Kouzmanov and Pokrovski (2012) for pyrite-
magnetite-hematite-saturated fluids with pH=5 at 0.3 to 1 kbar pressure.

Figure 5. Porosity-depth relationships in volcanic-hosted geothermal reservoirs,
and volcanic conduits. Geothermal well drillcore data (open and filled black

symbols) are redrafted from Fig. 46.14 of Stimac et al (2015), and distinguish
between dense (e.g. lavas, intrusives) and fragmental (breccias, tuffs,
volcanoclastics) rock types. The blue line shows well-log data for the Unzen
conduit from Fig. 2 of Ikeda et al (2008). The red squares show porosities of
individual drill-core rock samples from Kakonda (Fujimoto et al, 2000). The solid
red line shows the depth-porosity relationship adopted in the reservoir model of
McGuinness et al (1995). This model, wherein porosity is set homogeneous in a
succession of horizontal layers, decreasing with depth from 18.3% in the top
layer to 8.5% in the bottom layer, is based on the sonic and density log results of
well WD-1 series and on the analysis result of the electric logs of other Kakkonda
wells reported by Sakagawa et al. (2000). The mismatch between the Kakkonda
model and the drill-core rock samples shows that, at reservoir scale, the rocks
are significantly more porous than at the sample. The purple dashed line in the
depth-porosity relationship in the brine-lens modelling of Afanasyev et al
(2018).

Figure 6. Modelled distribution of salinity, Cu and temperature in the sub-
volcanic region for the reference scenario of Afanaysev et al. (2018) after 10,000
1406 years of degassing in terms of (from left to right): (a) bulk fluid salinity
expressed as molar fraction NaCl_{eq} ; (b) concentration of dissolved Cu (as ppm
Cu) for the no-sulfur case; (c) concentration of dissolved Cu (as ppm Cu) for the
unlimited-sulfur case; (d) precipitated Cu for the unlimited-sulfur case,
expressed as kg Cu per m^3 of rock; and (e) temperature ($^{\circ}\text{C}$). The conduit radius
is 0.33 km, centred on the vertical axis; input flux of fluid (738°C , 4 wt% NaCl_{eq} ,
700 ppm Cu) is 3×10^9 kg/yr originating from 7 km depth. Black dots in all panels

indicate halite precipitation. The volume containing hypersaline liquids with
>5000 ppm Cu is outlined in black in panels (b) to (e). Panels have identical
vertical and horizontal scales, and are axisymmetric about the left-hand axis.

Figure 7. Time (up to 10,000 yrs) evolution of total mass of dissolved (orange
line) and precipitated Cu (dark blue line) for unlimited-sulfur case, and dissolved
Cu (pale blue line) for the no-sulfur case. Models are for the reference scenario in
Figure 6. The total Cu input to the modelled domain is 21 Mt. Copper masses are
integrated across the reference volume defined in the text; there is very little
stored Cu in the modelled domain beyond this volume. Note the tendency
towards steady-state mass of dissolved Cu after 10,000 yrs.

Supplementary Figure. Comparison of model outputs 10,000 years after the
onset of fluid discharge from (right hand panel) our reference scenario model
and (left hand panel) from the dynamic permeability model of Weis et al (2012),
as provided in their supplementary material database S1. Vertical and horizontal
scales are in metres. Colours denote fluid salinity (in wt% NaCl_{eq}), including
precipitated halite, in the pore space. Where salinity exceeds 50%, due to the
presence of considerable halite, the scale is truncated and halite precipitation
shown with the pink shading. Our reference model set-up is as follows: $T = 738$
1433 °C (temperature of the injected fluid), $X_{\text{NaCl}} = 0.04$ (salinity of the injected fluid),
fluid flux = 3×10^9 kg/yr. For comparison, the equivalent parameters in the Weis
et al. (2012) model are: $T = 700$ °C, $X_{\text{NaCl}} = 0.10$, $\sim 1 \times 10^9$ kg/yr. The reader is
referred to Weis et al. (2012) for full model set-up details.

Figure 1 - Phase Maps of Brine Inclusions with Estimates of Salinity and Ranges of Possible Cu Concentrations

676x1166mm (72 x 72 DPI)

361x270mm (72 x 72 DPI)

Appendix B

17th May 2021

Dear Wolf,

Thank you for your editorial handling of our manuscript “*The Economic Potential of Metalliferous Sub-volcanic Brines*” submitted to *Open Science*, and your encouraging comments. We are very glad that you found the manuscript to be ‘interesting and stimulating’, as that was our aspiration in writing this as a *Perspectives* piece.

Reviewer #1 is very complimentary about the topic and the presentation, and there is little to modify in the manuscript. We now include mention of deleterious elements, such as As and Hg, and modify wording in one or two places, as suggested.

Reviewer #2 has a rather different take on the manuscript, on the one hand challenging the credibility of the concept of hot sub-volcanic brines, and on the other asserting that the economic geology community has known all about metal-rich brines for over 40 years. What we can agree on, however, is the undoubted scientific benefits of direct drilling into such systems, as the reviewer himself has advocated for magmatic systems (Lowenstern et al., 2017).

Overall, the reviewer makes a number of constructive comments that we have taken into account in preparing this revised manuscript. Most notably we have included a section on new, direct analysis of high-temperature brine fluid inclusions from two volcanoes (Montserrat and Larderello) to demonstrate their existence and metal endowment. The data we present are representative of a much larger dataset that we intend to publish elsewhere. *Perspectives* did not seem the place to present a lot of new data, but we reassure the reviewer that hot metalliferous brines occur in fluid inclusions at every volcano or geothermal well that we have studied so far. The new data are presented in a table, and the analytical methods and results described in sufficient detail to reassure the skeptical reader. In both examples the drill-core brines do indeed ‘contain thousands of ppm Cu’.

Here are our responses to the reviewer’s other comments (shown in italics for reference):

It is not a review in that it makes little effort to cite a lot of relevant older literature that first explored the importance of brines in porphyry ore deposits and geothermal systems.

Of course, there is a huge literature on porphyry systems, and our intention is not to review it all. That does not mean that we have not read any of it! Instead, we rely heavily on the excellent recent synthesis of Sillitoe (2010), embellished with selected additional references where appropriate.

The paper’s introduction envisions a future where humans can avoid the hundreds of near-surface ore-grade Cu porphyry deposits to instead extract Cu from much deeper, high-temperature intrusions that might be actively creating Cu-rich brines. At this point in time, it is extremely expensive to drill even a single hole in a 2-4 km-deep high-temperature environment... one that would sample only a tiny volume of liquid brine relative to the entire intrusion-related “liquid-lode” that would contain the envisioned 1.4 Mt of copper. Moreover, once syphoned off, any cooling of the fluid

would likely cause precipitation of both the ore and gangue (e.g., silica), reducing the ability of any tapped fluids to flow out of the well. The authors pass this off as an issue for the chemical engineers to solve, but it's a lot more daunting that they admit. . How can you suck all the liquid brine in the cupola up to the surface with only a few wells, or are the permeabilities envisioned to be sufficiently low that the intrusion would behave like an oil field?

These are good questions! The costs of drilling a hole must be tensioned against the economic benefits of the fluids that are extracted. Just how much metal can be extracted to the surface depends on how much brine is in the lens and the permeability of the host rock. We agree that small quantities of brine in a very tight, impermeable reservoir will not make for a big return on metals. However, there is permissive evidence from many geophysical and geothermal studies that porosities may be of the order of a few per cent and permeabilities quite high. We now present calculations for the Kakkonda deep well, the only one where saline fluids have actually been sampled down-hole, to show that the observed 1 S/m MT anomaly is consistent with ~9 vol% brine of the salinity of the recovered fluid

Extracting fluid to the surface, without scaling the entire well, is also a challenge, and one that we are well aware of. It is one of the key challenges being addressed in the supercritical geothermal realm, and some of those lessons could be applied to brine mining. Supercritical geothermal projects are well underway in a number of countries (Japan, Iceland, USA, Italy, Mexico). There is a significant literature on current research to address these challenges, a selection of which we now cite in the revised manuscript.

In the end, it comes down to: IF the common geophysical anomalies (MT) are actually revealing brines at the tops of intrusions, and IF such brines remain sufficiently S-deficient and high-temperature that they still contain high dissolved Cu, then maybe they COULD be tapped as metal ore IF we can figure out how to prevent the liquid from crystallizing AND IF we can drill into such environments cheaply and IF we can somehow suck the liquid out of this ductile, hot environment to the surface so that we wouldn't have to drill dozens or hundreds of holes. I'm sorry, but this just seems a bit too much, even for a perspective paper.

There are a succession of "if", "could" statements here that we will attempt to address in sequence.

Most of the authors of the many volcano MT studies that we have reviewed (there are about 40 or so) favour brine as the cause of the ~1 S/m conductive anomalies at 2-4 km depth. Just how much intergranular fluid, and of what salinity, are hard to gauge from MT alone, and so too is the transition to magma with depth. At Kakkonda, as we now show, brine is entirely consistent with the observed MT anomaly for the measured down-hole porosity. It is not our intention to critically evaluate each interpretation of brine, simply to state that there is a groundswell of support for brine in the volcano MT community.

The brines will contain elevated Cu if they are hot and/or sulfur deficient. This is self-evident. Clearly target volcanoes would be those where there is evidence for both. We now present fluid inclusion data from two geothermal wells to show that Cu-rich

brines do exist.

Crystallisation en route to the surface (or scaling) is a perennial geothermal problem, and the industry is awash with ideas of how to mitigate this problem, either by chemical modification of the extracted fluid, or coatings that inhibit scale nucleation. We are not quite a pessimistic as the reviewer about the prospect of solving these issues.

Whether the fluid will flow to the well-bore once drilled depends on the reservoir characteristics. Enhanced geothermal systems (EGS) stimulate the reservoir to improve hydraulic conductivity, and that possibility is also available for brine mining.

It is true that costs of drilling are currently high, largely because of the cost of the various metals used for casings and well-head structures. There is a lot of ongoing research into corrosion-resistant coating that can be used to reduce material costs and increase well lifetimes. Drilling 2-4 km deep holes is unlikely to ever become cheap, but then neither is excavating a giant underground mine or digging a 1 km deep pit.

I just don't think that this concept is really credible. It comes across more speculative than a typical scientific article, but with the veneer that it provides an important insight for our hoped-for carbon-free energy future.

The concept of mining sub-volcanic brines was developed by us in 2016, and we have spent the last five years attempting to de-risk it. We cannot present all of the de-risking data in a single article; our aim is just to lay out a framework for brine lens exploration and exploitation. This is not an invitation to invest, nor is it some half-baked speculation that we dreamed up one day and then deliberately green-washed. The reviewer may have underestimated the amount of thought and background reading that have gone into this work.

To be publishable, I think the authors need to be a bit more honest with the reader ...but to do so, may sufficiently undermine the premise of the article, that a) these metal-rich brines are as common as implied, and b) that economic recovery is plausible and advantageous compared with our current mines.

Ultimately, this is a *Perspectives* piece that introduces a new concept and musters supporting evidence where available. We consider it to be honest (certainly not dishonest!) and thought-provoking. Short of trying to drill and succeeding or failing, we are not sure what more we can do at this stage.

It wasn't clear to me how the model in this paper for Cu porphyry generation differs from Weis et al. (2012) or what it is adding that is new. Is this more an effort to reproduce aspects of their model so that you can predict the Cu concentration of the hydrosaline liquids?

Developing a new model is not a subject of this paper. Since CSMP++, the model used by Philip Weis, was (and still is) not freely available on the Internet, we had to design our own model to do the presented research. At the time, the manuscript was written, CSMP++ didn't have even a searchable webpage. Now, it has a webpage

(<http://igp-sim.ethz.ch/index.php/csmp>) but the download links there are not working. Therefore, we adapted our own hydrodynamic code, MUFITS, to do the research. Our results compare favourably to CSMP++ (see Supplementary Figure), allowing us to conduct an extensive parametric study for different temperatures and salinities of magmatic fluid (Afanasyev et al., 2018). As the reviewer correctly surmises, the results we present are adapted from the Afanasyev et al paper and post-processed with a Cu solubility model, based on published work, to make calculations of the potential Cu content of the brine lens

Though the paper is well written and with good organization, it's not always clear what you're aiming for, particularly with respect to the Cu porphyry model. Is the goal of the paper to stress the importance of transcrustal systems in Cu porphyry generation, or is the purpose to demonstrate the Cu-rich saline fluids are sitting there "ripe-for-the-picking." At times, it's not clear where you are heading, or why. Is the purpose of this paper REALLY just to promote the idea of brine-dissolved Cu for ore generation? Or is it to review some of the authors' other papers on Cu porphyries?

Our objective in describing the transcrustal magmatic system work is two-fold. The first accords with the aims of *Perspectives* articles in *Open Science*, namely "*Perspectives* take the form of a review that provides the reader with an overview of the subject and gives a personal insight into the advances and challenges the future may hold. *Perspectives* can be selective in their coverage rather than an in-depth review of an area." As written we think that article achieves this aim. On a scientific note, our aim is to place mineralisation into a wider magmatic context, as the end-product of a long-term and large-scale igneous geochemical cycle, and to explain how the idea of mining brines is a logical end-point of this thought process

You also seem conflicted about the importance of S. It's clear that a lot of people have argued that the fluids streaming through the ore shell contain a lot of S, either in a separate vapor phase or in supercritical fluids. But that would seem to undermine the argument that the hypersaline brines are sitting there awaiting sufficient S or temperature reduction to allow for precipitation of Cu sulfides. To me, the stability of Cu in the brines without precipitation would be very temporary.

The sulfide required to precipitate ore can come from a number of different sources: the original fluid that delivers the metals; a later addition of magmatic sulfur; the host rocks into which the fluids are injected. All of these possibilities have their proponents and detractors. We have tried to be fair in covering these possibilities, rather than asserting our particular preference for one or another. We suspect that this is where any apparent 'conflict' arises. As for the long-term stability of brines, what we can say with certainty, is that if they do not contain sufficient sulfide (or Fe) then they won't precipitate chalcopyrite or bornite.

The geophysics of course in another conundrum. These anomalies can be explained by saline fluids, by clays, by sulfides, and partly by temperature variations. It's much more exciting to interpret them as saline brines, but we need to admit that there are (I believe) zero examples of such anomalies that have been confirmed to be hypersaline brines by actual drilling. So this article is built on speculation after speculation. Could these anomalies be showing saline, Cu-bearing brines? Yes. Some could be. But I doubt a high percentage.

I am not sure why the reviewer asserts that there are zero examples of drilled magmatic brines. Most geothermal wells do not penetrate the brine lens because they sit below the brittle-ductile transition that marks the interface between the convective and conductive realms. However, at Kakkonda dense brines (39-55 wt% NaCl_{eq}) flowed into the wellbore at 3.5 km depth and 500 °C. The region from which these brines were recovered at Kakkonda lies within the magmatic reservoir below the convective-conductive interface, and has an MT conductive anomaly of 1 S/m, similar to that under many volcanoes where MT surveys have been used to infer brines. This is the closest we have come to drilling into a shallow conductive anomaly and recovering brine. Elsewhere there is evidence for hot brine fluid inclusions, even though there is no clear evidence of hypersaline liquid flowing into the well bore.

We are well aware of the other possibilities for conductive anomalies advanced by the reviewer, but all have some explicatory limitations. For example, conductive smectite clays require temperatures below about 250 °C, whereas sulfide minerals would have to be very well connected through the rock to create large conductive bodies. There are sulfides at Kakkonda and these contribute to the overall conductive anomaly, but by no means can explain all of it on the basis of the drilling results. Temperature alone cannot create conductive anomalies unless it is acting on an already conducting phase, such as brine.

I agree that a concept of transcrustal magmas is consistent with the large-scale scavenging of metals and volatiles from large crustal magma bodies to small near-surface ore concentrations. But this concept of Cu and Mo porphyries has been with us for decades. There are hundreds of studies that promote the concept that fluxing of volatiles from deep mafic magmas combined with convective degassing of intermediate magmas are critical to ore formation. There's a rich literature on alkaline magmas providing Cu. There's a literature on sulfide saturation with later breakdown and release of Cu. As the authors point out, though, it is a challenge to tie together this long-term whole-crust view of the magmatic system, with the geological evidence for fracturing, brecciation, and mineralization within a much shorter time scale. It appears that a long-term process of metal accumulation is ultimately finalized and preserved relatively quickly.

We recognise that there appear to be two contrasted timescales: long-term accumulation of metal-rich fluids; and short-term mineralization. The latter could be related to abrupt addition of sulfur, as envisaged by Blundy et al (2015), or rapid cooling and crystallization, for example due to a sudden pressure drop. For the purposes of this manuscript, the exact mechanism does not matter, provided that there was, at some stage, a reservoir of metal-rich brines. Our fluid inclusion analyses tend to confirm that possibility. The longevity of a reservoir is an interesting question. Our models (and those of Philip Weis) suggest lifetimes of the order >100,000 years.

I fully agree the rapid crystallization of Cu-sulfides during cooling of hydrothermal fluid could result in high Zn/Cu ratios in fluid inclusions at Kakkonda as well as minimal Cu in the extracted fluids. But the problem here, is you can't have it both ways... if indeed Cu is dumped out of solution during cooling, then it won't stay in solution in lenses of brine sitting in rocks straddling the ductile/brittle transition

(~400°C).

Catastrophic decompression (boiling) or very rapid cooling will drive sudden precipitation, which is why scales form in most geothermal systems. It is also likely to explain the low Cu at Kakkonda, as the reviewer agrees. Uncontrolled scale formation along the well-bore would be a problem for metal extraction from brine lenses, and we make that very clear in the manuscript. The question is whether brines can be recovered to the surface with their metal endowment intact, either by inhibiting scale formation or by entraining all precipitated particles. We consider that this in an opportunity for technological innovation rather than a cause for despair.

Other comments:

Line 66: The central point of Hedenquist and Lowenstern (1994).

We did already cite this paper in our manuscript. We reassure the reviewer that we have read it repeatedly and in detail. We now make an explicit citation at the relevant line.

Line 83: Worth Looking at Stanton (1994) Ore Elements in Arc Lavas, which is a phenomenal summary of metals concentrations in magmas focused.

The cited summary by Chiaradia makes the same point, and is more recent.

Line 112: There are lots of good references for alteration around intrusions, especially around porphyries.

The review by Sillitoe has a lot on alteration, and for reasons give above we would like to stick with that review, rather than run the risk of omitting some key references.

Line 230: If you add CO₂ into the mix, then it's pretty much impossible to have a truly supercritical situation. You'll get a CO₂-H₂O vapor and a liquid. That liquid may then later separately unmix to give you a hypersaline phase and a vapor, the latter or which could probably mix with the early exsolved CO₂-H₂O vapor. My main point is that in a lot of magmas at P<3000 bars, the presence of CO₂ will mean that Cl-bearing magmas will saturate with more than one H₂O-CO₂-HCl-NaCl fluid (not even accounting for the S!). The relevance of subcritical behavior was pointed out by Shinohara et al. (1989), Metrich and Rutherford (1992) and Lowenstern (1994). Roedder's 1984 book made clear the importance of brines in Cu porphyries, and the relevance of unmixing, albeit at lower temperatures.

We are aware of the effect of CO₂ on the H₂O-NaCl solvus. We are not sure that any of the suggested papers discuss the implications for mineralization in the way that is suggested, and we don't think that the presence of CO₂ will change greatly the ideas presented as hypersaline phase separation will still occur at low pressure. In fact, CO₂ greatly expands the depth range over which brines can form, just as the reviewer points out. We now make mention of the effect of CO₂ and include a citation.

Line 290: Why mention subduction zones here. How is this relevant? Geothermal systems are also found in a lot of non-arc-settings. Olkaria, Iceland, The Geysers,

Salton Buttes.

The point is well made and we rephrase the sentence.

Line 373: In reality, the discussion didn't really prove that Cu was enriched in these tapped high-T geothermal fluids.

We now include fluid inclusion analyses that show the metal contents in hypersaline fluids from two geothermal wells; Montserrat and Larderello. As noted above the data are illustrative and representative – we have many, many more (almost 1000!) such analyses that we will present elsewhere. The bottom line is that volcanic systems contain brine inclusions with base metal endowments comparable to PCDs.

Line 398: A lot may come down to whether there's sulfate in the brines as well.

This is true, and we describe the sulfur story in a bit more detail. We should really refer to 'sulfide-deficiency' rather than 'sulfur-deficiency' because if all sulfur in brines is S^{6+} then no chalcopyrite etc can precipitate.

Line 436; There are also a lot of high Vp anomalies under volcanoes, often linked to crystallized intrusions.

We are aware of the high-Vp features and their interpretation. However, our focus is on the relatively low Vp conduit regions.

Line 480: Not clear what you're really doing with the model, and why, and how this model is designed to be different than what Weis et al. (2012) were doing.

Philip Weis uses a numerical model CSMP++ that is not publicly available. Thus we adapted the MUFITS reservoir model, designed originally by Andrey Afanasyev, to do this research. Our numerical implementation of the model appeared to be very effective and allowed us to conduct an extensive parametric study for different temperatures and salinities of magmatic fluid (Afanasyev et al., 2018).

Line 488: Cloke and Kesler (1979) was the critical paper on recognizing the presence of halite.

These authors did not exactly find evidence for halite crystals in PCDs, they simply showed that fluids evolve along halite saturation curves, which infers halite saturation. Still, it is a useful reference and has now been included.

Line 490: Fournier (1999) in Economic Geology was an entire paper on the importance of lenses of brines in intrusions. And that paper followed up on another classic paper he wrote in 1987. Both are widely cited. This topic is not new.

This is certainly a pair of classic papers that were influential in introducing the importance of brines to the PCD community. We cite both papers now. However, it is worth pointing out that there is some difference to the way that Fournier envisaged the cap that holds the lens in place: both our models and those of Weis identify the importance of halite, in addition to rock rheology, in creating a seal.

Line 495: That would seem to imply that there's no incursion of a <400°C hydrothermal system for over a million years....?

The brine lens is isolated from the overlying hydrothermal convection region by the brittle-ductile transition, as envisaged by Fournier. Thus there are two isolated reservoirs, one containing relatively static, hot magmatic brine, and the other containing vigorously convecting meteoric (\pm magmatic) fluid. There is relatively little communication between the two reservoirs, consistent with observations of many geothermal wells. This is the reason that the brine lenses last so long.

Line 736: This ignores the fact that the best ore is supergene. Where the sulfide ore is oxidized, dissolved and re-precipitated into a high concentration ore tied to a paleo-water table.

We are very familiar with supergene ore, but confine ourselves in this paper to hypogene ores. We now make this distinction clear in the text.

Line 898: It's not clear to me how any recent developments have changed our understanding here. We've known about lenses of brines for 40 years relative to porphyries. We drilled into this environment at Kakkonda in the late 90s. The new modeling, first done by Weis et al. (2012) provided insights that the brines might be located in a small volume, and have relatively high Cu, but whether that Cu would remain un-precipitated for any significant amount of time is speculative. As is the thought that we could mine this liquid at a price even close to what we can get from surface deposits. The tie to MT surveys is still quite speculative as well.

The reviewer's concluding comments express his doubts that magmatic brines can ever be mined commercially, despite them being known about for 40 years. There have been some attempts to mine hot brines, for example for Zn and Li at Salton Sea, but these are quite different in origin to those envisaged here. Likewise, we can learn a lot from Kakkonda, but this was a geothermal exploration well drilled into granite, not a putative 'brine mine'. Neither of these projects drilled directly into conductive MT anomalies in the roots of dormant volcanoes with the express aim of recovering hot, magmatic brines. We are proposing that this environment is worth exploring; the reviewer thinks it would be a waste of time and money. This is just the sort of debate that *Perspectives* articles were designed to stimulate!

We hope that you will consider the revised manuscript addresses the issue raised by the reviewers and is now suitable for publication in Open Science. We think that the comments of the second reviewer, in particular, have helped us to produce a much improved manuscript.

Best,

Jon Blundy (on behalf of all authors)